   

EMBO Molecular Medicine

# RBMS1 orchestrates cardiac hypertrophy by facilitating CTTN splice-switching and sarcomere dynamics

Liangliang Li [iD] [1,8], Tianyu Li [1,8], Bin Wang [2,8], Jiayue Feng [1,8], Nan Zhang [3], Jing Zhang [1], Zhihui Niu [iD] [1], Wei Li [1], Huiying Gao [1], Qianqian Wang [1], Yang Liu [1], Yi Chen [1], Yixin Zhang [1], Yu Bian [1], Tengfei Pan [1], Siqi Sheng [4], Xuelian Li [1], Jinping Liu [iD] [5 ✉], Baofeng Yang [iD] [1,6 ✉] & Haihai Liang [iD] [1,6,7 ✉]

## Abstract

**Cardiac hypertrophy is one of the significant causes of heart failure and is closely related to the rising rate of hospitalization and readmissions. Given the diverse regulatory roles of alternative splicing in cardiovascular diseases, RNA-binding proteins have attracted increasing research attention. Here, for the first time, we discovered elevated expression of RBMS1 in heart tissues of patients with dilated cardiomyopathy and in mice with cardiac hypertrophy. We demonstrated that RBMS1 activated the PI3K/AKT signaling pathway by promoting the splicing CTTN to generate CTTN-Δe11 splicing isoform, resulting in cytoskeleton and sarcomere damage in cardiomyocytes. Additionally, pharmacological inhibition of RBMS1 by nortriptyline alleviated cardiac hypertrophy and heart failure. These results provide a new perspective for developing novel therapeutic approaches for cardiac hypertrophy and establish a theoretical basis for targeting RBMS1 in the clinical treatment of cardiac hypertrophy.**

**Keywords** Cardiac Hypertrophy; Alternative Splicing; RBMS1; CTTN; Sarcomere
**Subject Category** Cardiovascular System

## Introduction

Cardiac hypertrophy is defined as an increase in myocardial mass, representing the heart's adaptation to increased workload or pathological stimuli (Weeks and McMullen, 2011). Pathological cardiac hypertrophy arises in response to sustained pathological stresses, which commonly include hemodynamic overload (e.g., hypertension, valvular disease) or volume overload, but can also be triggered by diverse factors such as genetic mutations, neurohormonal dysregulation (e.g., excessive catecholamines, angiotensin II), metabolic disorders, or certain toxins. This maladaptive form of hypertrophy is strongly associated with cardiovascular disease (CVD) and is a major risk factor for the development of heart failure (HF), contributing significantly to global morbidity and mortality (Tham et al, 2015). Key features often observed in pathological hypertrophy include cardiomyocyte enlargement, accumulation of excess extracellular matrix, and, as it progresses towards HF, potential cardiomyocyte loss (Zhang et al, 2022b; Rockey et al, 2015). Despite advancements in evidence-based medicine and postoperative care, the risk of relapse and readmission for patients with pathological cardiac hypertrophy remains unacceptably high in many regions due to limitations of current treatment strategies (Heymans et al, 2023). Molecular mechanism studies at multiple levels are crucial and may facilitate the development of novel interventions and therapeutic approaches for this condition.

Alternative splicing (AS) is a ubiquitous post-transcriptional modification process that allows the production of multiple unique mRNA and proteins from a single gene and plays a key regulatory role in mammalian gene expression (Wright et al, 2022). The human genome encodes more than 20,000 protein-coding genes, more than 95% of which undergo alternative splicing (International, 2004; Nilsen and Graveley, 2010). The diversity of RNA-protein networks contributes to the complexity of the

[1]State Key Laboratory of Frigid Zone Cardiovascular Diseases (SKLFZCD), Department of Pharmacology (State Key Labratoray-Province Key Laboratories of Biomedicine-Pharmaceutics of China, Key Laboratory of Cardiovascular Research, Ministry of Education), College of Pharmacy, Harbin Medical University, Harbin 150081, China. [2]Department of Cardiovascular Ultrasound, Zhongnan Hospital of Wuhan University, Wuhan, China. [3]Department of Systems Biology, College of Bioinformatics Science and Technology, Harbin Medical University, Harbin 150081, China. [4]The Key Laboratory of Cardiovascular Disease Acousto-Optic Electromagnetic Diagnosis and Treatment in Heilongjiang Province, The First Affiliated Hospital of Harbin Medical University, Harbin, China. [5]Department of Cardiovascular Surgery, Zhongnan Hospital of Wuhan University, Wuhan, China. [6]Research Unit of Noninfectious Chronic Diseases in Frigid Zone (2019RU070), Chinese Academy of Medical Sciences, Harbin 150081, China. [7]Department of Nephrology, The Second Affiliated Hospital of Harbin Medical University, 150081 Harbin, China. [8]These authors contributed equally: Liangliang Li, Tianyu Li, Bin Wang, Jiayue Feng. ✉E-mail: liujinping@znhospital.cn; yangbf@ems.hrbmu.edu.cn; lianghaihai@ems.hrbmu.edu.cn

transcriptome and proteome (Ule and Blencowe, 2019). Recent studies have shown that aberrant alternative splicing caused by abnormal expression and/or activity of splicing factors is a major cause of CVD. RNA-binding motif protein (RBM), named by the Hugo Gene Nomenclature Committee, contains ribonucleoprotein motif, RNA binding motif, and RNA recognition motif, and is one of the typical splicing regulators involved in various cardiovascular diseases such as dilated cardiomyopathy (DCM), heart development, arrhythmia, and heart failure (Kornienko et al, 2023; Blech-Hermoni and Ladd, 2013; Gao and Dudley, 2013). Among RBM proteins, RNA binding motif single-stranded interacting protein 1 (RBMS1) has attracted much attention due to its important role in multiple pathophysiological processes (Zhang et al, 2021). RBMS1 exhibits considerable sequence and functional conservation between humans and mice, with a homology of 96.77%. However, the specific contribution of RBMS1 to the regulation of cardiac hypertrophy remains to be elucidated.

Cortactin (CTTN) is an essential actin-binding protein that plays a pivotal role in the regulation of the actin cytoskeleton and the secretion of extracellular matrix (ECM) proteins (Stradal and Costa, 2017). Cortactin is composed of several domains: a filamentous actin (F-actin) binding domain, an N-terminal acidic (NTA) domain, an alpha-helical region, a C-terminal Src homology 3 (SH3) domain, and a proline-rich domain (Uruno et al, 2001; Weaver, 2008). CTTN facilitates the stabilization and polymerization of actin by interacting with the actin-related protein (Arp)2/3 complex through its NTA and F-actin binding domains (Fregoso et al, 2023). Recently, CTTN has emerged as a vital regulatory through multiple molecular mechanisms such as intracellular motility, adherence, and invasion of microbial pathogens, participating in chronic obstructive pulmonary disease, arrhythmia, heart failure, and tumor predisposition and progression (Sharafutdinov et al, 2022; Bandela et al, 2021; Dun et al, 2014; Patel et al, 2013; Yin et al, 2017). Despite its emerging role in these conditions, the function of CTTN in cardiac hypertrophy remains undefined.

In this study, we found a significant upregulation of RBMS1 in heart tissues from DCM patients and in mouse models of cardiac hypertrophy. Overexpression of RBMS1 aggravated transverse aortic constriction (TAC)-induced cardiac hypertrophy, whereas cardiac-specific RBMS1 deficiency attenuated cardiac hypertrophy and improved cardiac dysfunction in TAC-induced mice. Furthermore, we uncovered that CTTN was spliced by RBMS1 to generate a variant, CTTN-Δe11, which promoted the disorganization of sarcomere and cytoskeleton in cardiomyocytes by activating the PI3K/AKT pathway. Targeting RBMS1 could offer a novel strategy to ameliorate pathological cardiac remodeling and improve outcomes in patients with cardiac hypertrophy.

# Results

## Increased expression of RBMS1 is associated with cardiac hypertrophy

To elucidate the potential role of RBMS1 in cardiac hypertrophy, we conducted a comprehensive analysis of RNA sequencing dataset (GSE135055). Notably, we observed a significant upregulation of RBMS1 in DCM patients (Fig. 1A), which was significantly correlating with NPPA and NPPB (Fig. 1B). Furthermore, the expression of RBMS1 was markedly increased in the hearts of patient with hypertrophic cardiomyopathy (HCM) from snRNA sequencing dataset (SCP1303) (Fig. 1C). Moreover, the protein and mRNA levels of RBMS1 were dramatically increased in DCM hearts (Fig. 1D,E). Additionally, significant increase in RBMS1 mRNA and protein were found in TAC mice (Fig. 1F,G), demonstrating that the significant increase of RBMS1 was associated with cardiac hypertrophy. To explore the mechanistic role of RBMS1 in hypertrophic hearts, we isolated cardiomyocytes from mice subjected to Angiotensin II (Ang II) and TAC treatments, revealing a progressive increase in RBMS1 expression (Fig. 1H–J). These results were also observed in neonatal mouse cardiomyocytes (NMCMs) challenged with Ang II and isoproterenol (ISO) (Fig. 1K–N).

## Cardiac-specific overexpression of RBMS1 aggravates cardiac hypertrophy

To investigate the effects of RBMS1 on cardiac hypertrophy, we conducted a targeted overexpression study using adeno-associated virus 9 (AAV9) driven by the cardiomyocyte-specific troponin T2 (cTnT) promoter. Mice were systemically injected via the tail vein with either AAV9-Vector or AAV9-RBMS1 and subsequently underwent sham or TAC surgery one week later, with evaluation performed after 8 weeks (Fig. 2A). We determined the over-expression efficiency of RBMS1 through immunofluorescence staining of RBMS1 and α-Actinin (Appendix Fig. S1A). Meanwhile, the overexpression efficiency of RBMS1 in cardiomyocytes isolated from myocardial tissue was validated using western blot and qRT-PCR (Fig. 2B,C). Strikingly, cardiac-specific overexpression of RBMS1 led to a downtrend in survival rate (Appendix Fig. S1B) and exacerbated heart function (Fig. 2D), as evidenced by a reduction in left ventricular ejection fraction (LVEF) and fractional shortening (LVFS), and an increase in left ventricular internal diameter (LVID) (Fig. 2E; Appendix Fig. S1C), with no effect on left ventricular posterior wall thickness (LVPW) (Appendix Fig. S1D). Moreover, cardiac-specific overexpression of RBMS1 significantly aggravated TAC-induced cardiac hypertrophy, characterized by an increase heart size, heart weight/body weight ratio, and heart weight/tibia length ratio (Fig. 2F,G). Masson's staining showed that TAC-mediated perivascular and interstitial fibrosis were significantly exacerbated in AAV9-RBMS1 mice (Fig. 2F,H; Appendix Fig. S1E). Furthermore, wheat germ agglutinin (WGA) staining further demonstrated that the cross-sectional area of cardiomyocytes was significantly increased in response to TAC mice with RBMS1 overexpression (Fig. 2F,I). As anticipated, disordered cardiac tissue structure and inflammatory infiltration in fibrotic areas induced by TAC were aggravated upon RBMS1 overexpression as evidence by hematoxylin-eosin (H&E) staining (Fig. 2F; Appendix Fig. S1G). Meanwhile, immunohisto-chemical (IHC) analysis also indicated an upregulation of hypertrophic markers in RBMS1-overexpressing mice (Appendix Fig. S1G). In addition, RBMS1 overexpression mice had a promoting hypertrophy reaction to TAC surgery based on the expression levels of myosin chain heavy β (β-MHC), atrial natriuretic polypeptide (ANP), and brain natriuretic peptide (BNP) (Fig. 2J–L).

Subsequently, to explore the direct regulatory effects of RBMS1 on cardiomyocyte hypertrophy in vitro, RBMS1 was overexpressed

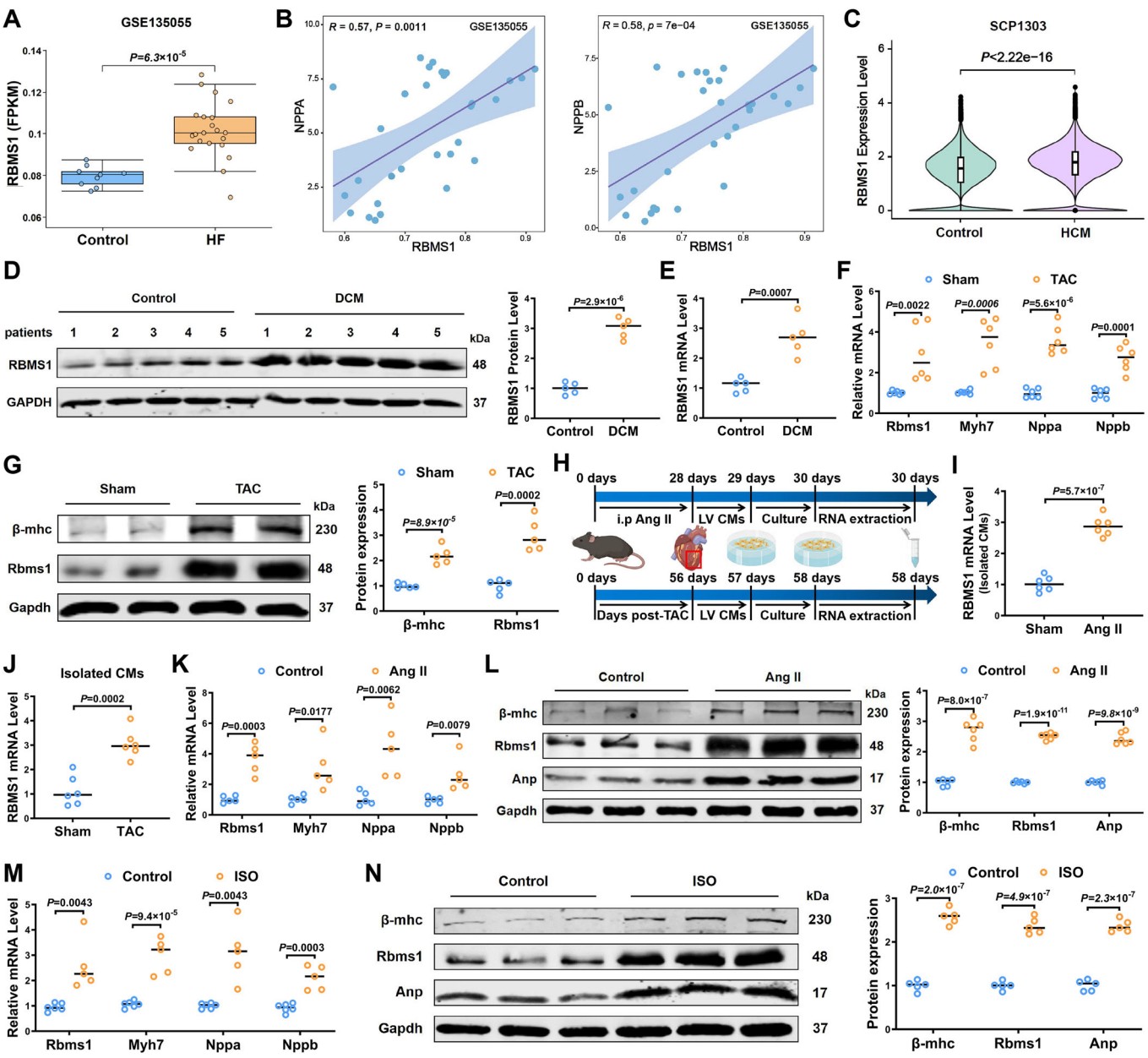

**Figure 1. RBMS1 expression is increased in cardiac hypertrophy.**

(A) Transcriptional expression of RBMS1 in RNA-seq data (GSE135055) including Control ($n = 9$) and HF patients ($n = 21$). Control (minimum: 0.072, Q1: 0.075, median: 0.08, Q3: 0.083, maximum: 0.087), HF (minimum: 0.069, Q1: 0.095, median: 0.1, Q3: 0.108, maximum: 0.128). (B) Pearson correlation analysis of the expression of RBMS1 with NPPA and NPPB in RNA-seq data (GSE135055). (C) Transcriptional expression of RBMS1 in snRNA-seq data (SCP1303) ($n = 16$). Control (minimum: 0, Q1: 1.045, median: 1.42, Q3: 1.90, maximum: 4.23), HCM (minimum: 0, Q1: 1.33, median: 1.69, Q3: 2.12, maximum: 4.59). (D, E) Western blot and qRT-PCR showing the protein and mRNA expression of RBMS1 in Control and DCM patients ($n = 5$). (F) Quantification of mRNA levels of RBMS1, MYH7, NPPA, and NPPB in Sham and TAC mice ($n = 6$). (G) Western blotting and quantification showing the protein expression of β-MHC and RBMS1 ($n = 5$). (H) Schematic diagram showing the isolation of cardiomyocytes from Ang II and TAC mice. (I, J) Quantification of mRNA levels of RBMS1 in cardiomyocytes isolated from Ang II and TAC mice ($n = 6$). (K) Quantification of mRNA levels of RBMS1, MYH7, NPPA, and NPPB in NMCMs induced by Ang II ($n = 5$). (L) Western blotting showing protein expression levels of β-MHC, RBMS1, and ANP ($n = 6$). (M) Quantification of mRNA levels of RBMS1, MYH7, NPPA, and NPPB in NMCMs induced by ISO ($n = 5$). (N) Western blotting showing protein expression levels of β-MHC, RBMS1, and ANP ($n = 5$). Data information: data are shown as mean ± SEM, $P$ values were analyzed with unpaired Student's t test (A, C, D, E, F for MYH7, NPPA, and NPPB, G, I, J, K for RBMS1, MYH7, and NPPA, L, M for MYH7 and NPPB, N) and Mann–Whitney U test (F for RBMS1, K for NPPB, M for RBMS1 and NPPA). 2.2e-16 is the threshold commonly used in R language output, indicating a highly statistically significant result (C). A dot represents an independent biological sample. Source data are available online for this figure.

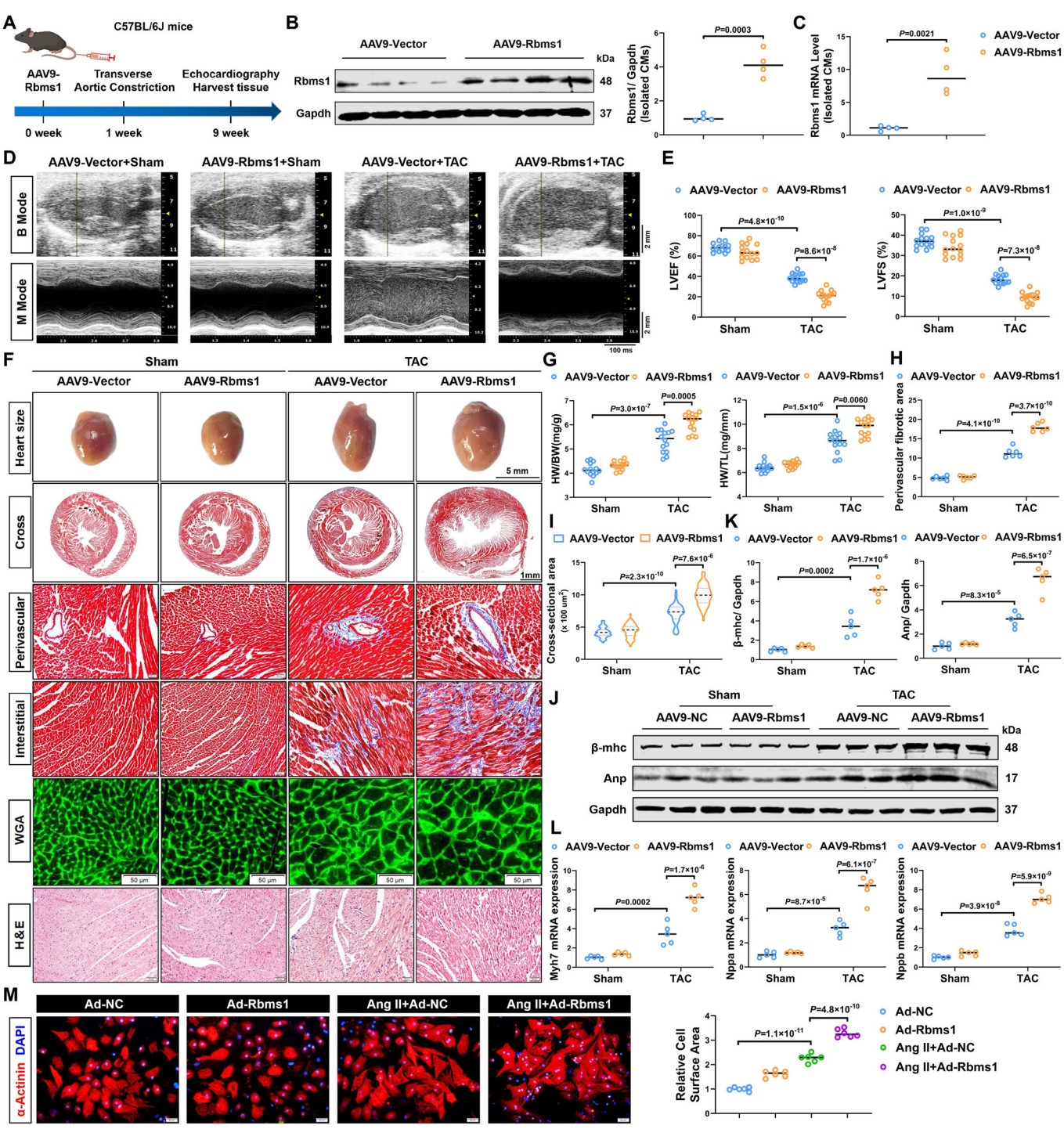

with adenovirus carrying RBMS1 plasmid (Ad-RBMS1) in NMCMs, followed by treatment with or without Ang II. Western blot and qRT-PCR analyses showed that the upregulation of hypertrophic markers, β-MHC, ANP, and BNP, induced by Ang II were markedly amplified upon RBMS1 overexpression (Appendix Fig. S1H,I). Moreover, RBMS1 overexpression enhanced cardiomyocyte hypertrophy in response to Ang II (Fig. 2M). Collectively, these findings manifest that cardiac-specific overexpression of RBMS1 exacerbates cardiac hypertrophy both in vivo and in vitro.

## RBMS1 deficiency mitigates cardiac hypertrophy in vivo and in vitro

To further evaluate the role of RBMS1 in cardiac hypertrophy, cardiomyocyte-specific RBMS1 knockout mice (RBMS1-cKO) were generated. Exon 3 of RBMS1 genes was deleted in these mice by CRISPR Cas9 (Appendix Fig. S2A) and the successful integration of loxp and MYH6-Cre sites was confirmed through mouse tail genotyping (Appendix Fig. S2B). RBMS1 knockout

**Figure 2. Overexpression of RBMS1 aggravates cardiac hypertrophy.**

(A) Experimental protocol of mice with AAV9 injection and TAC surgery, C57BL/6J mice were administered AAV9-RBMS1 via tail vein injection at a concentration of 5 × $10^{11}$ vg/mL. (B and C) Western blot and qRT-PCR showing protein and mRNA levels of RBMS1 in cardiomyocytes isolated from AAV9-RBMS1 mice ($n = 4$). (D) Representative echocardiography of WT mice injected with AAV9-Vector or AAV9-RBMS1 for 1 week and subsequently subjected to sham or TAC surgery for 8 weeks. (E) Quantification of LVEF ($n = 14$). (F) Heart size and histological sections of heart samples were stained with Masson staining, WGA staining, and H&E staining. For heart size, scale bar = 5 mm; for cross-section of the heart, scale bar = 1 mm; for Masson, WGA, and H&E staining, scale bar = 50 μm. (G–I) HW/BW, HW/TL ($n = 14$), perivascular fibrotic area ($n = 6$), and cross-sectional area of cardiomyocytes ($n = 70$) were examined. (J, K) Western blotting and quantification showing protein levels of β-MHC and ANP ($n = 5$). (L) Quantification of mRNA levels of MYH7, NPPA, and NPPB ($n = 5$). (M) Representative immunofluorescence staining of α-actinin and quantification in NMCMs transfected with Ad-RBMS1 in response to Ang II ($n = 6$). Scale bar = 50 μm. Data information: data are shown as mean ± SEM, $P$ values were analyzed with unpaired Student's t test (B, C) and one-way ANOVA test (E, G, H, I, K, L, M). A dot represents an independent biological sample. Source data are available online for this figure.

efficiency in cardiomyocytes isolated from RBMS1-cKO mice was substantiated by western blotting and qRT-PCR (Appendix Fig. S2C,D). The mice underwent TAC surgery and were monitored for 8 weeks (Fig. 3A). The survival rate of RBMS1-cKO mice exhibited a upward trend after TAC surgery (Appendix Fig. S2E). Concurrently, we observed a significantly improved cardiac function in RBMS1-cKO mice with TAC surgery (Fig. 3B), as evidenced by increased LVEF, LVFS, and LVPW;s (Fig. 3C; Appendix Fig. S2F,G), as alongside a reduction in LVID (Appendix Fig. S2H,I). However, RBMS1 deficiency did not affect the LVPW;d (Appendix Fig. S2J). Meanwhile, the TAC-induced enlargement of heart size, heart weight/body weight ratios, and heart weight/tibia length ratios were significantly decreased in RBMS1-cKO mice (Fig. 3D,E). RBMS1 deficiency attenuated cardiac fibrosis in the hypertrophic hearts as evidenced by Masson's staining (Fig. 3D,F; Appendix Fig. S2K). Furthermore, WGA staining revealed a significantly reduced cross-sectional area of cardiomyocytes in RBMS1-cKO mice following hypertrophic stimuli (Fig. 3D,G). Moreover, H&E staining showed that disordered cardiac tissue structure and inflammatory infiltration in fibrotic areas were reduced in RBMS1-cKO mice subjected to TAC surgery (Fig. 3D; Appendix Fig. S2L). Additionally, immunohistochemical staining showed that hypertrophic phenotypes were accompanied by decreased hypertrophic markers in RBMS1-cKO mice (Appendix Fig. S2M). Consistently, the TAC-induced upregulation of hypertrophic genes β-MHC, ANP, and BNP were significantly suppressed upon RBMS1 deficiency (Fig. 3H–J).

Furthermore, to elucidate the function of RBMS1 on cardiac hypertrophy in vitro, RBMS1 was silenced in NMCMs with small interfering RNA targeting RBMS1 (si-RBMS1). RBMS1 inhibition decreased the hypertrophic response to Ang II stimulation in NMCMs based on the the reduction of cardiomyocyte size, and dampened expression of hypertrophic markers (Fig. 3K; Appendix Fig. S2N,O). Taken together, these in vivo and in vitro findings underscore the pro-hypertrophic role of RBMS1 in cardiomyocyte hypertrophy.

## RBMS1 facilitates cardiac hypertrophy in hiPSC-CMs

To clarify the conserved function of RBMS1 in regulating cardiac hypertrophy among species, we induced human induced pluripotent stem cell-derived cardiomyocytes (hiPSC-CMs) and constructed a plasmid that overexpressing human RBMS1. We found that overexpression of RBMS1 enhanced the expression of Ang II-induced cardiac hypertrophy markers β-MHC, ANP, and BNP in

hiPSC-CMs (Fig. EV1A–C). Additionally, overexpression of RBMS1 facilitated Ang II-induced hypertrophy (Fig. EV1D). Similarly, we constructed a si-RBMS1 and downregulated the expression of RBMS1 in hiPSC-CMs (Fig. EV1E). It was discovered that silencing RBMS1 effectively reduced the Ang II-induced increase in the size of hiPSC-CMs (Fig. EV1F). Given the results of inhibiting RBMS1 expression in mice through genetic interventions, along with the function of RBMS1 in hiPSC-CMs, there is substantial evidence to suggest that RBMS1 holds significant potential as a therapeutic target for the management of cardiac hypertrophy and heart failure.

## RBMS1 induces morphological disarray through alternative splicing

To elucidate the molecular mechanism by which RBMS1 promotes cardiac hypertrophy, we inspected the cellular localization of RBMS1. Immunofluorescence staining and western blot further revealed a robust upregulation of nuclear RBMS1 in NMCMs, suggesting a nuclear function of RBMS1 in the hypertrophic process (Fig. 4A; Appendix Fig. S3A). Therefore, we hypothesized that RBMS1 was involved in multitudinous transcriptional regulation in the nucleus. Subsequently, we performed high-throughput RNA sequencing in NMCMs transfected with Ad-NC and Ad-RBMS1. A summary volcano plot depicted a substantial number of differentially expressed genes in response to RBMS1 overexpression (Appendix Fig. S3B), and these genes were markedly enriched in pathways related to cytoskeleton, ECM, and hypertrophic cardiomyopathy (Appendix Fig. S3C,D). Meanwhile, we identified that a total of 45,241 alternative splicing events that were altered after RBMS1 overexpression, with 71.5% of these events corresponding to skipping exons (Fig. 4B; Appendix Fig. S3E). Exhaustive alternative splicing patterns of differentially expressed gene can be found in Appendix Fig. S3F and the splicing targets regulated by RBMS1 were exhibited in Dataset EV1. Moreover, Gene ontology (GO) analysis further revealed that biological processes associated with differential splicing genes, including I band, Z disc, actomyosin, and cytoskeleton (Fig. 4C).

Transmission electron microscope and immunofluorescence staining showed that TAC-mediated sarcomere disorder including decreased sarcomere length and random orientation of sarcomere arrangement were significantly aggravated in AAV9-RBMS1 mice (Figs. 4D,E and EV2A). In contrast, the disordered sarcomere induced by TAC surgery were alleviated in RBMS1-cKO mice (Figs. 4F,G and EV2B). Additionally, immunofluorescence staining of ACTN2 and phalloidine demonstrated that overexpression of

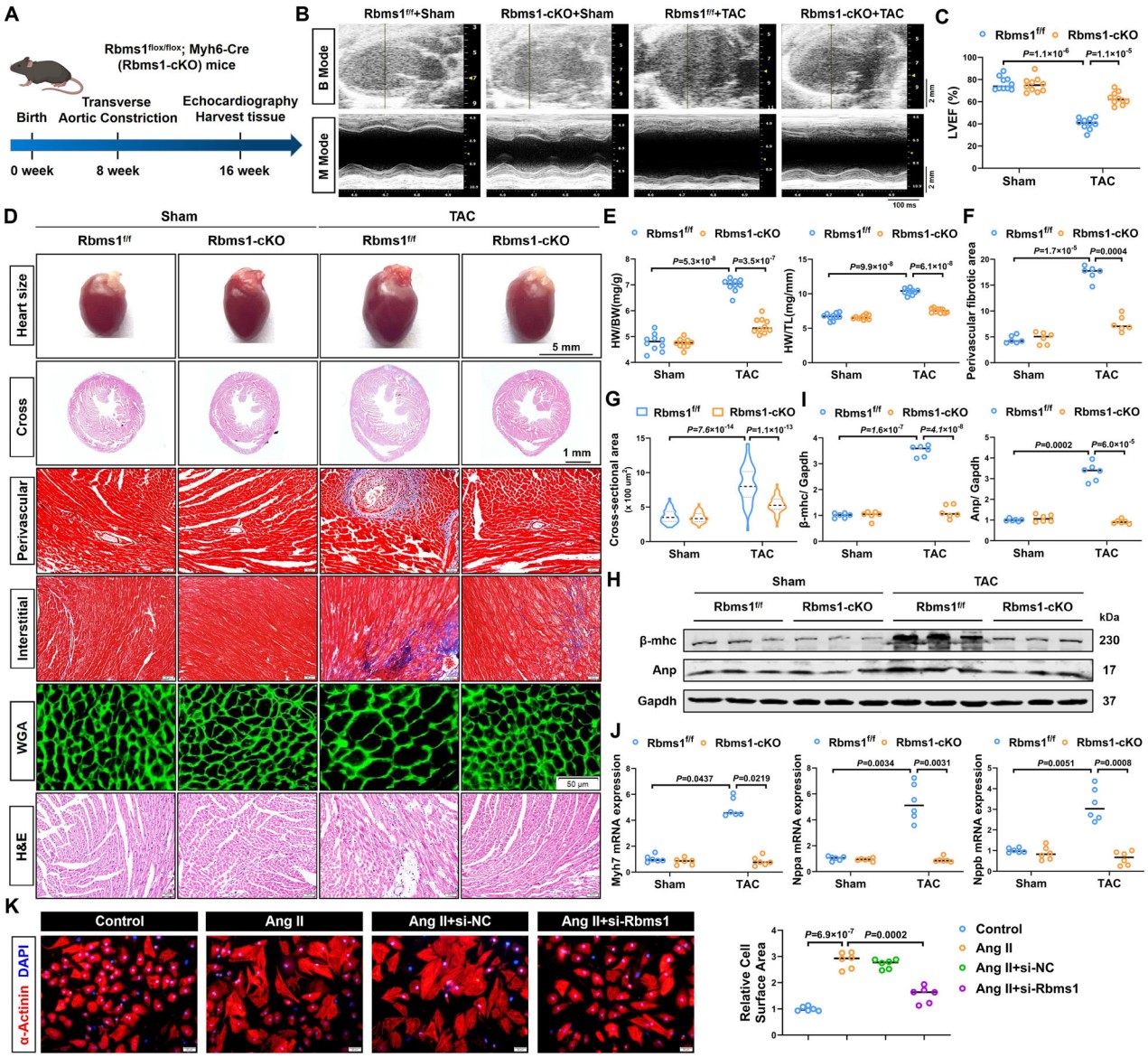

**Figure 3. Cardiac-specific RBMS1 deficiency attenuates cardiac hypertrophy.**

(A) Cardiomyocyte-specific RBMS1 knockout (RBMS1-cKO) mice were reared until 8 weeks of age, subjected to either sham or TAC surgery, and then maintained for an additional 8 weeks until tissue harvest. RBMS1-cKO mice were generated by crossing RBMS1$^{flox/flox}$ mice with MYH6-Cre mice. (B) Representative echocardiography of control and RBMS1-cKO mice with sham or TAC surgery. (C) Quantification of LVEF ($n = 10$). (D) Heart size and histological sections of heart samples were stained with Masson, WGA, and H&E staining. For heart size, scale bar = 5 mm; for cross section of the heart, scale bar = 1 mm; for Masson, WGA, and H&E staining, scale bar = 50 μm. (E–G) HW/BW, HW/TL ($n = 10$), perivascular fibrotic area ($n = 6$ mice), and cross sectional area of cardiomyocytes ($n = 50$) were examined. (H, I) Western blotting and quantification showing protein levels of β-MHC and ANP ($n = 6$). (J) Quantification of mRNA levels of MYH7, NPPA, NPPB ($n = 6$). (K) Representative immunofluorescence staining of α-actinin and quantification in NMCMs transfected with si-RBMS1 and subsequently treated with Ang II ($n = 6$). Scale bar = 50 μm. Data information: data are shown as mean ± SEM, $P$ values were analyzed with one-way ANOVA test (C, E, F, I, J for NPPA and NPPB, K) and Kruskal–Wallis test (G, J for MYH7). A dot represents an independent biological sample. Source data are available online for this figure.

RBMS1 significantly exacerbated the disarray of sarcomere and cytoskeleton in response to Ang II (Fig. EV2C,D). Coversely, knockdown of RBMS1 significantly preserved the alignment of sarcomere and cytoskeleton upon Ang II induction (Fig. EV2E,F). Furthermore, overexpression of RBMS1 in hiPSC-CMs effectively decreased sarcomere length and aggravated the random orientation of sarcomere arrangement in response to Ang II stimulation (Fig. 4H). Similarly, it was discovered that inhibition of RBMS1

could availably restore the shortened sarcomere length and the disordered cytoskeleton induced by Ang II (Fig. 4I).

## RBMS1 binds to and regulates the alternative splicing of CTTN

To explore the mechanism of RBMS1 in cardiac pathological processes, we further analyzed the splicing genes regulated by

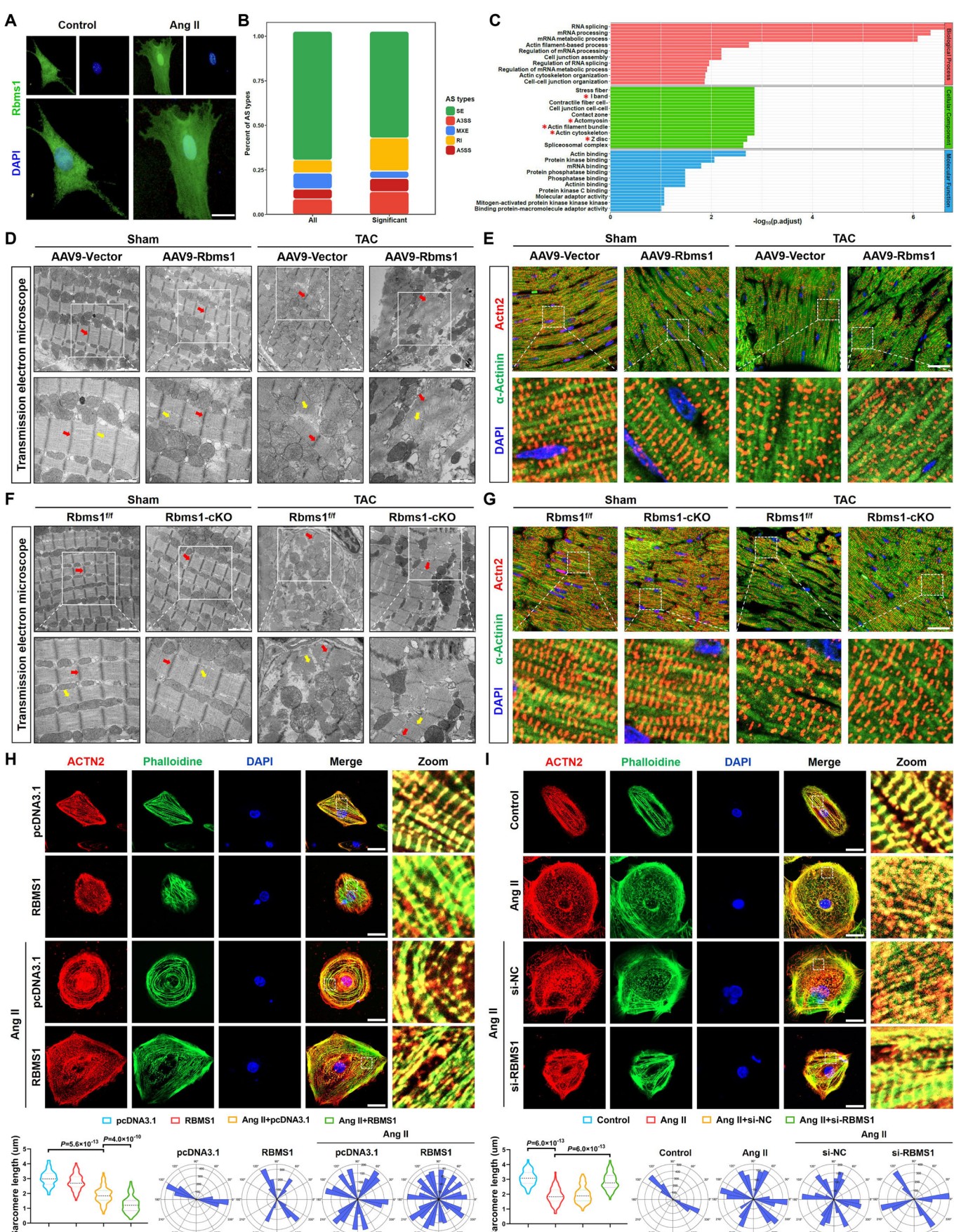

**Figure 4. RBMS1 promotes the disorganization of sarcomere and cytoskeleton in CMs.**

(**A**) Representative immunofluorescence staining of RBMS1. Scale bar = 20 μm. (**B**) Percent of AS types in NMCMs transfected with Ad-RBMS1 ($n = 3$). Skipping Exon (SE), Alternative 3′ Splice Site (A3SS), Mutually Exclusive Exons (MXE), Retained intron (RI), Alternative 5′ Splice Site (A5SS). (**C**) GO analysis of differential splicing genes. (**D**) Transmission electron microscope showed the disorganization of sarcomeres in RBMS1 overexpression mice treatment with TAC surgery ($n = 4$), scale bar = 1 and 2 μm. The red and yellow arrows respectively represent the Z-disc and M-band. (**E**) Immunofluorescence of α-Actinin and ACTN2 in RBMS1 overexpression mice, scale bar = 20 μm. (**F**) Transmission electron microscope showed the disorganization of sarcomeres in RBMS1-cKO mice, scale bar = 1 and 2 μm. The red and yellow arrows respectively represent the Z-disc and M-band. (**G**) Immunofluorescence of α-Actinin and ACTN2 in RBMS1-cKO mice, scale bar=20 μm. (**H**) Immunofluorescence staining of ACTN2 and phalloidine showed the sarcomere and cytoskeleton of hiPSC-CMs and quantification of sarcomere length and representative polarity histogram of sarcomere organization ($n = 70$), scale bar = 20 μm. (**I**) Immunofluorescence staining of ACTN2 and phalloidine showed the sarcomere and cytoskeleton of hiPSC-CMs and quantification of sarcomere length and representative polarity histogram of sarcomere organization ($n = 70$), scale bar = 20 μm. Data information: data are shown as mean ± SEM, $P$ values were analyzed with one-way ANOVA test (**H, I**), polarity histograms were generated by MATLAB R2024b (**H, I**). Source data are available online for this figure.

RBMS1. The volcano plot illustrated significant alterations in the expression of the differential splicing genes in response to RBMS1 overexpression (Fig. 5A). Combining volcanic map and GO analysis to identify the differential splicing genes enriched in HCM and DCM, we identified five potential target genes subjected to RBMS1 regulation (Fig. 5B), among which CTTN exhibited a distinct exon 11 skipping pattern (CTTN-Δe11), and which exhibited the most pronounced upregulation in RBMS1-overexpressed NMCMs (Fig. 5C; Appendix Fig. S4A–D). snRNA sequencing dataset (SCP1303) analysis showed that CTTN was predominantly expressed in cardiomyocytes (Fig. 5D) and was downregulated in HCM patient samples (Appendix Fig. S4E). Further study showed an inverse relationship between CTTN expression and RBMS1 level, with the high CTTN expression group (CTTN$^+$) displaying lower RBMS1 expression (Appendix Fig. S4F). Gene sets associated with heart morphogenesis, cardiac cell development, cytoskeleton, hypertrophic cardiomyopathy, and cardiac muscle contraction were derived from GO terms and KEGG pathways. Scoring these gene sets in cardiomyocytes revealed that the CTTN$^+$ group consistently exhibited higher scores (Appendix Fig. S4G,H). Furthermore, CTTN was down-regulated in cardiomyocytes with high expression of RBMS1 (RBMS1$^+$) (Appendix Fig. S4I). Thus, we hypothesized that the aberrant splicing of CTTN by RBMS1 led to cytoskeleton and sarcomere disorder in cardiomyocytes. Consistent with this hypothesis, CTTN-Δe11 was found to be upregulated in hearts induced by TAC (Fig. 5E; Appendix Fig. S4J), as well as in NMCMs stimulated by Ang II (Fig. 5F). Moreover, the expression of CTTN-Δe11 was upregulated and significantly correlating with RBMS1 in DCM patients (Fig. 5G,H). To substantiate these findings, RNA pull-down and RNA immunoprecipitation (RIP) assays were performed and confirmed the direct binding of RBMS1 protein to CTTN mRNA (Fig. 5I,J). To explore the potential mechanism of CTTN splicing regulated by RBMS1, we designed a small interfering RNA to knock out CTTN (Appendix Fig. S4K), and constructed a minigene encompassing exon 10 to exon 12 of the CTTN gene (Fig. 5K; Appendix Fig. S4L). As shown in Fig. 5K, RBMS1 could similarly splice into exogenous minigene and increase CTTN-Δe11 expression after CTTN knockout. To further determine the specific mechanism of RBMS1 binding and splicing of CTTN, we utilized the RBPmap resource (rbpmap.technion.ac.il/index.html) to identify two binding sites between RBMS1 and intron 10 of CTTN. By mutating the two binding sites in a mutant minigene construct (Fig. 5L; Appendix Fig. S4M), we found that RBMS1 could no longer induce splicing of the exogenous mutant

minigene following CTTN deletion. These results unravel the alternative splicing events regulated by RBMS1 in cardiac hypertrophy.

## CTTN-Δe11 ablation alleviates cardiac hypertrophy by preserving sarcomere and cytoskeletion intergrity

To evaluate the function of CTTN-Δe11 on cardiac hypertrophy, CTTN-Δe11 was silenced in cardiomyocytes by tail vein injection of AAV9-cTnT-short hairpin RNA targeting CTTN-Δe11 (sh-Δe11), followed by TAC to induce hypertrophy for 8 weeks (Fig. 6A). As shown in Appendix Fig. S5A, RT-PCR validated the efficient silencing of CTTN-Δe11 in isolated cardiomyocytes. Our findings demonstrated that cardiac-specific silencing of CTTN-Δe11 increased the survival rate of TAC mice (Appendix Fig. S5B). Echocardiography assessments showed that CTTN-Δe11 knockdown significantly improved cardiac function in response to TAC surgery (Fig. 6B), evidenced by increased LVEF, LVFS, and LVPW, and decreased LVID, heart size, HW/BW, and HW/TL (Fig. 6C–E; Appendix Fig. S5C–G). Meanwhile, morphological analyses showed that CTTN-Δe11 deficiency attenuated TAC-induced cardiac fibrosis (Fig. 6D,F; Appendix Fig. S5H), reduced cardiomyocyte cross-sectional area (Fig. 6D,G), mitigated cardiac remodeling and inflammatory infiltration (Appendix Fig. S5I,J), and inhibited hypertrophy pathological gene expression (Appendix Fig. S5I). Additionally, the increased expression of pathological genes induced by TAC were also significantly counteracted in CTTN-Δe11 deficiency mice (Fig. 6H,I; Appendix Fig. S5K). Transmission electron microscope and immunofluorescence staining manifested that TAC-induced decreased sarcomere length and random orientation of sarcomere arrangement of myocardial tissue was significantly alleviated in the absence of CTTN-Δe11 (Fig. 6J,K; Appendix Fig. S5L).

To further delineate the role of CTTN-Δe11 on cardiac hypertrophy in vitro, we developed small interference RNAs to silence the expression of CTTN-Δe11 (si-Δe11) in NMCMs (Appendix Fig. S6A). As anticipated, CTTN-Δe11 knockdown effectively restrained Ang II-induced cardiomyocyte hypertrophy, as evidence by the reduced expression of β-MHC, ANP, and BNP (Appendix Fig. S6B,C) and a concomitant decrease in cardiomyocyte size (Fig. 6L). Furthermore, immunofluorescence staining demonstrated that CTTN-Δe11 knockdown significantly corrected the Ang II-induced disordered sarcomere and cytoskeleton (Appendix Fig. S6D,E). In contrast, forced expression of CTTN-Δe11 with adenovirus carrying CTTN-Δe11 plasmid (Ad-Δe11) in NMCMs exacerbated Ang II-induced hypertrophic responses

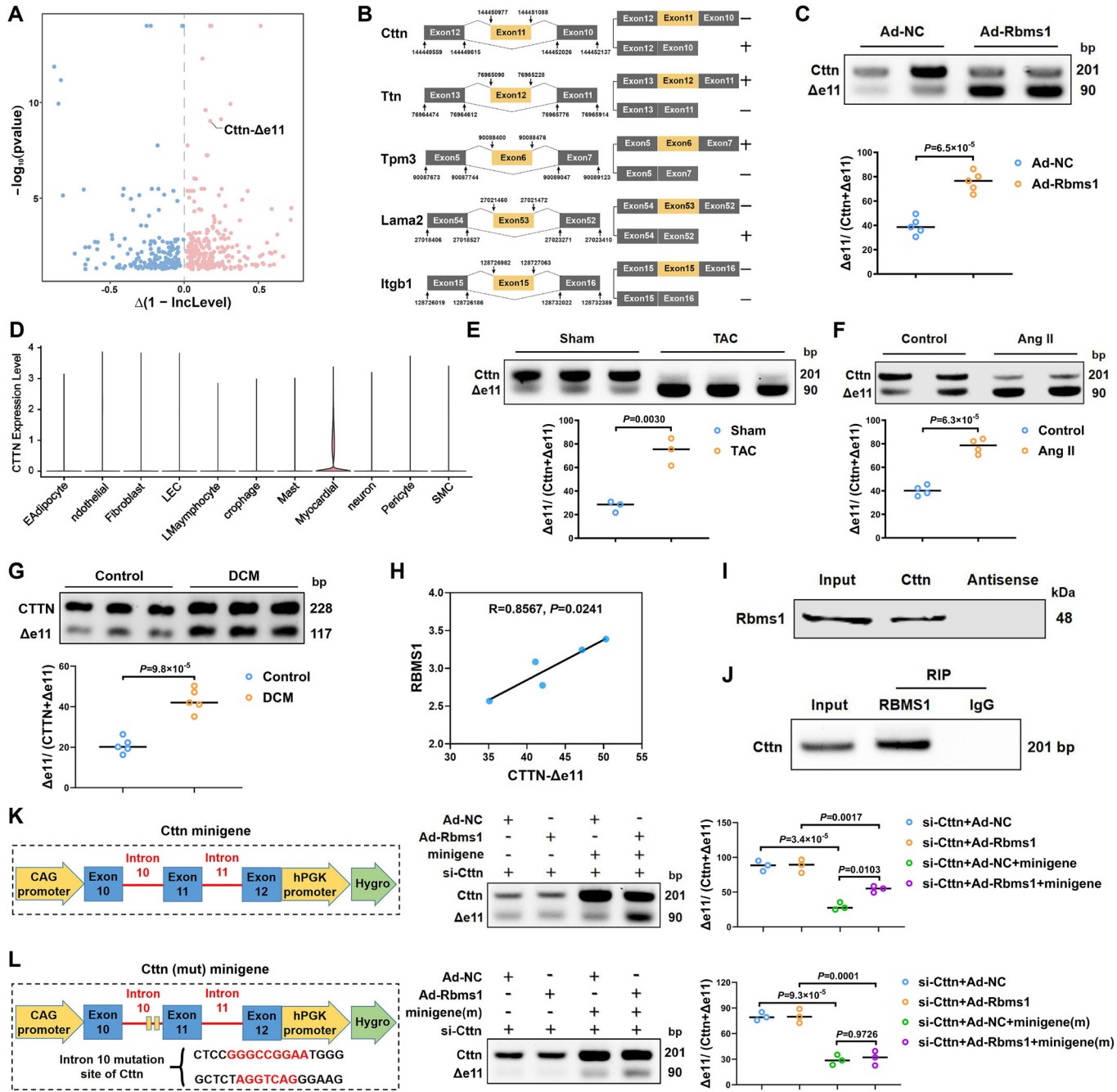

**Figure 5. RBMS1 binds to CTTN pre-mRNA and regulates the skipping of exon 20 through alternative splicing.**

(A) Volcanic plot elucidated the differential splicing genes. (B) Schematic diagrams showed the detailed information of differential splicing genes. (C) Splicing pattern and quantification of CTTN in NMCMs transfected with Ad-RBMS1 ($n = 5$). (D) snRNA-seq dataset (SCP1303) analysis of the distribution of CTTN expression in various cardiac cells. (E) Splicing pattern and quantification of CTTN in sham and TAC mice ($n = 3$). (F) Splicing pattern and quantification of CTTN in NMCMs treatment with Ang II ($n = 4$). (G) Splicing pattern and quantification of CTTN in Control and DCM patients ($n = 5$). (H) Pearson correlation analysis of the expression of CTTN-Δe11 with RBMS1 in DCM patients. (I and J) RNA pull-down and RIP assay showed the binding of RBMS1 protein and CTTN mRNA in NMCMs. (K) Left, Schematic diagram of minigene with the exon 10 to exon 12 of CTTN. Right, the splicing pattern and quantification of CTTN ($n = 3$). (L) Left, Schematic diagram of CTTN mutation minigene with RBMS1-binding sites. Right, the splicing pattern and quantification of CTTN ($n = 3$). Data information: data are shown as mean ± SEM, $P$ values were analyzed with unpaired Student's t test (C, E, F, G) and two-way ANOVA test (K, L). A dot represents an independent biological sample. Source data are available online for this figure.

(Appendix Fig. S7A–C). Meanwhile, forced expression of CTTN-Δe11 enhanced the enlarged cardiomyocyte size induced by Ang II (Appendix Fig. S7D). Moreover, immunofluorescence staining showed that overexpression of CTTN-Δe11 exacerbated the organization of sarcomere and cytoskeleton triggered by Ang II (Appendix Fig. S7E–G). Taken together, these findings suggest that inhibition of CTTN-Δe11 mitigates cardiac hypertrophy via safeguarding the sarcomere and cytoskeleton of cardiomyocytes.

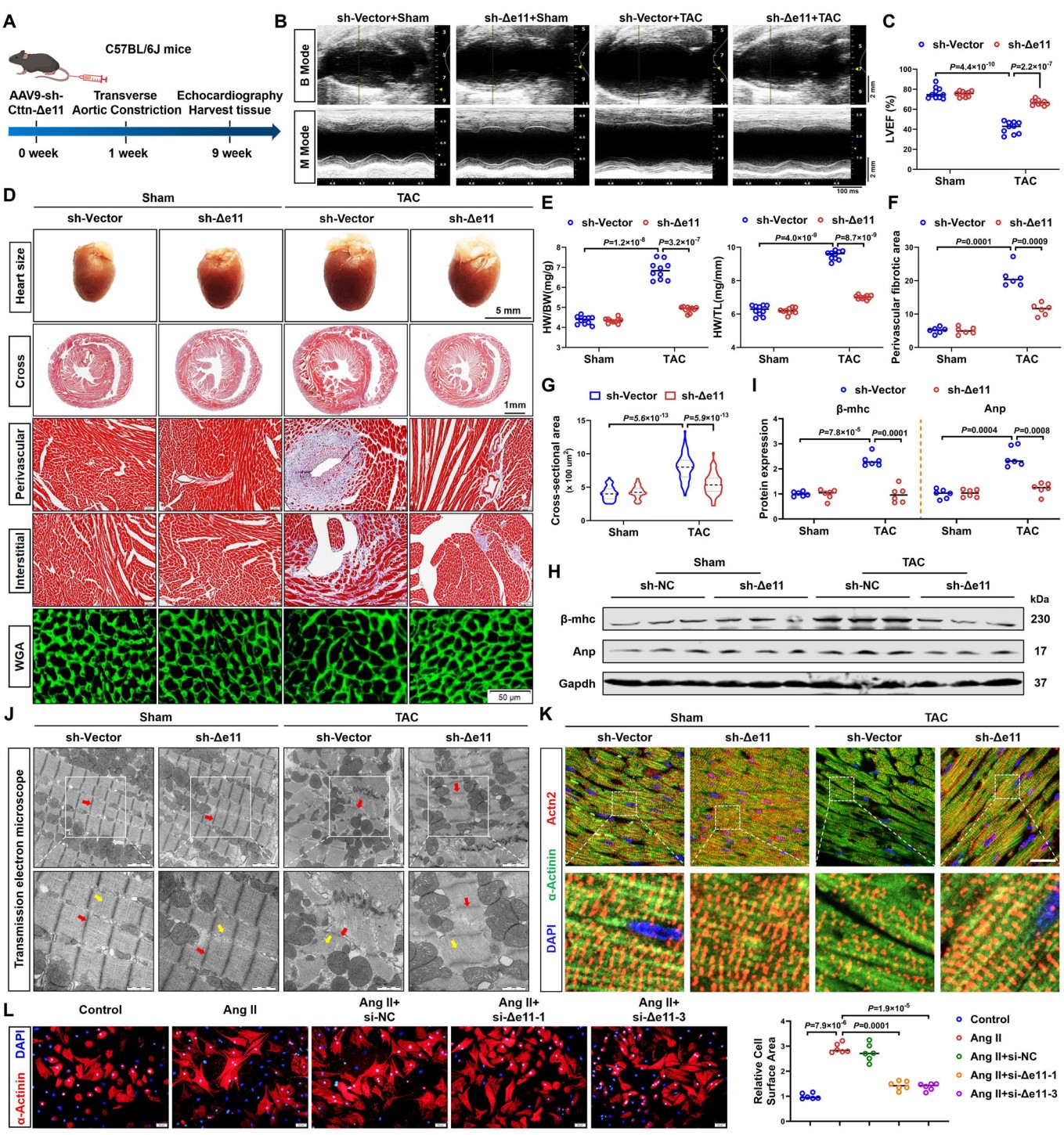

**Figure 6. Cardiac-specific deletion of CTTN-Δe11 attenuates cardiac hypertrophy.**

(A) Experimental protocol of mice with AAC9 injection and TAC surgery. C57BL/6J mice were administered AAV9-sh-CTTN-Δe11 via tail vein injection at a concentration of $1 \times 10^{11}$ vg/mL. (B) Representative echocardiography of mice. (C) Quantification of LVEF, LVFS ($n = 10$). (D) Heart size and histological sections of heart samples were stained with Masson and WGA staining. For heart size, scale bar = 5 mm; for cross section of the heart, scale bar = 1 mm; for Masson and WGA staining, scale bar = 50 μm. (E–G) HW/BW, HW/TL ($n = 10$), perivascular fibrotic area ($n = 6$), and cross sectional area of cardiomyocytes ($n = 70$) were examined. (H, I) Western blotting and quantification showing protein levels of β-MHC and ANP ($n = 6$). (J) Transmission electron microscope showed the disorganization of sarcomeres ($n = 4$), scale bar = 1 and 2 μm. The red and yellow arrows respectively represent the Z-disc and M-band. (K) Immunofluorescence showed the disorganization of sarcomeres ($n = 6$), scale bar = 20 μm. (L) Representative immunofluorescence staining of α-actinin and quantification in NMCMs transfected with si-Δe11 in response to Ang II ($n = 6$), scale bar = 50 μm. Data information: data are shown as mean ± SEM, $P$ values were analyzed with one-way ANOVA test (C, E, F, G, I) and two-way ANOVA test (L). A dot represents an independent biological sample. Source data are available online for this figure.

## RBMS1 promotes cardiac hypertrophy through regulation of CTTN alternative splicing

Subsequently, we assessed whether RBMS1 is involved in cardiac hypertrophy by splicing of CTTN. Mice were respectively injected via the tail vein with either AAV9-RBMS1 or AAV9-sh-CTTN-Δe11, followed by TAC surgery and an 8-week observation period (Fig. 7A). As shown in Appendix Fig. S8A, CTTN-Δe11 inhibition significantly increased the survival rate of mice with RBMS1 overexpression. Consistently, the detrimental cardiac dysfunction induced by RBMS1 overexpression was significantly ameliorated by silencing CTTN-Δe11 in response to TAC treatment (Fig. 7B,C; Appendix Fig. S8B–G), with no significant change observed in the LVPW;d (Appendix Fig. S8H). Moreover, CTTN-Δe11 silencing in cardiomyocytes of RBMS1-overexpressing mice significantly mitigated the pathological manifestations of cardiac hypertrophy induced by TAC surgery (Fig. 7D,E), reduced cardiac fibrosis (Fig. 7D,F; Appendix Fig. S8I), improved cardiac architecture and inflammatory infiltration (Appendix Fig. S8J,K), and decreased cardiomyocyte cross-sectional area (Fig. 7D,G). Additionally, the upregulation of hypertrophic markers induced by RBMS1 was significantly attenuated following CTTN-Δe11 inhibition in response to hypertrophic stimuli (Fig. 7H,I; Appendix Fig. S8J,L). Moreover, transmission electron microscope and immunofluorescence staining manifested that the reductive sarcomere length and non-directional array of sarcomere induced by both TAC and RBMS1 overexpression were significantly attenuated upon CTTN-Δe11 silencing (Fig. 7J,K; Appendix Fig. S8M). Meanwhile, the upregulation of CTTN-Δe11 induced by TAC and RBMS1 was effactually eliminated in CTTN-Δe11 inhibition mice (Appendix Fig. S8N).

Consistently, silencing CTTN-Δe11 significantly reversed the upregulation of pathological hypertrophy gene expression (Fig. EV3A,B) and reduced the enlarged cardiomyocyte size in response to Ang II treatment (Fig. EV3C,D). Meanwhile, the disordered sarcomere and cytoskeleton of cardiomyocytes induced by both Ang II and RBMS1 were dramatically eliminated after CTTN-Δe11 knockdown (Fig. EV3E–G). Taken together, these results demonstrate that the regulation of cardiac hypertrophy by RBMS1 is relied on the splicing of CTTN to generate CTTN-Δe11.

## PI3K/AKT is indispensable for the pro-hypertrophic function of CTTN splicing regulated by RBMS1

To uncover the underlying pathway of RBMS1 in pathological cardiac hypertrophy, we utilized WGCNA approach to construct RBMS1 co-expression module based on RNA sequencing dataset (GSE135055) (Appendix Fig. S9A,B), and identified 279 genes (MEyellow) co-expressed with RBMS1 were the strongest association with HF (Appendix Fig. S9C). The cardiomyocytes of SCP1303 dataset were scored using AddmoduleScore function for yellow module genes based on GSE135055 data, and high and low groups were scored using the mean value (RBMS1$^+$ and RBMS1$^-$) (Fig. 8A). The FindMarkers function was used to seek marker genes in RBMS1$^+$ cardiomyocytes, and functional enrichment analysis showed that marker genes were significantly enriched in the PI3K/AKT pathway (Fig. 8B,C). As shown in Fig. 8D, overexpression of RBMS1 activated proteins within the PI3K/AKT pathway. Conversely, knockout of RBMS1 reduced the TAC-induced upregulation of p-PI3K and p-AKT (Fig. 8E). Gene ontology analysis revealed the relationship between differential splicing genes regulated by RBMS1, such as HCM, DCM, and PI3K/AKT pathway (Fig. 8F; Appendix Fig. S9D). Previous studies have implicated CTTN in promoting esophageal squamous cell carcinoma via the PI3K/AKT pathway (Du et al, 2009). Coherently, the activated PI3K/AKT pathway proteins were significantly repressed by CTTN-Δe11 deficiency (Appendix Fig. S9E). Thus, we speculated RBMS1 activated PI3K/AKT pathway through generating CTTN-Δe11 to regulated cardiac hypertrophy. In addition, the activation of PI3K/AKT pathway induced by RBMS1 in response to TAC was significantly reduced in CTTN-Δe11 inhibition mice (Fig. 8G).

To elucidate the regulatory effect of RBMS1/CTTN-Δe11 axis on the PI3K/AKT signaling pathway, our findings demonstrated that RBMS1 promoted the expression of P100α, while knockdown of Δe11 significantly suppressed P100α production (Fig. 8H; Appendix Fig. S9F). Furthermore, PI3K/AKT pathway was inactivated by Alpelisib, which is a specific inhibitor of P100α, competitively binds to the ATP-binding pocket of P110α, thereby inhibiting the generation of PIP3. Alpelisib significantly reversed the upregulation of pathological hypertrophy gene expression (Appendix Fig. S10A,B) and reduced the enlarged cardiomyocyte size (Appendix Fig. S10C) induced by RBMS1 and Ang II. Meanwhile, the upregulation of p-PI3K and p-AKT induced by RBMS1 overexpression were significantly decreased by Alpelisib in response to Ang II (Fig. 8I). Furthermore, Alpelisib drastically attenuated the RBMS1-induced disarray of sarcomere and cytoskeleton (Fig. 8J; Appendix Fig. S10D). Similarly, Alpelisib could also decrease the expression of hypertrophy genes (Fig. EV4A,B), reduce the area of enlarged cardiomyocytes (Fig. EV4C,D), elongate the sarcomere length, and orient in the fiber direction in response to CTTN-Δe11 overexpression (Fig. EV4E–G). Altogether, the results show that RBMS1 activates the PI3K/AKT pathway via splicing CTTN to promote sarcomere chaos and cardiac hypertrophy.

## Nortriptyline-mediated pharmacological inhibition of RBMS1 alleviates cardiac hypertrophy and heart failure

Previous studies have shown that Nortriptyline (NTP) possessed the ability to inhibit the expression of RBMS1 (Zhang et al, 2021). To further clarify RBMS1 as a potential therapeutic target for cardiac hypertrophy, mice were administered NTP (5 mg/kg) intraperitoneally, alongside the positive control drug Valsartan (VAL) (10 mg/kg) one week after TAC surgery (Appendix Fig. S11A). The efficiency of NTP in reducing RBMS1 levels in isolated cardiomyocyte was confirmed by western blot and qRT-PCR (Appendix Fig. S11B,C). Notably, NTP treatment significantly enhanced the survival rate of TAC mice (Appendix Fig. S11D). Echocardiography analysis showed that NTP treatment markedly improved cardiac function induced by TAC surgery (Fig. 9A–C; Appendix Fig. S11E–I), with no significant change observed in LVPW;d (Appendix Fig. S11J). Meanwhile, NTP treatment substantially reversed pathological hypertrophy (Fig. 9D,E; Appendix Fig. S11K), cardiac remodeling and inflammatory infiltration (Fig. 9F; Appendix Fig. S11L), cardiac fibrosis (Fig. 9G,H; Appendix Fig. S11M), the cross-sectional area of cardiomyocyte (Fig. 9I,J), and hypertrophic markers induced by TAC (Fig. 9K–M). Additionally, the TAC-induced upregulation of CTTN-Δe11 was

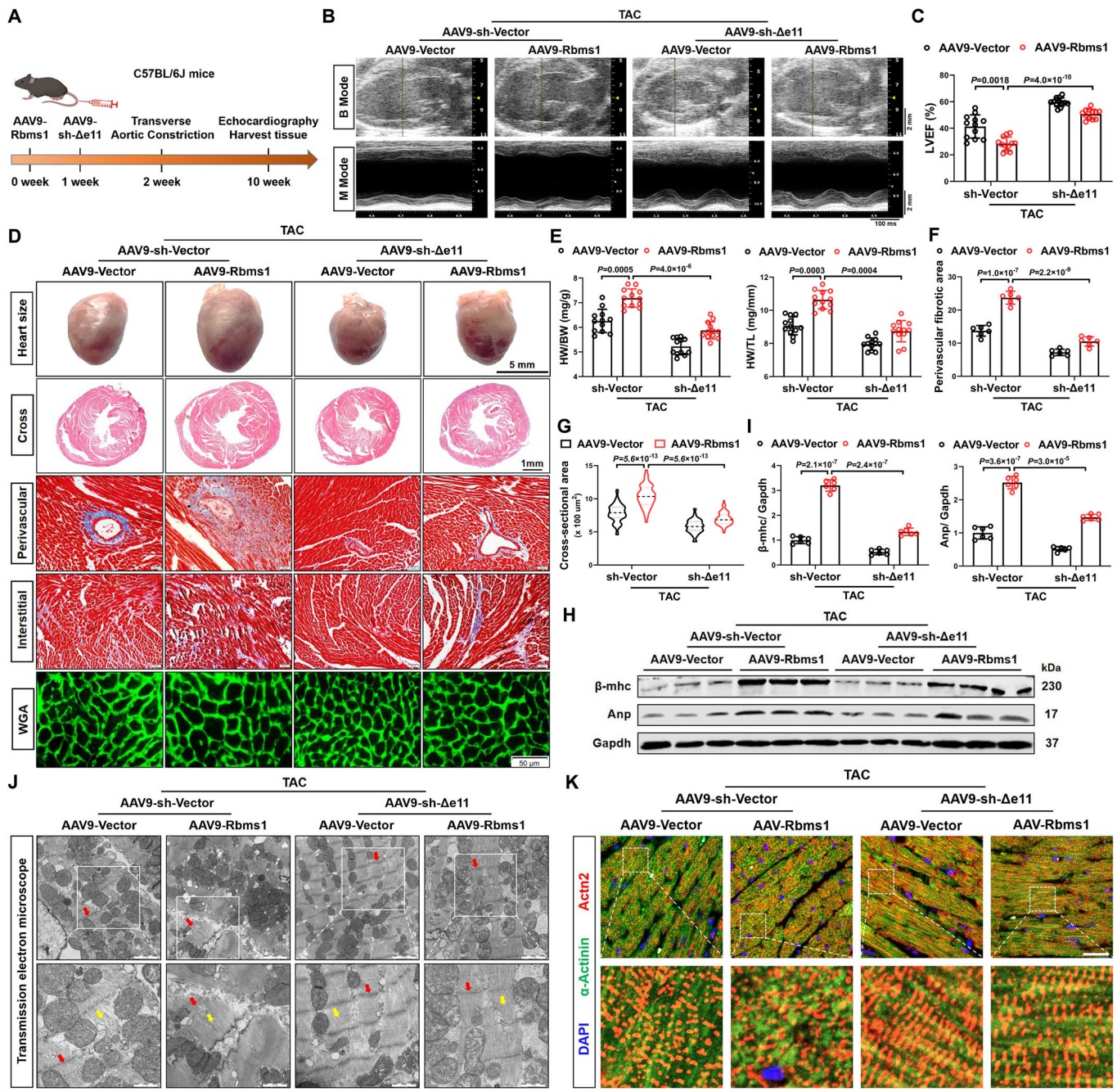

**Figure 7. RBMS1 exerts cardiac hypertrophic effects via regulation of CTTN splicing.**

(A) Experimental protocol of mice with AAV9 injection and TAC surgery. C57BL/6J mice were administered AAV9-RBMS1 (5 × 10¹¹ vg/mL) and AAV9-sh-CTTN-Δe11 (1 × 10¹¹ vg/mL) via tail vein injection. (B) Representative echocardiography of different group mice. (C) Quantification of LVEF ($n = 12$). (D) Heart size and histological sections of heart samples were stained with Masson and WGA staining. For heart size, scale bar = 5 mm; for cross-section of the heart, scale bar = 1 mm; for Masson and WGA staining, scale bar = 50 μm. (E–G) HW/BW, HW/TL ($n = 12$), perivascular fibrotic area ($n = 6$), and cross-sectional area of cardiomyocytes ($n = 70$) were examined in different groups. (H, I) Western blotting and quantification showing protein levels of β-MHC and ANP ($n = 6$). (J) Transmission electron microscope showed the disorganization of sarcomeres ($n = 4$), scale bar = 1 and 2 μm. The red and yellow arrows respectively represent the Z-disc and M-band. (K) Immunofluorescence showed the disorganization of sarcomeres ($n = 6$), scale bar = 20 μm. Data information: data are shown as mean ± SEM, $P$ values were analyzed with one-way ANOVA test (C, E, F, G, I). A dot represents an independent biological sample. Source data are available online for this figure.

significantly decreased after treatment with NTP (Appendix Fig. S11N). Moreover, the activaction of PI3K/AKT pathway was significantly decreased after treatment with NTP in response to TAC surgery (Appendix Fig. S11O). Together, these results demonstrate that treatment with NTP is capable of alleviating cardiac hypertrophy and heart remodeling, and pharmacologically silencing RBMS1 provides proactive treatment strategies for the clinical treatment of cardiac hypertrophy.

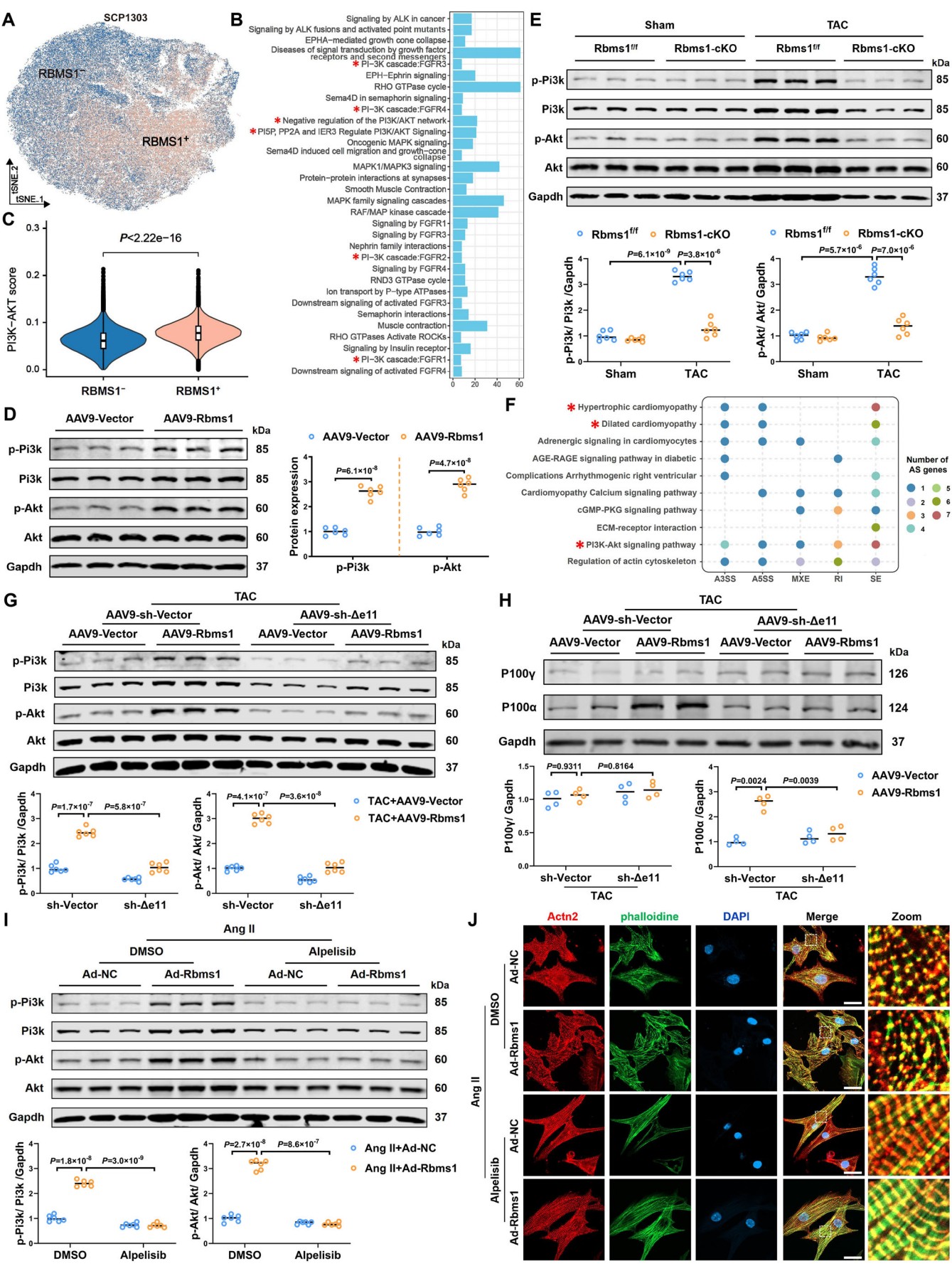

**Figure 8.  RBMS1 actives PI3K/AKT pathway via splicing CTTN to generate CTTN-Δe11 isoform.**

(A) Division of cardiomyocytes with high and low expression of RBMS1 in snRNA-seq dataset (SCP1303). (B) Functional enrichment analysis of cardiomyocytes with high expression of RBMS1. (C) The score of PI3K/AKT in cardiomyocytes with high expression of RBMS1. RBMS1⁻ (minimum: −0.037, Q1: 0.044, median: 0.061, Q3: 0.077, maximum: 0.214), RBMS1⁺ (minimum: −0.02, Q1: 0.063, median: 0.078, Q3: 0.093, maximum: 0.211). (D) Western blotting and quantification showing protein levels of p-PI3K and p-AKT in AAV9-RBMS1 mice ($n = 6$). (E) Western blotting and quantification showing protein levels of p-PI3K and p-AKT in RBMS1-cKO mice with TAC surgery ($n = 6$). (F) Results of KEGG enrichment analysis based on differentially expressed differential splicing genes. (G) Western blotting and quantification showing protein levels of p-PI3K and p-AKT in CTTN-Δe11 knockdown mice with TAC surgery ($n = 6$). (H) Western blotting and quantification showing protein levels of P100α and P100γ in RBMS1 overexpression NMCMs treatment with Alpelisib in response to Ang II stimulation ($n = 4$). (I) Western blotting and quantification showing protein levels of p-PI3K and p-AKT in Ad-RBMS1 NMCMs treatment with Alpelisib ($n = 6$). (J) Immunofluorescence staining of ACTN2 and phalloidine showed the disorganization of sarcomere and cytoskeleton, scale bar = 20 μm. Data information: data are shown as mean ± SEM, $P$ values were analyzed with unpaired Student's t test (C, D) and one-way ANOVA test (E, G, H, I). 2.2e-16 is the threshold commonly used in R language output, indicating a highly statistically significant result (C). A dot represents an independent biological sample. Source data are available online for this figure.

## Discussion

In this study, we demonstrated that RBMS1 is critical for pathological cardiac hypertrophy, for the following reasons: (1) Overexpression of RBMS1 aggravated cardiac hypertrophy via disorganizing the sarcomere and cytoskeleton of cardiomyocyte in response to hypertrophy surgery; (2) The hazardous function of RBMS1 on pathological cardiac hypertrophy relied on alternative splicing of CTTN to active PI3K/AKT pathway; (3) Nortriptyline significantly improved cardiac function and alleviated pathological cardiac hypertrophy. In conclusion, these findings provide the first insight that RBMS1 promotes cardiac hypertrophy by generating CTTN-Δe11 isoform and activating PI3K/AKT signaling pathway to disorganize the sarcomere and cytoskeleton of cardiomyocyte. The Synopsis was generated on the https://www.home-for-researchers.com/#/.

Recent studies have identified RBMS1 as a tumor promoter that reduces susceptibility to breast, lung, gastric, colorectal, and prostate cancers in mice. It has been shown that loss of RBMS1 enhances anti-tumor immunity in breast cancer by blocking PD-L1 checkpoints (Zhang et al, 2022a). In addition, elevated RBMS1 was observed in lung cancer patients, and RBMS1 deficiency inhibited the translation of SLC7A11 and promoted ferroptosis through the translation initiation factor eIF3d (Zhang et al, 2021). Overexpression of RBMS1 in HGC-27 and SGC-7901 cells increased the migration and invasion of gastric cancer cells by binding to the transcription factor MYC and activating the IL-6/JAK2/STAT3 signaling pathway (Liu et al, 2022). Meanwhile, inhibition of RBMS1 induced ferroptosis and restored the sensitivity of tumor cells to anticancer drugs (Xu et al, 2023). Other studies demonstrated that RBMS1 deficiency decreased the expression of Pax6, Tbr2, and Satb2, and reduced the differentiation of neuronal progenitor cells (Zhang et al, 2023). These studies established RBMS1 as an oncogenic factor in various tumors that promotes cell proliferation and inhibits ferroptosis; however, its specific mechanism in cardiovascular disease remains unclear. Here, our results showed that cardiomyocyte-specific overexpression of RBMS1 via AAV9 injection aggravated cardiac hypertrophy induced by TAC. Furthermore, RBMS1 deficiency attenuated pathological cardiac hypertrophy both in vivo and in vitro. It is important to note that in this study, RBMS1ᶠˡᵒˣ/ᶠˡᵒˣ mice were used as the control instead of aMHC-Cre mice when comparing with RBMS1-deficient mice. This choice was based on the consideration that aMHC-Cre mice may develop cardiotoxic effects after 6 months, whereas no cardiac dysfunction is expected to occur within the 4-month duration of

the present experiment. RBMS1 is highly conserved between humans and mice, and it played a similar role in hiPSC-CMs as in NMCMs. These findings suggest that RBMS1 may be a novel therapeutic target for the clinical treatment of cardiac hypertrophy.

Recent studies have shown that the role of alternative splicing (AS) regulated by RNA-binding proteins (RBPs) in cardiovascular disease has gained increasing attention. RBM24 regulates the assembly and integrity of the sarcomere by splicing exon 6 of ACTN2 and facilitates cardiac myofibrillogenesis (Lu et al, 2022). Additionally, cardiac-specific knockout of RBM24 in mice led to dilated cardiomyopathy due to misregulation of AS in multiple contractile genes (Liu et al, 2019). As a splicing factor, RBM20 regulates the splicing of heart-related genes involved in ventricular diastolic function and sarcomere assembly, including TTN, Ryr2, and CAMKII (Guo et al, 2012; Kayvanpour et al, 2017; Ma et al, 2016), thereby contributing to various cardiovascular diseases such as arrhythmia, dilated cardiomyopathy (DCM), myocardial insufficiency, and extensive myocardial fibrosis (Lennermann et al, 2020). In addition, the expression of RBM25 was increased in heart samples from patients with heart failure, and it co-mediated the AS of SCN5A with LUC7L3, thus participating in heart failure and arrhythmia (Gao et al, 2011). However, as a member of the RNA-binding protein family, RBMS1 has not been as thoroughly elucidated as other RBPs in post-transcriptional regulation. Here, we comprehensively screened and identified AS events regulated by RBMS1 through high-throughput RNA sequencing. Most of these events belonged to exon skipping and were extensively involved in cardiac hypertrophy-related functions, such as the I band, Z disc, sarcomere, and cytoskeleton. Consistently, transmission electron microscopy and immunofluorescence demonstrated that RBMS1 exacerbated cardiac hypertrophy by disrupting the sarcomere and cytoskeleton of cardiomyocytes. Taken together, these results identify RBMS1 as a splicing factor that modulates the sarcomere and cytoskeleton of cardiomyocytes through AS.

To further investigate the relationship between RBMS1 and the cardiomyocyte cytoskeleton, we analyzed splice variants generated by RBMS1 overexpression using high-throughput RNA sequencing. By combining volcano plot and GO analysis, we identified differentially expressed genes enriched in hypertrophic and dilated cardiomyopathy, including TTN, TPM3, CTTN, LAMA4, and ITGB1. Titin (TTN), one of the largest known proteins, is predominantly found in cardiac and striated muscle. It spans half of the sarcomere, from the Z line to the M line, and serves as a scaffold and elastic element within the muscle, providing structural support, elasticity, and signal transduction capabilities for myofibrils (Herman et al, 2012). Previous research

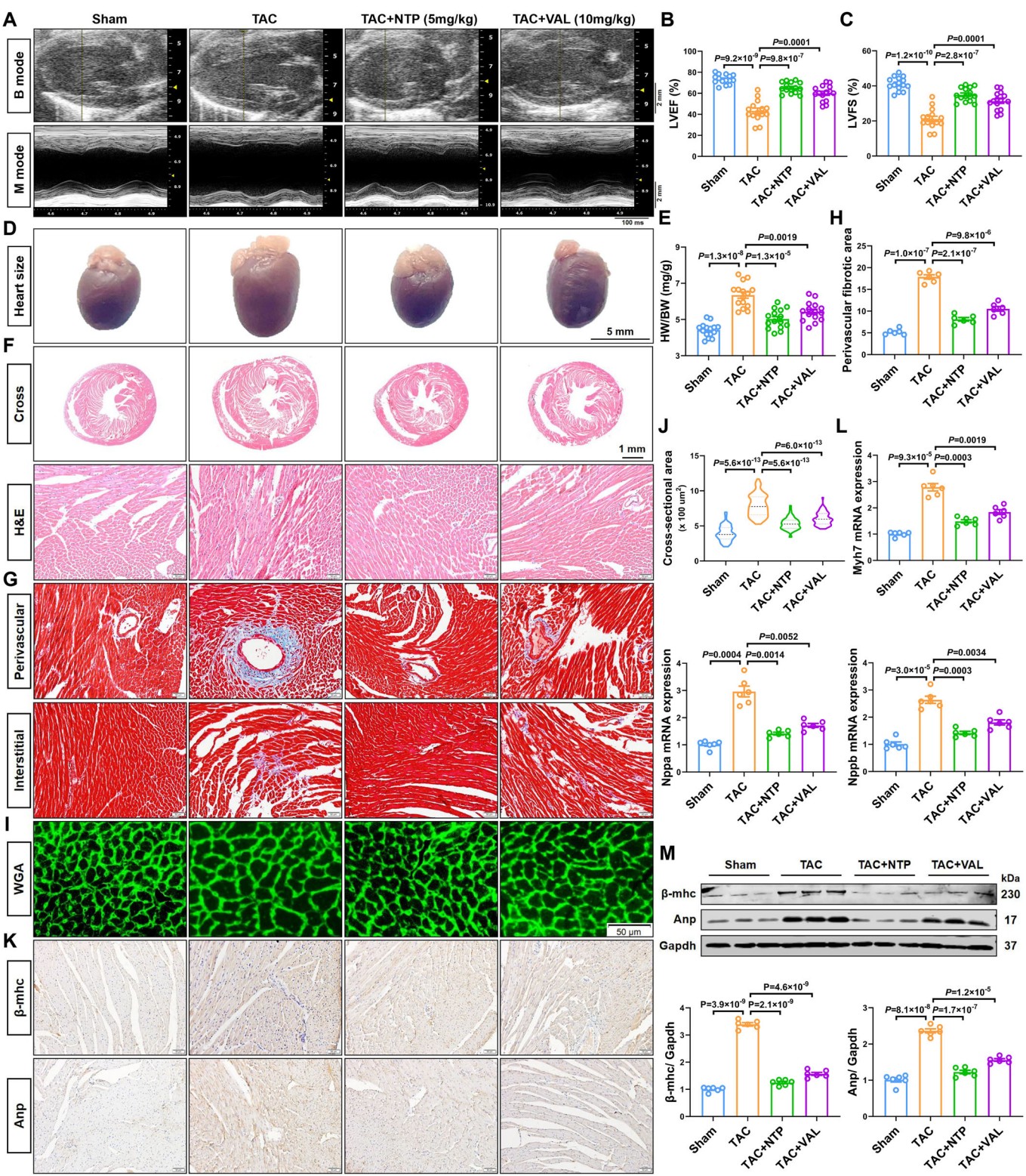

has shown that mutations in the TTN gene can compromise the structural integrity and mechanical properties of cardiac muscle fibers and are a common cause of heart diseases, including DCM, HCM, and HF (van den Hoogenhof et al, 2018; Arimura et al, 2009; Hein et al, 1994). Nevertheless, among the five genes, CTTN showed the most significant changes. CTTN, an actin-binding protein, regulates the actin cytoskeleton and the secretion of ECM proteins (Stradal and Costa, 2017). Previous studies have reported its association with intracellular motility, adherence, and invasion of microbial pathogens (Sharafutdinov et al, 2022). However, the role of CTTN in cardiac

**Figure 9. Pharmacological administration of nortriptyline suppresses cardiac hypertrophy.**

(A) Representative echocardiography of mice with NTP or VAL treatment. (B, C) Quantification of LVEF and LVFS ($n = 15$). (D) Heart size of TAC mice treatment with NTP or VAL surgery, scale bar = 5 mm. (E) Quantification of HW/BW ($n = 15$). (F) H&E staining of heart samples, scale bar = 50 μm. (G) Masson staining of heart sections, scale bar = 50 μm. (H) Quantification of the perivascular fibrotic area ($n = 6$). (I) WGA staining of heart sections, scale bar = 50 μm. (J) Quantification of cross-sectional area ($n = 70$). (K) IHC staining of heart sections, scale bar = 50 μm. (L) Quantification of mRNA levels of MYH7, NPPA, and NPPB ($n = 6$). (M) Western blotting and quantification showing protein levels of β-MHC and ANP ($n = 6$). Data information: data are shown as mean ± SEM, P values were analyzed with one-way ANOVA test (B, C, E, H, J, L, M). A dot represents an independent biological sample. Source data are available online for this figure.

hypertrophy remains unclear. Our data suggested that RBMS1 binds to intron 10 of CTTN and promotes its splicing to generate the CTTN-Δe11 isoform. Translation of exon 11 of CTTN produces amino acids 264-300, which constitute the Cortactin 6 domain. This domain stabilizes actin branches by linking activated Arp2/3 to nucleated actin filaments and plays a crucial role in biological processes such as cytoskeletal reorganization, cell migration, and adhesion (Protein Data Bank in Europe Knowledge Base). We identified CTTN as a downstream target of RBMS1 and hypothesized that RBMS1 regulates the organization of the sarcomere and cytoskeleton in cardiomyocytes by splicing CTTN to produce the CTTN-Δe11 isoform in cardiac hypertrophy. We found that cardiomyocyte-specific deficiency of CTTN-Δe11 attenuated pathological cardiac hypertrophy by increasing sarcomere length and promoting aligned orientation of sarcomeres both in vivo and in vitro. In addition, we determined that the detrimental effects of RBMS1 overexpression in hypertrophy were eliminated by CTTN-Δe11 deficiency, demonstrating that the function of RBMS1 in cardiac hypertrophy depends on CTTN-Δe11.

To further elucidate the pro-hypertrophic effect of RBMS1, we thoroughly investigated the signaling pathway regulated by RBMS1. snRNA sequencing dataset (SCP1303) analysis indicated that the PI3K/AKT pathway was enriched in cardiomyocytes with high expression of RBMS1. PI3K is a phosphatidylinositol kinase that transmits intracellular signals and is involved in the regulation of a variety of biological processes, including cell proliferation, migration, metabolism, and survival (Wang et al, 2022). It has been reported that PI3K is involved in a variety of cardiovascular diseases, including hypertension, heart failure, ischemic cardiomyopathy, and arrhythmias (Loo et al, 2024; Wohlschlaeger et al, 2007; Ye et al, 2018; Kim and Fishman, 2013). We found that the PI3K/AKT pathway was activated by RBMS1 overexpression, and the upregulation of p-PI3K and p-AKT induced by TAC surgery was significantly decreased in RBMS1-cKO mice. Previous studies have shown that CTTN promotes esophageal squamous cell carcinoma by activating PI3K/AKT (Du et al, 2009). Additionally, high-throughput RNA sequencing analysis showed that splicing genes, such as CTTN, were significantly enriched in the PI3K/AKT pathway. A considerable number of studies have demonstrated that the PI3K/AKT pathway regulates myocardial hypertrophy by influencing cardiomyocyte autophagy, mitochondrial function, oxidative stress, etc (Ghafouri-Fard et al, 2022). Nevertheless, the PI3K/AKT pathway appears to play an enigmatic role in regulating the progression of cardiac hypertrophy. Our results imply that the upregulation of the PI3K/AKT pathway by RBMS1 and CTTN-Δe11 is a pathogenic factor in cardiac hypertrophy. Some reports and data depict the PI3K/AKT pathway as conferring cardiac protection, while others suggest it causes cardiac injury. If the injury factors persist, long-term activation of this pathway will lead to cardiomyopathic hypertrophy and myocardial fibrosis, resulting

in the deterioration of cardiac function (Takimoto et al, 2005). In our study, the activation of the PI3K/AKT pathway by RBMS1 overexpression was eliminated by Δe11 deficiency.

Current research has preliminarily demonstrated that P100α plays a predominant role in physiological hypertrophy rather than in pathological hypertrophy (McMullen et al, 2003), whereas P100γ is primarily involved in the induction of pathological hypertrophy (Oudit et al, 2003). During physiological hypertrophy, early activation of p110α enhances myocardial contractility and helps maintain cardiac output in the short term. Evidence suggests that P100γ, rather than P100α, is primarily responsible for early compensatory pathological hypertrophy (Naga Prasad et al, 2000). However, the regulatory role of P100α in advanced pathological hypertrophy remains ambiguous. Furthermore, sustained activation of p110α leads to excessive reactive oxygen species (ROS) production, mitochondrial dysfunction, disruption of calcium homeostasis, and myocardial fibrosis, thereby facilitating the progression of decompensated pathological hypertrophy (Bass-Stringer et al, 2025). We demonstrated that overexpression of RBMS1 significantly promoted the expression of P100α, while knockdown of Δe11 suppressed P100α production. Moreover, the P100α inhibitor-Alpelisib significantly decreased the activation of PI3K/AKT induced by RBMS1 or Δe11 overexpression. Taken together, these findings indicate that the RBMS1/Δe11 axis contributes to pathological cardiac hypertrophy by facilitating the conversion of PIP2 to PIP3 through P100α, consequently leading to the activation of the PI3K-AKT signaling pathway.

To further investigate the function of RBMS1 in cardiac hypertrophy, we explored the therapeutic effect of targeting RBMS1 on cardiac hypertrophy. Previous studies have shown that nortriptyline participated in the progression of lung cancer by inhibiting RBMS1 expression (Zhang et al, 2021). Thus, we further explored the role of nortriptyline in cardiac hypertrophy in the heart. Nortriptyline is widely used clinically to treat a variety of mental disorders such as depression and anxiety disorders (Gayral and Roux, 1968). Nortriptyline is a relatively reliable medication in the treatment of depression and anxiety, but there are still some controversies regarding its cardiovascular effects, such as heart rate, blood pressure, and other cardiovascular side effects (Roose et al, 1986; Bonaccorsi and Garattini, 1978). In general, nortriptyline has relatively few cardiac effects and can be used safely in most patients. In our studies, we demonstrated that the decreased cardiac function and cardiac hypertrophy induced by TAC were eliminated upon treatment with a low dose of nortriptyline (5 mg/kg). Valsartan is a commonly used antihypertensive drug, belonging to the angiotensin II receptor antagonist class. It is widely used in the clinical treatment of cardiac hypertrophy due to its effects of reducing cardiac burden, lowering blood pressure, and protecting the myocardium (Ho et al, 2021). Therefore, we used Valsartan as a

positive control for nortriptyline. In our research, it was discovered that nortriptyline exhibited greater efficacy than Valsartan in the treatment of cardiac hypertrophy. This can be attributed to the fact that nortriptyline influences the reuptake of neurotransmitters such as norepinephrine and serotonin; affects various proteins or pathways, including Kv channels, beta-adrenergic receptors, and other receptors (Micó et al, 1997); alleviates chronic pain associated with cardiovascular diseases and enhances patients' quality of life (Heymann et al, 2001); and improves depression, which is a complication of cardiovascular diseases.

Although our investigation identified CTTN as a key functional mediator through which RBMS1 contributes to cardiac hypertrophy, we recognize that the study has certain limitations. Our findings confirm that RBMS1 is an RNA-binding protein capable of interacting with CTTN and modulating the PI3K-AKT pathway and myocardial cytoskeletal dysfunction. However, in addition to CTTN, there are likely other unidentified target genes of RBMS1 that may play roles in cardiac hypertrophy via distinct biological processes or pathways. While this study validated CTTN as a downstream target of RBMS1, the potential contributions of other RBMS1-regulated targets warrant further investigation. The focus of this study is to elucidate the mechanism by which RBMS1 participates in myocardial hypertrophy through CTTN splicing variant in cardiomyocytes. Interestingly, this phenomenon differs from the alternative splicing events we previously observed in fibroblasts. In our earlier research, we demonstrated that RBMS1 in fibroblasts promotes cardiac fibrosis and heart failure by regulating LM07 alternative splicing, which subsequently activates the AP-1/TGF-β pathway (Li et al, 2025). However, in cardiomyocytes, we identified splicing events that are distinctly different from those in fibroblasts. This indicates that the alternative splicing regulated by RBMS1 is cell-type-specific, potentially due to variations in transcriptional profiles or splicing regulatory mechanisms across different cell types. As an RBP, RBMS1 typically interacts with target RNA in a sequence-specific manner through its RNA recognition motif (RRM). The regulation of alternative splicing by RBPs is a highly complex process, involving spliceosome assembly, splice site selection, and the recruitment or inhibition of core spliceosomal components or other regulatory proteins. Further investigation is required to delineate the precise molecular mechanisms through which RBMS1 governs the splicing of critical target genes.

# Methods

### Reagents and tools table

| Reagent/Resource | Reference or source | Identifier or Catalog Number |
|---|---|---|
| C57BL/6 J | Changsheng Biotechnology | |
| RBMS1flox/flox | Cyagen Biosciences | |
| MYH6-Cre | Cyagen Biosciences | C001831 |

| Reagent/Resource | Reference or source | Identifier or Catalog Number |
|---|---|---|
| hiPSC | Cellapy® | CA4025106 |
| **Recombinant DNA** | | |
| siRNA-Mouse RBMS1-1 | Genecreate | GGAAGATCGTGTCCACAAA |
| siRNA-Mouse RBMS1-2 | Genecreate | CCATACAGTATTGCCACAA |
| siRNA-Mouse RBMS1-3 | Genecreate | CCGTGTTAACTCCCTCAAT |
| siRNA-Mouse CTTN-1 | Genecreate | CAAAAGCATCAGAATCCAT |
| siRNA-Mouse CTTN-2 | Genecreate | GAACAAGGAAGCTCGGATT |
| siRNA-Mouse CTTN-3 | Genecreate | GCGATCACAATTTTTTGAA |
| siRNA-Mouse CTTN-Δe11-1 | Genecreate | CCCTTTGATTCAAAAGGAA |
| siRNA-Mouse CTTN-Δe11-2 | Genecreate | GATTCAAAAGGAAGCTCGG |
| siRNA-Mouse CTTN-Δe11-3 | Genecreate | TCAAAAGGAAGCTCGGATT |
| siRNA-Human RBMS1 | Genecreate | CCAUAUACCUUUCAACCUAAU |
| Minigene | Cyagen Biosciences | |
| Ad-RBMS1 | Han Heng Biology | |
| Ad-CTTN-Δe11 | Han Heng Biology | |
| AAV9-RBMS1 | Shanghai Genechem Co., LTD | |
| AAV9-sh-CTTN-Δe11 | Genecreate | TCCCAAAAAGACTATGCCA |
| **Antibodies (dilution)** | | |
| β-MHC (WB 1:500) | Proteintech | 22280-1-AP |
| RBMS1 (WB 1:500, IF 1:300) | Proteintech | 11061-2-AP |
| ANP (WB 1:500) | Proteintech | 27426-1-AP |
| p-PI3K (WB 1:400) | Cell Signaling Technology | 4228S |
| PI3K (WB 1:500) | Cell Signaling Technology | 4257S |
| p-AKT (WB 1:400) | Cell Signaling Technology | 4060S |
| AKT (WB 1:500) | Cell Signaling Technology | 4691S |
| Lamin B (WB 1:1000) | Proteintech | 12987-1-AP |
| GAPDH (WB 1:2000) | Proteintech | 15613-1-AP |
| α-Actinin (IF 1:500) | GeneTex | GTX84959 |

| Reagent/Resource | Reference or source | Identifier or Catalog Number |
|---|---|---|
| ACTN2 (IF 1:500) | Proteintech | 14221-1-AP |
| P100α (WB 1:500) | Proteintech | 67071-1-Ig |
| P100γ (WB 1:500) | Proteintech | 20662-1-AP |
| **Oligonucleotides and other sequence-based reagents** | | |
| Human RBMS1-Forward (F) | Integrated DNA Technologies | ATGGGCAAAGTGTGGAAACAG |
| Human RBMS1-Reverse (R) | Integrated DNA Technologies | CTTGGCTTGCAGATACTGGGG |
| Human MYH7-F | Integrated DNA Technologies | TCACCAACAACCCCTACGATT |
| Human MYH7-R | Integrated DNA Technologies | CTCCTCAGCGTCATCAATGGA |
| Human NPPA-F | Integrated DNA Technologies | CAACGCAGACCTGATGGATTT |
| Human NPPA-R | Integrated DNA Technologies | AGCCCCCGCTTCTTCATTC |
| Human NPPB-F | Integrated DNA Technologies | TGGAAACGTCCGGGTTACAG |
| Human NPPB-R | Integrated DNA Technologies | CTGATCCGGTCCATCTTCCT |
| Human CTTN-Δe11-F | Integrated DNA Technologies | TGGAGGAAAATTTGGTGTGCAG |
| Human CTTN-Δe11-R | Integrated DNA Technologies | CCGCCGAATCCTTTGGAGT |
| Human GAPDH-F | Integrated DNA Technologies | GGAGCGAGATCCCTCCAAAAT |
| Human GAPDH-R | Integrated DNA Technologies | GGCTGTTGTCATACTTCTCATGG |
| Mouse RBMS1-F | Integrated DNA Technologies | CTGAGCAAGACAAACCTCTACAT |
| Mouse RBMS1-R | Integrated DNA Technologies | GGCCTTATCCAAAATCGCCTT |
| Mouse MYH7-F | Integrated DNA Technologies | TTACTTGCTACCCTCAGGTGG |
| Mouse MYH7-R | Integrated DNA Technologies | CTCCTTCTCAGACTTCCGCA |
| Mouse NPPA-F | Integrated DNA Technologies | TCGGAGCCTACGAAGATCCA |
| Mouse NPPA-R | Integrated DNA Technologies | GTGGCAATGTGACCAAGCTG |
| Mouse NPPB-F | Integrated DNA Technologies | CTGGCATACTCTTGCAGCCT |
| Mouse NPPB-R | Integrated DNA Technologies | CTGCCTTGTGAAGGGGTGAT |
| Mouse CTTN-F | Integrated DNA Technologies | TAAGACTGGTTTCGGAGGCA |
| Mouse CTTN-R | Integrated DNA Technologies | GCTTGGCCAATCTCTCCTTGT |
| Mouse CTTN-Δe11-F | Integrated DNA Technologies | GCTGGGACCATCAGGAGAAGC |
| Mouse CTTN-Δe11-R | Integrated DNA Technologies | GATCCTTCTGCACCCCATACTT |

| Reagent/Resource | Reference or source | Identifier or Catalog Number |
|---|---|---|
| Mouse TTN-Δe12-F | Integrated DNA Technologies | GCTGCATCCATGGTAGTGGT |
| Mouse TTN-Δe12-R | Integrated DNA Technologies | GGCCATGGCTTCTCTAGACC |
| Mouse TPM3-Δe6-F | Integrated DNA Technologies | GGAACTCCAGGAAATCCAGC |
| Mouse TPM3-Δe6-R | Integrated DNA Technologies | CTCCGTGCGTTCCAAGTCT |
| Mouse LAMA2-Δe53-F | Integrated DNA Technologies | TGTCGACATCGATTCTAACCAGG |
| Mouse LAMA2-Δe53-R | Integrated DNA Technologies | GAACAGCCTTTGGTCACACC |
| Mouse ITGB1-Δe15-F | Integrated DNA Technologies | ATCCCAATTGTAGCAGGCGT |
| Mouse ITGB1-Δe15-R | Integrated DNA Technologies | CTGCTGTGAGCTTGGTGTTG |
| Mouse GAPDH-F | Integrated DNA Technologies | TGGCCTTCCGTGTTCCTAC |
| Mouse GAPDH-R | Integrated DNA Technologies | GAGTTGCTGTTGAAGTCGCA |
| **Chemicals, Enzymes and other reagents** | | |
| Lipofectamine 2000 Reagent | Thermofisher | 11668019 |
| Opti-MEM Reagent | Thermofisher | 31985062 |
| BCA Reagent | Beyotime | P0009 |
| BSA Reagent | Solarbio | A8010 |
| TRIzol Reagent | Thermofisher | 16096020 |
| H&E staining kit | Solarbio | G1120 |
| Masson staining kit | Solarbio | G1340 |
| ReverTra Ace qPCR RT kit | TOYOBO | FSQ-201 |
| Wheat Germ Agglutinin | Thermofisher | W11263 |
| Paraformaldehyde Reagent | Beyotime | P0099-3L |
| Type II collagenase | Biosharp | 9001-12-1 |
| Fetal bovine serum | VivaCell | C2910-0500 |
| Calf serum | Merck | B7446 |
| DMEM Medium | VivaCell | C3113-0500 |
| Three antibiotics | Solarbio | P1410 |
| Ang II | Medchemexpress | HY-13948 |
| Nortriptyline | Medchemexpress | HY-118620 |
| **Software** | | |
| GraphPad Prism 8 | GraphPad | Prism 8.0.2 |
| ImageJ software | NIH Image | ImageJ 1.54k |
| Matlab | MathWorks | MATLAB R2024b |

Detailed Materials and Methods are provided in Supplemental Materials.

## Human studies

Human left ventricle samples from 5 patients with DCM and 5 healthy subjects were collected at Zhongnan Hospital of Wuhan University. Healthy donor heart tissues were obtained from hearts with no history of cardiac disease at Zhongnan Hospital of Wuhan University, which were originally intended for heart transplantation. These organs were deemed unsuitable for transplantation due to donor-recipient incompatibility. Inclusion criteria for patients with DCM: Patients meeting the clinical indicators according to the American Heart Association (AHA) and American College of Cardiology (ACC) diagnostic guidelines; patients provide informed consent; aged 18 years or older. Exclusion criteria for DCM patients: History of other heart diseases; conditions significantly affecting heart function; recent heart surgery; pregnancy; use of medications significantly affecting heart function. Informed consent was obtained from prospective donors before collecting heart tissue. The study was approved by the Ethics Committee of Zhongnan Hospital of Wuhan University (Approval No. 2022075K) in full accordance with the ethical principles outlined in the World Medical Association Declaration of Helsinki and Department of Health and Human Services Belmont Report. Specific patient characteristics are provided in Appendix Tables S1 and S2.

## Animals studies

MYH6-Cre mice and RBMS1$^{flox/flox}$ transgenic mice were purchased from Cyagen Biosciences (Suzhou, China). Based on the genomic structure of RBMS1 and the conservation of its functional protein domains, two single-guide RNAs (sgRNAs) with high on-target scores and low off-target effects were designed using the online tool CRISPOR (http://crispor.tefor.net/). These sgRNAs were selected to target the 5' and 3' ends of exon 3 of RBMS1, respectively, with their protospacer adjacent motif sequences located outside the intended loxP insertion sites to facilitate dual DNA cleavage and promote efficient homology-directed repair. The sgRNAs were prepared via in vitro transcription, which involved template preparation, transcription, purification, and quality control. A donor plasmid was constructed containing homologous arms, two loxP sites, and exon 3 of RBMS1. The assembly of these elements into a plasmid backbone was achieved using Golden Gate cloning. The donor plasmid was linearized by restriction enzyme digestion to enhance recombination efficiency. Purified sgRNAs were mixed with Cas9 protein at a molar ratio of 2:1 and incubated at room temperature for 10–15 min to form ribonucleoprotein complexes. The linearized donor plasmid was then added to the RNP mixture to prepare the final injection solution. Female C57BL/6J mice (3–4 weeks old) were superovulated by sequential injection of pregnant mare serum gonadotropin followed 48 h later by human chorionic gonadotropin, and then mated with fertile adult male mice. The following day, the females were euthanized, and zygotes were collected from the oviducts. The zygotes were maintained in KSOM medium droplets in a 37 °C incubator with 5% $CO_2$ before manipulation. The prepared injection mixture was delivered into the pronuclei of zygotes via pronuclear microinjection. After injection, the embryos were transferred to M16 medium and cultured in the incubator for 0.5–1 h. Subsequently, the embryos were transplanted into the oviducts of pseudopregnant female mice. The resulting F0 pups were genotyped using RBMS1-specific primers, followed by breeding and establishment of the mouse line. The RBMS1$^{flox/flox}$ transgenic mice and the MYH6-Cre tool mice were intercrossed to generate the αMHC-RBMS1$^{flox/flox}$ transgenic mice.

C57BL/6 J male mice were purchased from Changsheng Biotechnology (Liaoning, China). C57BL/6J male mice aged 8-10 weeks were anesthetized with 2.5% isoflurane, then placed in a supine position on a thermostatic pad. After hair removal and disinfection, the neck skin was incised and the aortic arch was exposed. A 26-G constricting needle was placed above the aortic arch, and the vessel was ligated with a 6-0 surgical suture around the needle. After withdrawing the constriction needle, the skin was sutured and iodophor was applied to the surgical incision. The sham group underwent the same procedures as the TAC group except for the constriction. Based on the titer of the adeno-associated virus 9, the mice were administered AAV9-RBMS1 via tail vein injection at a concentration of $5 \times 10^{11}$ vg/mL, and AAV9-sh-CTTN-Δe11 via tail vein injection at a concentration of $1 \times 10^{11}$ vg/mL.

## Ethical reviews

At the end of the experiment, animals were anesthetized with 2.5% isoflurane. Euthanasia was performed by cervical dislocation and bilateral thoracotomy to ensure death. Prior to tissue collection, the absence of vital signs was confirmed, including cardiac arrest, unresponsiveness to toe-pinch reflex, and fixed, dilated pupils. All animal and human procedures were approved by the Institutional Review Committee of Harbin Medical University (IRB2020724) and followed the NIH Guide for the Care and Use of Laboratory Animals.

## Generation of human induced pluripotent stem cell-derived cardiomyocytes

Human induced pluripotent stem cells (hiPSCs) were derived from human blood mononuclear cells using non-integrating methods (Cellapy, Beijing) and cultured in the PSCeasy® human pluripotent stem cell culture system. hiPSCs were maintained in PGM1 human pluripotent stem cell medium until reaching 80% confluence. CardioEasy® human cardiomyocyte differentiation media I, II, and III were subsequently added for 48 h each to induce differentiation into human induced pluripotent stem cell-derived cardiomyocytes (hiPSC-CMs). Once abundant beating hiPSC-CMs appeared, the cells were purified using CardioEasy® human cardiomyocyte purification medium. When the cardiomyocytes stopped beating, the medium was replaced with CardioEasy® human cardiomyocyte maintenance medium. Transfection was performed after the cells stabilized and were passaged.

## Western blot

Total protein from cells and tissues was extracted using lysis buffer (Beyotime, Jiangsu, China). Cytoplasmic and nuclear proteins were extracted with a kit from Beyotime Biotechnology (P0028). A 10% SDS-PAGE gel was prepared with a PAGE gel rapid preparation kit

(Yaseng, Shanghai, China). Electrophoresis was carried out at 70 V, and proteins were transferred to nitrocellulose membranes (Millipore, GSWP04700) after 2 h. The membranes were incubated with specific antibodies overnight on a shaker at 4 °C, including β-MHC, RBMS1, ANP, p-PI3K, PI3K, p-AKT, AKT, Lamin B, P100α, P100γ, and GAPDH. Each antibody were diluted with the primary antibody diluent (Beyotime, P0023A) at the appropriate ratio, respectively. The next day, membranes were incubated with species-appropriate secondary antibodies for 1 h at room temperature and washed with PBST three times for 7 min each. Protein expression was visualized using the Odyssey infrared imaging system.

## Ultrasound imaging measurements

Mice were anesthetized by inhalation of 2.5% isoflurane. Subsequently, the chest hair was removed by shaving, and the skin was further cleaned with depilatory cream to minimize ultrasound artifacts. The animals were then placed in a supine position on a physiological monitoring platform. An ultrasound probe (MX550D, 26–52 MHz) was selected for imaging. A small amount of conductive gel was applied to the copper plate of the monitoring platform, and the mouse's paw was affixed to it to acquire electrocardiographic and respiratory signals. A rectal temperature probe was inserted to continuously monitor core body temperature. The ultrasound probe was positioned vertically, perpendicular to the left ventricle. The probe angle was then adjusted to clearly visualize the mitral valve, left ventricular cavity, ascending aorta, and aortic valve. B-mode images were recorded once the mitral valve reached maximal contraction. The system was then switched to M-mode, and the M-mode cursor was placed at the posterior edge of the papillary muscle to measure left ventricular end-diastolic volume (EDV), end-systolic volume (ESV), left ventricular posterior wall in systole (LVPWs), posterior wall in diastole (LVPWd), internal dimension in systole (LVIDs), and left ventricular internal dimension in diastole (LVIDd), thereby obtaining the parasternal long-axis view (PLAX) of the left ventricle. Left ventricular ejection fraction (EF%) = (EDV – ESV)/EDV × 100. Fractional shortening (FS%) = [(LVIDd – LVIDs)/LVIDd] × 100. Next, the probe was rotated 90° to obtain the parasternal short-axis view (PSAX) of the left ventricle. In this view, the M-mode cursor was placed centrally within the left ventricle, typically at the level of the papillary muscles. Measurements obtained in this view demonstrated good consistency with those obtained in the long-axis view. The apical four-chamber view was used to assess mitral inflow velocities (E wave, A wave, and E/A ratio) using pulsed-wave Doppler (PW). When necessary, color Doppler imaging was employed to confirm the location of mitral inflow. On the standard PSAX view, the probe was advanced toward the aortic root until the aortic valve became visible. The sample volume was then placed beneath the mitral valve, allowing for the acquisition of mitral inflow data from the apical four-chamber view. Throughout the entire ultrasound procedure, vital signs of the mice were closely monitored, including body temperature (37 ± 0.5 °C) and heart rate (350–450 bpm). Anesthesia depth was adjusted as needed to maintain physiological stability. Ultrasound parameters, including gain, depth, and Doppler angle (<60°), were optimized to ensure high-quality image acquisition.

## Transmission electron microscope

Left ventricular tissue from mice was collected and cut into 1–2 mm³ pieces, which were fixed overnight with 2.5% glutaraldehyde. After rinsing, dehydrating, permeabilizing, and embedding, 70 nm sections were prepared and stained with lead citrate and uranyl acetate solutions. The ultrastructure of the myocardium was imaged using a JEOL 1200 transmission electron microscope.

## Isolation of neonatal mouse cardiac cardiomyocytes

Neonatal mice were purchased from the Animal Laboratory Center of the Second Affiliated Hospital of Harbin Medical University. The neonatal mice were sterilized with 75% alcohol, then the chest was opened and the heart was removed using curved forceps and placed into pre-cooled DMEM. After washing 3–5 times with PBS, the hearts were treated with 3 ml PBS and 2 ml trypsin and placed on a 4 °C shaker for 8–12 h. The next day, 16.8 mg collagenase was accurately weighed and dissolved in 21 ml DMEM, then filtered through a microporous filter. The trypsin solution was discarded, and digestion was terminated by adding 5 ml of complete medium followed by 2–3 washes. Collagenase was added for digestion on a 37 °C shaker for 10 min, repeated 4 to 6 times until the heart tissue was completely digested. The suspension was centrifuged at 1000 rpm for 5 min. The supernatant was discarded, and the cells were resuspended in complete culture medium and incubated in culture flasks at 37 °C for 1.5–2 h. Cardiomyocytes were resuspended and seeded into pre-prepared culture plates and maintained in a 37 °C incubator. Cell status was observed 48 h later.

## Cell transfection

Under light-protected conditions, 4 µl siRNA was mixed with 96 µl Opti-MEM, and separately, 96 µl Lipofectamine 2000 was mixed with 96 µl Opti-MEM, each incubated for 5 min. The two mixtures were combined gently and left for 20 min. The cell culture medium was replaced with 1.8 ml DMEM, and the transfection complex was added. The culture plate was gently shaken to distribute the complex evenly. After 6 h, the medium was replaced with DMEM containing 10% fetal bovine serum. The medium was subsequently changed every 24 h to 2 ml DMEM containing 1 µmol/L Ang II. After 48 h of treatment, cells were collected for subsequent experiments.

## Quantitative real-time PCR (qRT-PCR)

After RNA extraction, RNA purity and concentration were determined using a NanoDrop 8000. Samples with appropriate concentration and A260/280 ratios between 1.8 and 2.0 were selected for subsequent experiments. Reverse transcription was performed according to the ReverTra Ace qPCR RT kit instructions. The reaction conditions were: 42 °C for 15 min; 85 °C for 5 min; and 4 °C hold. The cDNA obtained was used as a template to amplify RBMS1, NPPA, NPPB, MYH7, and GAPDH genes from mouse heart and cardiomyocyte samples. All reactions were prepared in a 10 µl volume. The PCR conditions were: 95 °C for 5 min; followed by 40 cycles of 95 °C for 10 s, 55 °C for 15 s, and 72 °C for 20 s. Data were analyzed using the $2^{(-\Delta\Delta CT)}$ method.

## RNA sequencing

Forty-eight hours after transfection with Ad-NC and Ad-RBMS1, cardiomyocytes were harvested and total RNA was extracted. Genomic DNA contamination was removed by DNase treatment, and mRNA was enriched using a poly(A) selection kit. The mRNA was reverse-transcribed into cDNA, and the cDNA was fragmented by sonication (200–300 bp). After end repair, A-tailing, and adapter ligation, the cDNA fragments were amplified and purified by PCR. Library quality and concentration were assessed using an Agilent Bioanalyzer. The library was loaded onto the Illumina NovaSeq sequencing platform for high-throughput sequencing. Raw sequencing data were converted to FASTQ format using bcl2fastq software. FastQC software was used to assess the quality of the FASTQ files. HISAT2 software was used to align reads to the reference genome, and HTSeq software was used to quantify gene or transcript expression.

## Acquisition of public datasets

The GSE135055 dataset was obtained from the NCBI database (http://www.ncbi.nlm.nih.gov/geo/). It includes nine samples of healthy normal left ventricular tissues derived from brain-dead donors with normal cardiac function, as well as 21 samples from patients with heart failure, among which 18 were diagnosed with dilated cardiomyopathy (DCM), and the remaining three were diagnosed with myocarditis. The SCP1303 dataset was obtained from the Broad Institute Single Cell Portal (https://singlecell.broadinstitute.org/single_cell), which contains samples from 15 patients diagnosed with hypertrophic cardiomyopathy (HCM) and 16 samples from organ donors without a history of heart failure.

## Single-cell data collection and processing

We retrieved heart failure single-cell data from the Single Cell Portal (https://singlecell.broadinstitute.org/single_cell). We conducted preprocessing, quality control, and cell clustering analysis employing the Seurat (5.1.0) package within R software v4.3.0 (https://www.r-project.org). For single-cell RNA-seq data (SCP1303), low-quality cells were removed based on the following criteria: (1) total RNA counts (nCount_RNA) > 500 and <90,000; (2) number of detected genes (nFeature_RNA) > 300 and <10,000; and (3) percentage of mitochondrial transcripts (percent.mt) <5%. Cells meeting all thresholds were retained for downstream analysis. After quality control, 517,444 cells were obtained. The expression data were normalized using the 'LogNormalize' and 'ScaleData' functions, and the top 3000 most variable features were identified through the 'FindVariableFeatures' function. The integrated dataset was subsequently normalized using Seurat's standard normalization and scaling functions. Finally, cells were clustered using the 'FindClusters' function at a resolution of 0.3 and visualized with 2D Uniform Manifold Approximation and Projection (UMAP). Marker genes for each cluster were identified by the 'FindAllMarkers' function, applying a log-fold change threshold of 0.25. Cell types within each cluster were determined by comparing the marker genes to annotations from the original studies and characteristic genes from the Cell Taxonomy database (https://ngdc.cncb.ac.cn/celltaxonomy/).

## Correlation between RBMS1 and marker of HF

Based on bulk data from heart failure tissues (GSE135055), we calculated the correlation between RBMS1 and heart failure markers (NPPA and NPPB) using pearson correlation test.

## Enrichment analysis

Overrepresentation of Gene Ontology (GO) terms, Reactome pathways, or Kyoto Encyclopedia of Genes and Genomes (KEGG) pathways in the gene lists was assessed using the 'enrichGO' function in the R package clusterProfiler (v4.10.1). Gene symbols were converted to Entrez IDs using the 'bitr' function. Unless otherwise specified, we present only the top significant pathways, sorted by increasing $P$-value, and limited to those with an adjusted $P < 0.05$.

## Screening of cardiomyocyte subsets

We performed Weighted Gene Co-Expression Network Analysis (WGCNA), a systems biology method used to construct gene co-expression networks and identify gene modules associated with specific traits. Using GSE135055 bulk data, we filtered the top 75% of genes based on the median absolute deviation (MAD), selecting those with a MAD greater than 0.01 for WGCNA (v1.73, R package) to construct a co-expression network, with a soft threshold of 16. Genes were grouped into modules by hierarchical clustering, and each module was assigned a distinct color (e.g., turquoise, blue, yellow, green, magenta, pink, red, black, brown, gray) for identification. The color intensity within each cell represents the strength and direction of the correlation between the module eigengene (first principal component of the module expression matrix) and the sample trait. These colors do not carry intrinsic biological meaning but serve as identifiers for different modules.

Next, we used 'AddModuleScore' function (Seurat R package) to evaluate the genes from the MEyellow module, identified through WGCNA, within the single-cell data (SCP1303). We classified the myocardial cells into RBMS1+ and RBMS1- groups according to the average module score. After excluding cells with no expression of CTTN, we compared CTTN expression based on RBMS1$^{-/+}$ grouping. Cardiomyocytes were categorized into two groups: those expressing CTTN (CTTN$^+$) and those not expressing CTTN (CTTN$^-$), based on the levels of CTTN expression.

## Differential expression analysis in bioinformatics

The DESeq2 (v1.42.1) R package was utilized to calculate the differential gene expression threshold between Ad-NC and Ad-RBMS1, with a cutoff of logFC > 0.5 and $P < 0.05$. For bulk RNA-seq data, DESeq2 (v1.42.1) was applied to count data, while the limma (v3.58.1) package was used for FPKM data, with thresholds set to $P < 0.01$ and logFC > 0.8. In single-cell data analysis, the 'FindMarkers' function from Seurat v5.1.0 was employed to identify differentially expressed RNA-binding proteins (RBPs) between the heart failure and normal groups, using criteria of $P$.adjust < 0.05 and avg_log2FC > 0.25.

## Immunofluorescence staining

Primary cardiomyocytes were cultured in 24-well plates and treated when cell density reached 60%. They were washed three times with PBS, fixed with 500 μl of 4% paraformaldehyde at 37 °C for 15 min, permeabilized with 500 μl permeabilization solution for 1 h at room temperature, and blocked with 1 ml of 50% goat serum (goat serum:PBS = 1:1) for 1 h. Primary antibody incubation was performed overnight at 4 °C. The next day, cells were incubated with secondary antibody for 1 h; nuclei were stained with DAPI for 7 min at room temperature. Cell fluorescence intensity and cell size were observed under a fluorescence microscope. For WGA staining, we quantified the total area of three adjacent cardiomyocytes with ImageJ; each data point represented the average size of three adjacent cardiomyocytes. For cell surface area, we quantified the total area of all cardiomyocytes within the visual field; each data point represented the total area of cardiomyocytes divided by the number of DAPI-positive nuclei in a given field. For sarcomere length, we quantified the average length of three adjacent sarcomeres with ImageJ; each data point represented the average length of three adjacent sarcomeres. The orientation analysis of sarcomere arrangement was performed using Matlab software. A well-developed cytoskeleton exhibits highly concentrated orientation, while cytoskeleton remodeling and disturbance can lead to random orientation of sarcomere arrangement.

## Agarose gel

PCR products were analyzed by agarose gel electrophoresis. A 1.5% agarose gel was prepared by weighing 0.9 g agarose powder, adding 60 ml of 0.5× TAE buffer, mixing well, heating in a microwave oven until boiling, pouring into a gel-casting tray, and allowing it to cool and solidify. After electrophoresis, bands were visualized under ultraviolet light.

## RNA immunoprecipitation (RIP)

RNA immunoprecipitation experiments were performed using an RIP kit (17-700, Merck, USA). After transfection, cardiomyocytes were lysed with lysis buffer. RBMS1 antibody was immunoprecipitated with A/G magnetic beads, and the beads were incubated with cell lysate on a magnetic rack. The beads were washed to avoid non-specific binding. RNA bound to the beads was extracted, and the expression of the target RNA was detected by agarose gel electrophoresis.

## RNA pull-down

An RNA overexpression plasmid was constructed, and the transcriptional template and target RNA were obtained. Total protein was extracted and incubated with streptavidin lysate, then with streptavidin beads with slow rotation at room temperature for 1 h to reduce background binding. The RNA was incubated with cell lysate for 1 h and then with streptavidin-coated beads. After washing, 30 μl protein sample buffer was added, mixed thoroughly, and heated in boiling water for 10 min. Interacting proteins bound to magnetic bead-RNA probe complexes were detected by western blot for RBMS1.

## Immunohistochemistry (IHC) staining

Tissue sections were dewaxed in xylene, rehydrated through a graded alcohol series, and incubated with 3% $H_2O_2$ for 10 min. Antigen retrieval was performed, and sections were blocked with goat serum, followed by incubation with specific antibodies overnight at 4 °C. The next day, sections were incubated with secondary antibodies for 30 min, and color was developed using DAB. Finally, nuclei were counterstained with hematoxylin, and slides were mounted with neutral resin.

## Masson's and H&E staining

Staining was performed using a Masson's trichrome staining kit (Solarbio, Beijing, China). Paraffin sections were dewaxed and rehydrated, stained with hematoxylin for 7 min, differentiated in acid alcohol for 10 s, and rinsed in deionized water for 1 min. Sections were then stained with Masson blue solution for 4 min, ponceau red dye for 7 min, and rinsed in 0.1% acetic acid for 1 min. Phosphomolybdate differentiation solution was applied for 1–2 min. Finally, sections were stained with aniline blue working solution for 2 min, rinsed in 0.1% acetic acid for 5 min, mounted with neutral gum, and dried overnight in a 55 °C oven. For H&E staining, dewaxed and rehydrated paraffin sections were stained with hematoxylin for 5 min, differentiated for 30 s, washed in tap water for 10–15 min, stained with eosin solution for 30 s, rinsed with deionized water, dehydrated, cleared, and mounted. We quantified the cell density in H&E-stained tissue sections to evaluate inflammatory cell infiltration in the hearts of mice. The fibrotic area and cell density within the fixed field of view were calculated and counted using image J software.

## Wheat germ agglutin (WGA) staining

Paraffin sections were dewaxed, rehydrated, subjected to antigen retrieval, and blocked at room temperature for 1 h. WGA dye, diluted appropriately, was applied to cover the tissue surface, and slides were incubated in a humidified chamber for 30 min. Sections were washed three times with PBS for 10 min each. A drop of 50% glycerol was applied, coverslips were placed, and images were acquired using a confocal microscope.

## Statistical analysis

All statistical analyses and graphs were conducted and generated using GraphPad Prism 8. In all experiments, the number of data points corresponds to the number of biologically independent samples. Continuous variables are summarized as mean ± SEM. The D'Agostino- Pearson tests ($n \geq 8$) and Shapiro–Wilk tests ($n < 8$) were used to were used to assess normal distribution of all datasets. For data conforming to normal distribution, statistical differences were compared using the Student's t-test (unpaired, two-tailed) or one-way ANOVA with Tukey's post hoc test for multiple comparisons between two groups. Data with two factors were analyzed via two-way ANOVA with Tukey's post hoc test. For data with non-normal distribution, the Mann–Whitney U test (unpaired, two-tailed) was used for comparisons between two groups, or the Kruskal–Wallis test with the Benjamini and

**The paper explained**

**Problem**

Cardiac hypertrophy is a significant contributor to heart failure and is strongly associated with increased hospitalization and readmission rates. However, the underlying mechanism of alternative splicing in cardiac hypertrophy is not yet fully elucidated. Identifying key RNA-binding proteins may offer novel therapeutic targets for the management of cardiac hypertrophy.

**Results**

In the cardiac tissues of patients with DCM and in mice exhibiting myocardial hypertrophy, we observed elevated expression levels of RBMS1. Our findings demonstrate that RBMS1 activates the PI3K/AKT signaling pathway by inducing the production of CTTN-Δe11 splicing isoforms through alternative splicing of the CTTN gene, which subsequently leads to cytoskeletal disruption and dysfunction in cardiomyocytes. Furthermore, genetic knockout of RBMS1 or pharmacological inhibition of RBMS1 using nortriptyline were shown to ameliorate pathological cardiac hypertrophy and heart failure.

**Impact**

Our research findings indicate that genetic deletion or pharmacological inhibition of RBMS1 expression can alleviate cardiac hypertrophy, thereby confirming that RBMS1 is a promising therapeutic target for the treatment of cardiac hypertrophy and heart failure.

Hochberg false discovery rate method for multiple groups. Statistical significance was set at $P < 0.05$.

# Data availability

This study includes no data deposited in external repositories.

The source data of this paper are collected in the following database record: biostudies:S-SCDT-10_1038-S44321-025-00334-z.

# Peer review information

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

## Acknowledgements

This study was supported by the Noncommunicable Chronic Diseases-National Science and Technology Major Project (2024ZD0537900, 2024ZD0537908), the National Natural Science Foundation of China (U24A20645, 82330011, 32171127, 82470413); the CAMS Innovation Fund for Medical Sciences (CIFMS, 2020-I2M-5-003); and the China Postdoctoral Science Foundation (2024MD753941).

## Author contributions

Liangliang Li: Conceptualization; Data curation; Formal analysis; Supervision; Investigation; Visualization; Methodology; Writing—original draft; Writing—review and editing. Tianyu Li: Resources; Data curation; Formal analysis; Supervision; Investigation; Methodology. Bin Wang: Resources; Data curation; Supervision; Investigation; Methodology. Jiayue Feng: Data curation; Formal analysis; Supervision; Investigation; Visualization; Methodology. Nan Zhang: Formal analysis; Methodology. Jing Zhang: Formal analysis; Supervision. Zhihui Niu: Data curation; Formal analysis. Wei Li: Data curation; Methodology. Huiying Gao: Investigation; Methodology. Qianqian Wang: Investigation;

Methodology. **Yang Liu**: Investigation; Methodology. **Yi Chen**: Formal analysis; Methodology. **Yixin Zhang**: Formal analysis; Methodology. **Yu Bian**: Data curation; Investigation. **Tengfei Pan**: Data curation; Investigation. **Siqi Sheng**: Formal analysis; Supervision. **Xuelian Li**: Data curation; Methodology. **Jinping Liu**: Resources; Writing—review and editing. **Baofeng Yang**: Conceptualization; Validation; Writing—review and editing. **Haihai Liang**: Conceptualization; Resources; Data curation; Formal analysis; Supervision; Funding acquisition; Validation; Writing—original draft; Writing—review and editing.

Source data underlying figure panels in this paper may have individual authorship assigned. Where available, figure panel/source data authorship is listed in the following database record: biostudies:S-SCDT-10_1038-S44321-025-00334-z.

## Disclosure and competing interests statement

The authors declare no competing interests.

# Expanded View Figures

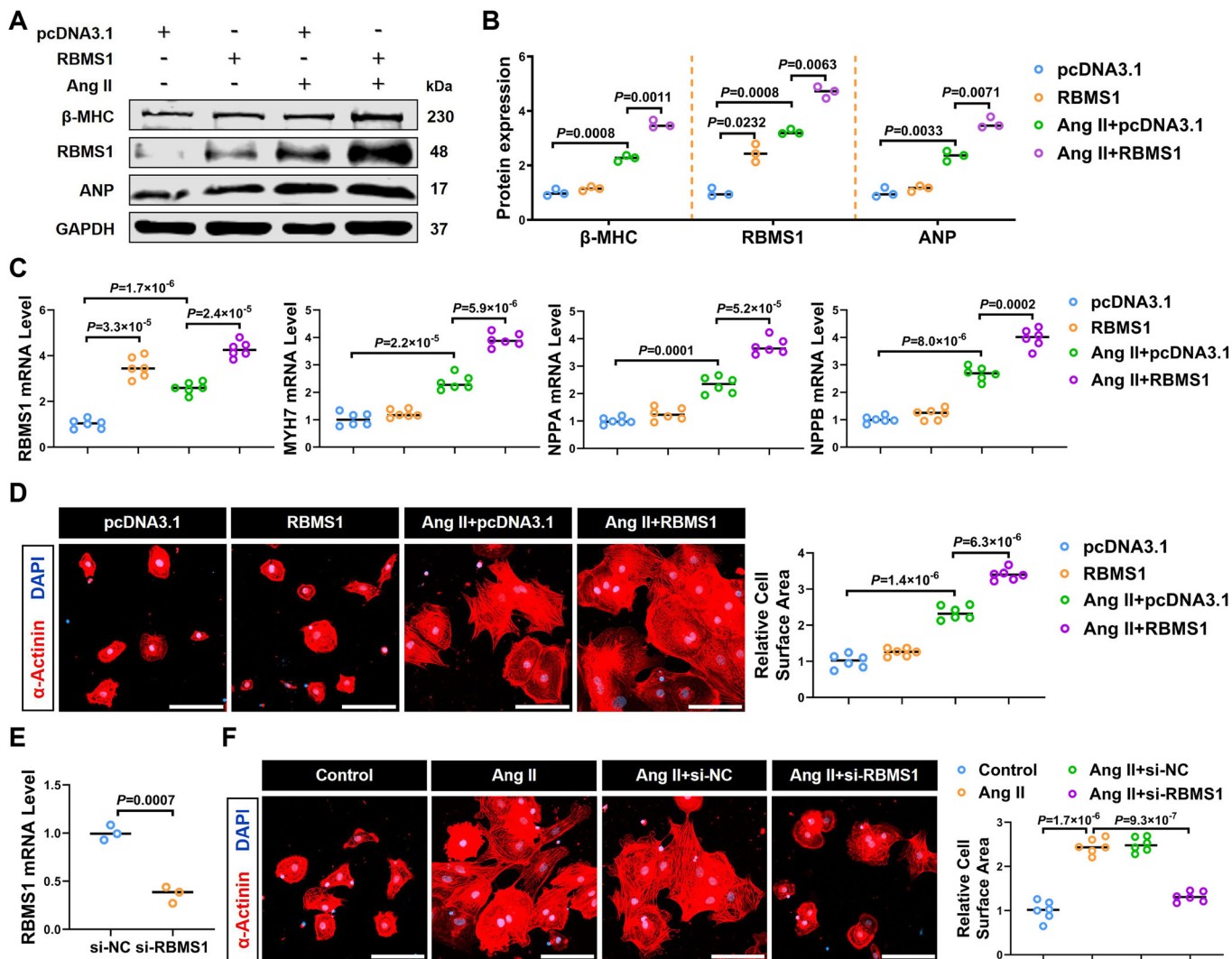

**Figure EV1. RBMS1 facilitates cardiac hypertrophy induced by Ang II in hiPSC-CMs.**

(A, B) Western blotting and quantification showing protein levels of β-MHC, RBMS1, and ANP in hiPSC-CMs transfected with RBMS1 in response to Ang II stimulation ($n = 3$). (C) Quantification of mRNA levels of RBMS1, MYH7, NPPA, and NPPB ($n = 6$). (D) Representative immunofluorescence staining of α-actinin and quantification in hiPSC-CMs transfected with RBMS1 ($n = 6$). Scale bar = 50 μm. (E) Quantification of mRNA levels of RBMS1 in hiPSC-CMs transfected with si-NC or si-RBMS1 ($n = 3$). (F) Representative immunofluorescence staining of α-actinin and quantification in hiPSC-CMs transfected with si-RBMS1 ($n = 6$). Scale bar = 50 μm. Data information: data are shown as mean ± SEM, P values were analyzed with unpaired Student's t test (E) and one-way ANOVA test (B, C, D, F). A dot represents an independent biological sample. Source data are available online for this figure.

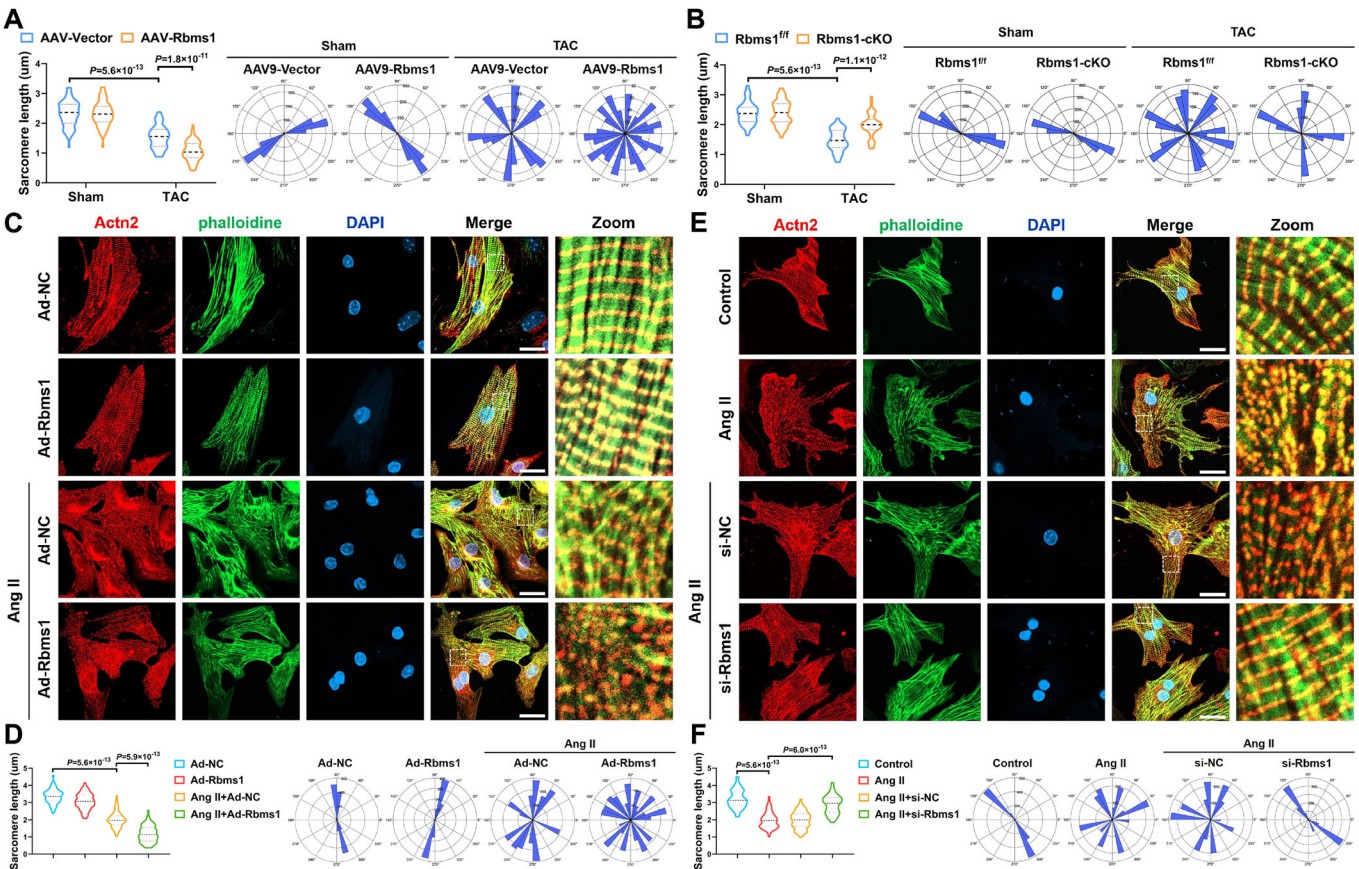

**Figure EV2.  RBMS1 regulates cardiac hypertrophy via the sarcomere and cytoskeleton of cardiomyocytes.**

(**A**) Quantification of sarcomere length and representative polarity histogram of sarcomere organization in RBMS1 overexpression mice treatment with TAC surgery ($n = 70$). (**B**) Quantification of sarcomere length and representative polarity histogram of sarcomere organization in RBMS1-cKO mice treatment with TAC surgery ($n = 70$). (**C**, **D**) Quantification of sarcomere length and representative polarity histogram of sarcomere organization in NMCMs transfected with Ad-RBMS1 in response to Ang II stimulation ($n = 70$). (**E**, **F**) Quantification of sarcomere length and representative polarity histogram of sarcomere organization in NMCMs transfected with si-RBMS1 and subsequently treated with Ang II ($n = 70$). Data information: data are shown as mean ± SEM, *P* values were analyzed with one-way ANOVA test (**A**, **B**, **D**, **F**), polarity histograms were generated by MATLAB R2024b (**A**, **B**, **D**, **F**). Source data are available online for this figure.

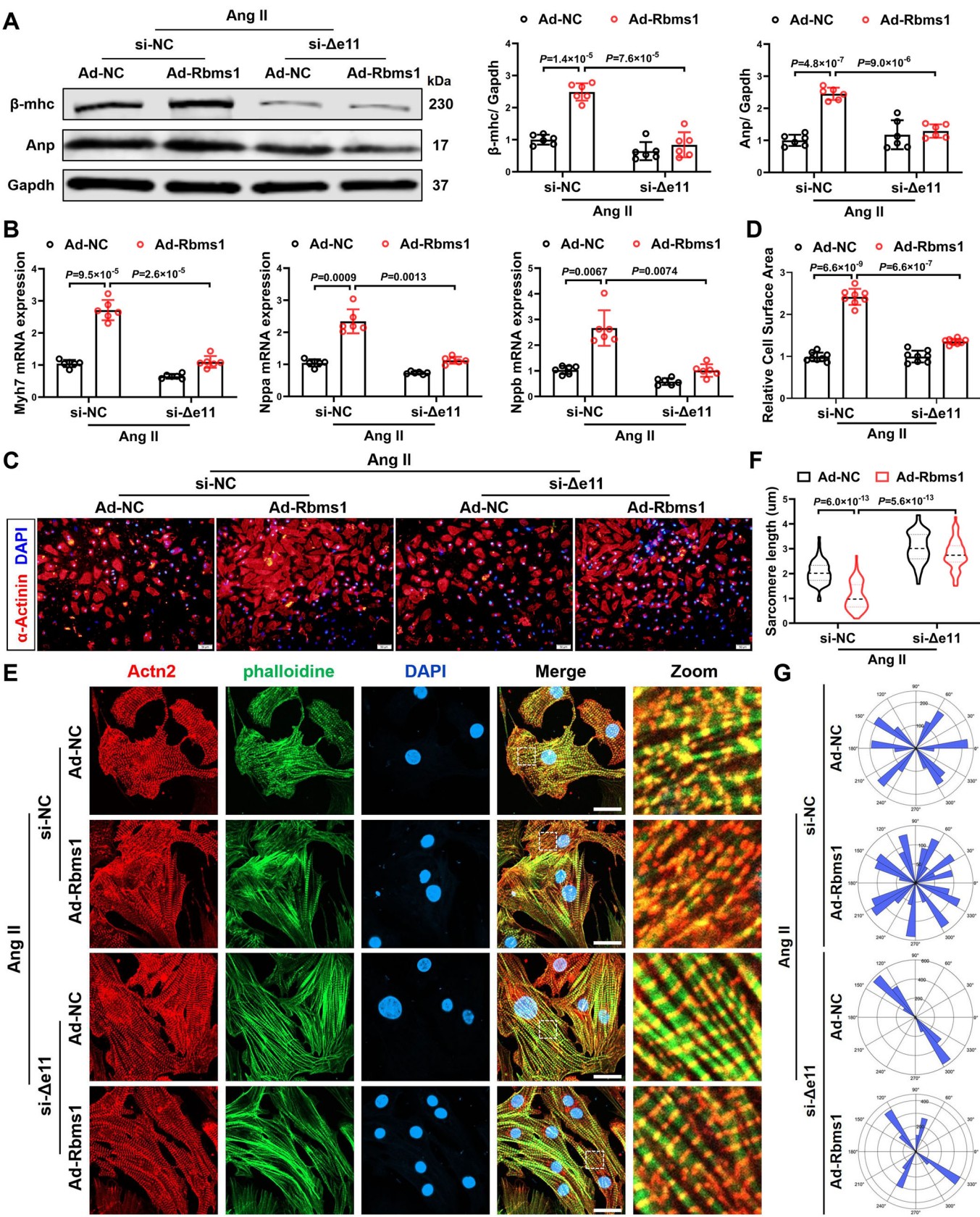

◀ **Figure EV3. The pro-hypertrophic function of RBMS1 depends on splicing CTTN.**

(A) Western blotting and quantification showing protein levels of β-MHC and ANP in RBMS1 overexpression NMCMs transfected with si-Δe11 in response to Ang II stimulation ($n = 6$). (B) Quantification of mRNA levels of MYH7, NPPA, and NPPB ($n = 6$). (C, D) Representative immunofluorescence staining of α-actinin and quantification of cell surface area ($n = 8$), scale bar = 50 μm. (E) Immunofluorescence staining of ACTN2 and phalloidine showed the disorganization of sarcomere and cytoskeleton in NMCMs, scale bar = 20 μm. (F, G) Quantification of sarcomere length and representative polarity histogram of sarcomere organization in NMCMs ($n = 70$). Data information: data are shown as mean ± SEM, $P$ values were analyzed with one-way ANOVA test (A, B, D, F), polarity histograms were generated by MATLAB R2024b (G). Source data are available online for this figure.

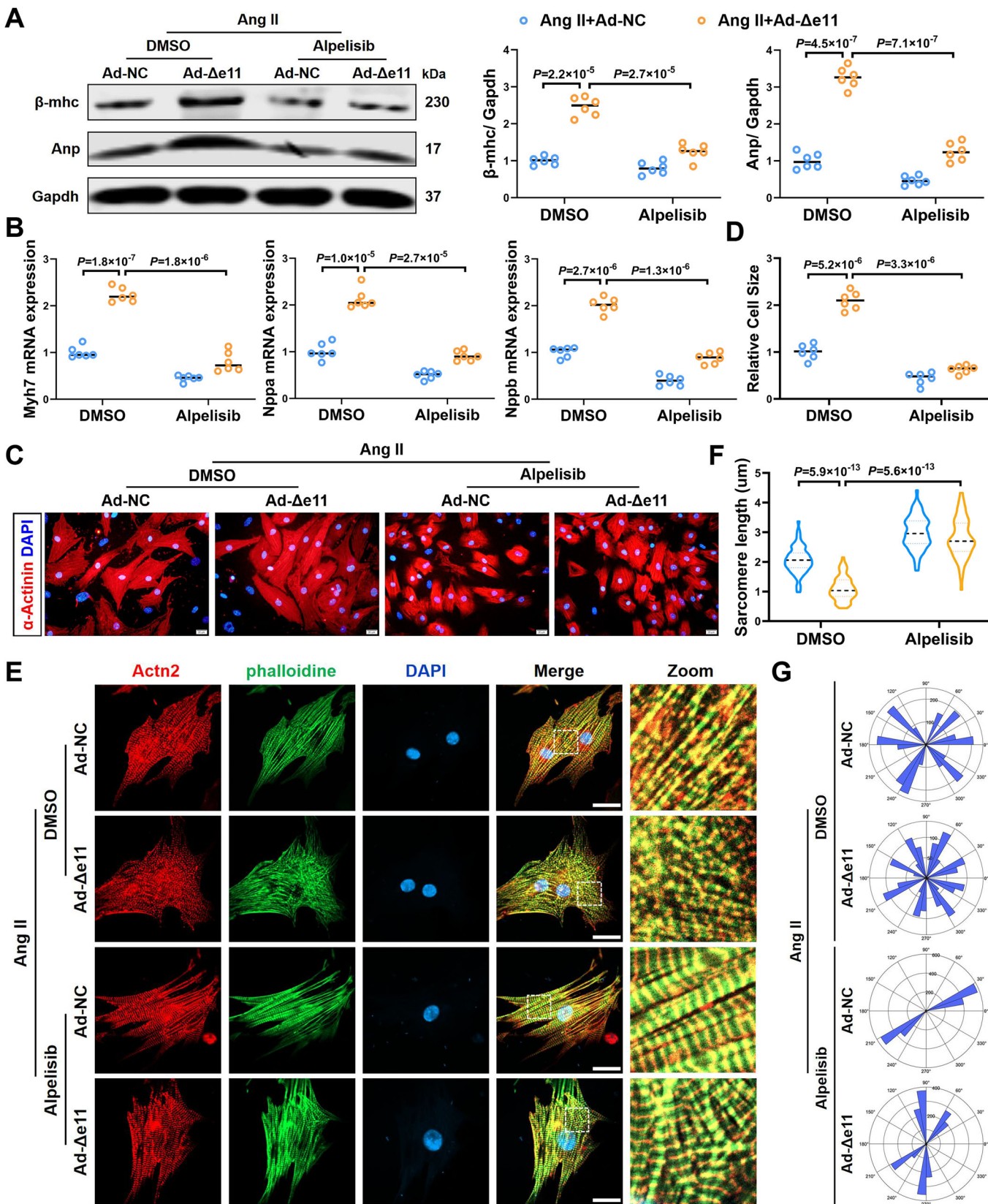

◀ **Figure EV4. Alpelisib mitigates cardiac hypertrophy induced by CTTN-Δe11.**

(A) Western blotting and quantification showing protein levels of β-MHC and ANP in CTTN-Δe11 overexpression NMCMs treatment with Alpelisib in response to Ang II stimulation ($n = 6$). (B) Quantification of mRNA levels of β-MHC, ANP, and BNP ($n = 6$). (C, D) Representative immunofluorescence staining of α-actinin and quantification of cell surface area ($n = 6$), scale bar $= 50$ μm. (E) Immunofluorescence staining of ACTN2 and phalloidine showed the disorganization of sarcomere and cytoskeleton, scale bar $= 20$ μm. (F, G) Quantification of sarcomere length and representative polarity histogram of sarcomere organization in NMCMs ($n = 70$). Data information: data are shown as mean ± SEM, P values were analyzed with one-way ANOVA test (A, B, D, F), polarity histograms were generated by MATLAB R2024b (G). Source data are available online for this figure.

