## [Peer Review File · EMBO Molecular Medicine]

RBMS1 Orchestrates Cardiac Hypertrophy by Facilitating CTTN Splice-switching and Sarcomere Dynamics

Haihai Liang, Liangliang Li, Tianyu Li, Bin Wang, Jiayue Feng, Nan Zhang, Jing Zhang, Zhihui Niu, Wei Li, Huiying Gao, Qianqian Wang, Yang Liu, Yi Chen, Yixin Zhang, Yu Bian, Tengfei Pan, Siqi Sheng, Xuelian Li, Jinping Liu, and Baofeng Yang

Corresponding authors: Haihai Liang (lianghaihai@ems.hrbmu.edu.cn) , Baofeng Yang (yangbf@ems.hrbmu.edu.cn), Jinping Liu (liujinping@znhospital.cn)

Review Timeline:

Submission Date:	5th Jun 25
Editorial Decision:	2nd Jul 25
Revision Received:	31st Aug 25
Editorial Decision:	23rd Sep 25
Revision Received:	9th Oct 25
Accepted:	17th Oct 25

Editor: Zeljko Durdevic

Transaction Report:

2nd Jul 2025

Dear Prof. Liang,

Thank you for the submission of your manuscript to EMBO Molecular Medicine. We have now received feedback from the three reviewers who agreed to evaluate your manuscript. As you will see from the reports, all three referees recognize potential interest of the study, but they also raise important concerns that should be addressed in a major revision. If you would like to discuss further the points raised by the referees, I am available to do so via email or video. Let me know if you are interested in this option.

We would welcome the submission of a revised version within three months for further consideration. Please let us know if you require longer to complete the revision.

I look forward to receiving your revised manuscript.

Yours sincerely,

Zeljko Durdevic

Zeljko Durdevic
Senior Editor
EMBO Molecular Medicine

We require:

- 1) A .docx formatted version of the manuscript text (including legends for main figures, EV figures and tables). Please make sure that the changes are highlighted to be clearly visible.
- 2) Individual production quality figure files as .eps, .tif, .jpg (one file per figure). For guidance, download the 'Figure Guide PDF': (<https://www.embopress.org/page/journal/17574684/authorguide#figureformat>).
- 3) A .docx formatted letter INCLUDING the reviewers' reports and your detailed point-by-point responses to their comments. As part of the EMBO Press transparent editorial process, the point-by-point response is part of the Review Process File (RPF), which will be published alongside your paper.
- 4) A complete author checklist, which you can download from our author guidelines (<https://www.embopress.org/page/journal/17574684/authorguide#submissionofrevisions>). Please insert information in the checklist that is also reflected in the manuscript. The completed author checklist will also be part of the RPF.
- 5) Please note that all corresponding authors are required to supply an ORCID ID for their name upon submission of a revised manuscript.
- 6) It is mandatory to include a 'Data Availability' section after the Materials and Methods. Before submitting your revision, primary

datasets produced in this study need to be deposited in an appropriate public database, and the accession numbers and database listed under 'Data Availability'. Please remember to provide a reviewer password if the datasets are not yet public (see <https://www.embopress.org/page/journal/17574684/authorguide#dataavailability>).

12) Author contributions: You will be asked to provide CRediT (Contributor Role Taxonomy) terms in the submission system. These replace a narrative author contribution section in the manuscript.

13) A Conflict of Interest statement should be provided in the main text.

14) Every published paper now includes a 'Synopsis' to further enhance discoverability. Synopses are displayed on the journal webpage and are freely accessible to all readers. They include a short stand first (maximum of 300 characters, including space) as well as 2-5 one-sentences bullet points that summarizes the paper. Please write the bullet points to summarize the key NEW findings. They should be designed to be complementary to the abstract - i.e. not repeat the same text. We encourage inclusion of key acronyms and quantitative information (maximum of 30 words / bullet point). Please use the passive voice. Please attach

these in a separate file or send them by email, we will incorporate them accordingly.

15) Include a Reagents and Tools Table as part of the Methods section, which can be downloaded from our author guidelines (<https://www.embopress.org/page/journal/17574684/authorguide#structuredmethods>)

***** Reviewer's comments *****

Referee #1 (Comments on Novelty/Model System for Author):

No comments. The models used in this study are sufficient if not excessive.

Referee #1 (Remarks for Author):

In this study, Kiu, Yang, and Liang and colleagues focused on the molecular mechanism of cardiac hypertrophic growth and heart failure under pressure overload. The authors did impressive amount of work and demonstrated a critical role of an RNA-binding protein (RBMS1) in cardiomyocyte hypertrophic growth. Using both gain- and loss-of-function approaches, the authors showed that RBMS1, when activated under pressure overload in cardiomyocytes, caused alternative splicing of CTTN. The resulting CTTN-deltae11 activated the pro-growth PI3K signaling. Inhibition of either CTTN-deltae11 or PI3K attenuated RBMS1 related cardiac hypertrophy and heart failure. Overall, this study is well designed and nicely executed. I have a few questions for the authors to consider.

1. In the RBMS1 deletion experiments, a WT control was used. It is unclear what genotype of this control. Is it pure WT, flox/flox, or aMHC-Cre? Ideally, both flox/flox and aMHC-Cre should be used to exclude any effects of gene floxing and Cre expression.
2. Although the authors identified CTTN as a key target of RBMS1, other targets may exist that contribute to the observed phenotypes of RBMS1. A discussion on this limitation should be included.
3. From my understanding, it remains unclear how CTTN-deltae11 activates PI3K. Does it involve PI3 production, PI3K translation, or any other mechanism?

Referee #2 (Comments on Novelty/Model System for Author):

There are technical concerns with the in vivo datasets as presented, but I suspect that the data are overall rigorous and reliable. See below.

Referee #2 (Remarks for Author):

This is a new submission by Li et al addressing the role of the splicing factor RBMS1 in pressure overload-associated cardiac remodeling, including the regulation of cortactin (CTTN) splicing and downstream PI3K-AKT signaling in cardiac myocytes. The study follows a similar study in press in the European Heart Journal by the same group in which fibroblast RBMS1 was studied in ischemic cardiomyopathy. However, this study presents the first description of the function RBMS1 and CTTN splicing in cardiac myocytes. This study is quite comprehensive including gain and loss of function in vivo for RBMS1 and CTTN, as well as elegant whole genome splicing analyses and molecular analyses. Proof-of-concept is provided for the treatment of pressure overload disease in mice using AAV gene therapy vectors and Nortriptyline, a small chemical known to inhibit RBMS1 expression. Besides the concerns mentioned below, the major shortcoming of the paper as currently presented is with the writing, which in many places is not clear, including figures that are not adequately explained by the text or legends (e.g. Fig. 4B). Nevertheless, the use of diverse methods in this study results in what is overall a convincing description of a novel cardiomyocyte regulatory pathway with potential translational significance.

Figure 2D - the TAC B-mode images do not seem to match the m-mode presented, raising a question about the analysis of systolic function. Are the B-modes in the same phase of the contractile cycle? Insufficient methodological detail is provided to allow assessment of the in vivo data. Likewise, the B-mode (?) images in Figure 3B are poor quality.

Figure 2F - What are the "heart size" pictures of? They do not seem to be intact hearts.

With regards to TAC surgery, are there any data that the induced pressure overload was similar among cohorts?

What exactly was measured in Supplementary 2L? In general, the figures legends need to be more explicit about what is shown.

"Polarity histogram of sarcomere organization" is not a standard hypertrophy assay and needs to be explained/validated. The round shape and immature phenotype of iPSC-CMs has made them a highly controversial model for hypertrophy. The iCM experiments would be better performed in adult rat ventricular myocytes or at least neonatal rat ventricular myocytes. Regardless, the number of cells analyzed and whether the datapoints represent independent CM lots needs to be specified.

Decreased sarcomere length is not usually associated with pathological hypertrophy. Were the cells/tissues relaxed before staining in the various experiments?

The authors should discuss briefly the mechanism by which RBMS1 regulates splicing, if known. Also, it would be appropriate to contrast briefly the findings presented herein with the results obtained in fibroblasts in their recent paper.

Regarding PI3K - p100 α and p100 γ have different roles in physiological and pathological hypertrophy. Which isoform is regulated by RBMS1-CTTN?

Minor comments:

Introduction lines 54-67 - The description of pathological cardiac hypertrophy and heart failure is not exactly accurate and should be re-written. For example, hypertrophy can occur in the absence of primary hypertension and volume overload.

The paper needs to be edited for English usage throughout, including the methods, e.g.:

Line 130 - "flanjoined"?

Line 134 - "selfing"

Line 237 - "dismissed"

Lines 173-187 - GSE135055 and SCP1303 need to be introduced. Likewise, the hearts used for protein levels. What was the physiological state of the TAC mice when the analysis was done - concentric hypertrophy or dilated?

Line 190 -What line of BL/6 mice were used, N or J? AAV dose should be stated for each experiment.

Supplementary Figure 1 - the figure legend does not match the figure.

Supplement - ultrasound description - Vevo 2000? The methods description is insufficient - the acquired ultrasound views need to be properly described, as well as how calculations were performed.

Supplement line 131 - Is "genes" correct?

Abbreviations need to be defined in the figures - Figure 4B and Supplementary Figure 4E,F are not interpretable.

Line 450 -WGCNA needs to be defined. Likewise, the meaning of the colors needs to be described.

ANF, BNP, and β MHC should preferably be referred to by their gene names when discussing mRNA levels or least be capitalized consistently in the figures and legends.

Tables should be provided of the genes regulated by alternative splicing.

Referee #3 (Comments on Novelty/Model System for Author):

Excellent work with vast arrays of very suitable tools and preclinical models

Referee #3 (Remarks for Author):

This is an excellent report highlighting the role of RBMS1 in cardiac hypertrophy using an impressive array of transgenic and experimental mouse models of heart disease. The work provides also very deep mechanistic insights on how RBMS1 facilitating

CTTN splice-switching and how it is linked to sarcomere dynamics/function and to heart disease. I have only some minor points for improvement:

1. Many transgenic mouse models were used in the study, e.g. RBMS1 cKO which was bought from a company. However, all detailed information was missing how exactly this floxed line was made. Even if it was acquired commercially, the authors have to provide more detailed information in the supplement, e.g. on exact targeted exon, CRISPR construct, targeting strategy, methods used to generate this and other mouse lines which were not described in detail.

2. The description of methods for echocardiography is weird and not standard for the field. Which instrument and which type of anesthesia was used for that in detail?

3. I appreciate patient data provided for DCM patient group. However, reporting of healthy donor heart tissues seems rather intransparent. Where did these subjects come from - hearts rejected for transplantation, surgeries for other indications or similar? More information, also on the ethical approval is needed here.

Response to Reviewers' Comments

Reviewer #1 (Remarks to the Author):

In this study, Liu, Yang, and Liang and colleagues focused on the molecular mechanism of cardiac hypertrophic growth and heart failure under pressure overload. The authors did impressive amount of work and demonstrated a critical role of an RNA-binding protein (RBMS1) in cardiomyocyte hypertrophic growth. Using both gain- and loss-of-function approaches, the authors showed that RBMS1, when activated under pressure overload in cardiomyocytes, caused alternative splicing of CTTN. The resulting CTTN-deltae11 activated the pro-growth PI3K signaling. Inhibition of either CTTN-deltae11 or PI3K attenuated RBMS1 related cardiac hypertrophy and heart failure. Overall, this study is well designed and nicely executed. I have a few questions for the authors to consider.

Reply: We thank the reviewer for your constructive comments and have addressed them carefully below.

1. In the RBMS1 deletion experiments, a WT control was used. It is unclear what genotype of this control. Is it pure WT, flox/flox, or aMHC-Cre? Ideally, both flox/flox and aMHC-Cre should be used to exclude any effects of gene floxing and Cre expression.

Reply: We sincerely appreciate the reviewers' insightful comments. In the RBMS1 deletion mouse experiment, we utilized flox/flox mice as controls. Unfortunately, due to an oversight during data organization, the genotypes of the control group were not clearly indicated, leading to confusion. We have since revised the results section and explicitly clarified the genotype information in the Materials and Methods.

In this study, we did not employ aMHC-Cre mice as controls for the RBMS1-deficient group. This decision was based on our understanding that

aMHC-Cre mice do not exhibit cardiac dysfunction within the timeframe of this experiment. Numerous studies have demonstrated that aMHC-Cre mice may develop cardiotoxic effects after six months, including reduced survival rates, impaired cardiac function, myocardial fibrosis, and DNA damage^[1-3]. However, in our experimental design, aMHC-Cre mice underwent TAC surgery at two months of age and were euthanized two months later for analysis—well before the onset of potential cardiotoxic effects. Furthermore, during the breeding process, we monitored the survival rates of both aMHC-Cre and flox/flox mice and observed no significant differences between the two genotypes (Response Fig. 1).

To avoid any misinterpretation regarding the use of aMHC-Cre mice in this study, we have included additional clarification in lines 521-526 of the Discussion section.

Response figure 1: Kaplan-Meier analysis of WT, RBMS1^{flox/flox}, and α-MHC-Cre mice (n=15).

2. Although the authors identified CTTN as a key target of RBMS1, other targets may exist that contribute to the observed phenotypes of RBMS1. A discussion on this limitation should be included.

Reply: We sincerely appreciate the reviewers' insightful comments. We fully concur with the reviewer's perspective that RBMS1 is likely to regulate numerous downstream target genes beyond CTTN. Accordingly, we have compiled a comprehensive list of potential downstream target genes regulated by RBMS1, as detailed in Supplementary Table 1. Our study primarily focused on elucidating the functional significance of the RBMS1-CTTN- $\Delta e11$ axis within the context of the PI3K-AKT pathway and myocardial cytoskeletal abnormalities. Nevertheless, we acknowledge that attributing the entire cardiac hypertrophy solely to CTTN- $\Delta e11$ may be an oversimplification. To address this concern, we have included a detailed discussion of the study's limitations in lines 667-693 of the Discussion.

Although our investigation identified CTTN as a key functional mediator through which RBMS1 contributes to cardiac hypertrophy, we recognize that the study has certain limitations. Our findings confirm that RBMS1 is an RNA-binding protein capable of interacting with CTTN and modulating the PI3K-AKT pathway and myocardial cytoskeletal dysfunction. However, in addition to CTTN, there are likely other unidentified target genes of RBMS1 that may play roles in cardiac hypertrophy via distinct biological processes or pathways. While this study validated CTTN as a downstream target of RBMS1, the potential contributions of other RBMS1-regulated targets warrant further investigation. The focus of this study is to elucidate the mechanism by which RBMS1 participates in myocardial hypertrophy through CTTN splicing variant in cardiomyocytes. Interestingly, this phenomenon differs from the alternative splicing events we previously observed in fibroblasts. In our earlier research, we demonstrated that RBMS1 in fibroblasts promotes cardiac fibrosis and heart failure by regulating LM07 alternative splicing, which subsequently activates the AP-1/TGF- β pathway^[4]. However, in cardiomyocytes, we identified splicing events that are distinctly different from those in fibroblasts. This indicates that the alternative splicing regulated by RBMS1 is cell-type-specific, potentially due to variations in transcriptional profiles or splicing regulatory mechanisms across different cell types. As an RBP, RBMS1 typically interacts with target RNA in a sequence-specific manner through its

RNA recognition motif (RRM). The regulation of alternative splicing by RBPs is a highly complex process, involving spliceosome assembly, splice site selection, and the recruitment or inhibition of core spliceosomal components or other regulatory proteins. Further investigation is required to delineate the precise molecular mechanisms through which RBMS1 governs the splicing of critical target genes.

3. From my understanding, it remains unclear how CTTN-deltae11 activates PI3K. Does it involve PI3 production, PI3K translation, or any other mechanism?

Reply: We thank the reviewer for this insightful question. Current research has preliminarily demonstrated that P100 α plays a predominant role in physiological hypertrophy rather than in pathological hypertrophy^[5], whereas P100 γ is primarily involved in the induction of pathological hypertrophy^[6]. During physiological hypertrophy, early activation of p110 α enhances myocardial contractility and helps maintain cardiac output in the short term^[7]. Evidence suggests that P100 γ , rather than P100 α , is primarily responsible for early compensatory pathological hypertrophy^[8]. However, the regulatory role of P100 α in advanced pathological hypertrophy remains ambiguous. Nevertheless, sustained activation of p110 α leads to excessive reactive oxygen species (ROS) production, mitochondrial dysfunction, disruption of calcium homeostasis, and myocardial fibrosis, thereby facilitating the progression of decompensated pathological hypertrophy^[9].

Our findings, validated at both *in vitro* and *in vivo* levels, demonstrate that RBMS1 promotes the expression of P100 α , while knockdown of $\Delta e11$ significantly suppresses P100 α production (Response Fig. 2a-b). Furthermore, Alpelisib, a specific inhibitor of P100 α , competitively binds to the ATP-binding pocket of P110 α , thereby inhibiting the generation of PIP3 (Supplementary Fig. 13 and 14). Our study confirms that Alpelisib effectively alleviates myocardial hypertrophy induced by RBMS1 and $\Delta e11$. Collectively, these results indicated

that in decompensated pathological hypertrophy, the RBMS1- Δ e11 axis promotes the conversion of PIP2 to PIP3 via P100 α , thereby activating the PI3K-AKT signaling pathway. The above content has been revised in the manuscript.

Response figure 2: (a) Western blotting and quantification showing protein levels of P100 α and P100 γ in RBMS1 overexpression NMCs treatment with Alpelisib in response to Ang II stimulation (n=4). (b) Western blotting and quantification showing protein levels of P100 α and P100 γ in RBMS1 overexpression NMCs transfected with si- Δ e11 in response to Ang II stimulation (n=4).

Reference

1. Agah R, Frenkel PA, French BA, Michael LH, Overbeek PA, Schneider MD. Gene recombination in postmitotic cells. Targeted expression of Cre recombinase provokes cardiac-restricted, site-specific rearrangement in adult ventricular muscle in vivo. *J Clin Invest.* 1997 Jul 1;100(1):169-79. doi: 10.1172/JCI119509.
2. Pugach EK, Richmond PA, Azofeifa JG, Dowell RD, Leinwand LA. Prolonged Cre expression driven by the α -myosin heavy chain promoter can be cardiotoxic. *J Mol Cell Cardiol.* 2015 Sep;86:54-61. doi: 10.1016/j.yjmcc.2015.06.019.

3. Rehmani T, Salih M, Tuana BS. Cardiac-Specific Cre Induces Age-Dependent Dilated Cardiomyopathy (DCM) in Mice. *Molecules*. 2019 Mar 26;24(6):1189. doi: 10.3390/molecules24061189.
4. Li L, Guo J, Feng J, Li T, Xu B, Li W, et al. Deficiency of the RNA-binding protein RBMS1 improves myocardial fibrosis and heart failure. *Eur Heart J*. 2025 Jun 5;ehaf370. doi: 10.1093/eurheartj/ehaf370
5. McMullen JR, Shioi T, Zhang L, Tarnavski O, Sherwood MC, Kang PM, et al. Phosphoinositide 3-kinase(p110alpha) plays a critical role for the induction of physiological, but not pathological, cardiac hypertrophy. *Proc Natl Acad Sci U S A*. 2003 Oct 14;100(21):12355-60. doi: 10.1073/pnas.1934654100.
6. Oudit GY, Crackower MA, Eriksson U, Sarao R, Kozieradzki I, Sasaki T, et al. Phosphoinositide 3-kinase gamma-deficient mice are protected from isoproterenol-induced heart failure. *Circulation*. 2003 Oct 28;108(17):2147-52. doi: 10.1161/01.CIR.0000091403.62293.2B.
7. McMullen JR, Amirahmadi F, Woodcock EA, Schinke-Braun M, Bouwman RD, Hewitt KA, et al. Protective effects of exercise and phosphoinositide 3-kinase(p110alpha) signaling in dilated and hypertrophic cardiomyopathy. *Proc Natl Acad Sci U S A*. 2007 Jan 9;104(2):612-7. doi: 10.1073/pnas.0606663104.
8. Naga Prasad SV, Esposito G, Mao L, Koch WJ, Rockman HA. Gbetagamma-dependent phosphoinositide 3-kinase activation in hearts with in vivo pressure overload hypertrophy. *J Biol Chem*. 2000 Feb 18;275(7):4693-8. doi: 10.1074/jbc.275.7.4693.
9. Bass-Stringer S, Bernardo BC, Yildiz GS, Matsumoto A, Kiriazis H, Harmawan CA, et al. Reduced PI3K(p110 α) induces atrial myopathy, and PI3K-related lipids are dysregulated in athletes with atrial fibrillation. *J Sport Health Sci*. 2025 Jan 16;14:101023. doi: 10.1016/j.jshs.2025.101023.

Reviewer #2 (Remarks to the Author):

This is a new submission by Li et al addressing the role of the splicing factor RBMS1 in pressure overload-associated cardiac remodeling, including the regulation of cortactin (CTTN) splicing and downstream PI3K-AKT signaling in cardiac myocytes. The study follows a similar study in press in the European Heart Journal by the same group in which fibroblast RBMS1 was studied in ischemic cardiomyopathy. However, this study presents the first description of the function RBMS1 and CTTN splicing in cardiac myocytes. This study is quite comprehensive including gain and loss of function in vivo for RBMS1 and CTTN, as well as elegant whole genome splicing analyses and molecular analyses. Proof-of-concept is provided for the treatment of pressure overload disease in mice using AAV gene therapy vectors and Nortriptyline, a small chemical known to inhibit RBMS1 expression. Besides the concerns mentioned below, the major shortcoming of the paper as currently presented is with the writing, which in many places is not clear, including figures that are not adequately explained by the text or legends (e.g. Fig. 4B). Nevertheless, the use of diverse methods in this study results in what is overall a convincing description of a novel cardiomyocyte regulatory pathway with potential translational significance.

Reply: We thank the reviewer for your constructive comments and have addressed them carefully below.

1. Figure 2D - the TAC B-mode images do not seem to match the m-mode presented, raising a question about the analysis of systolic function. Are the B-modes in the same phase of the contractile cycle? Insufficient methodological detail is provided to allow assessment of the in vivo data. Likewise, the B-mode (?) images in Figure 3B are poor quality.

Reply: Thank you for your detailed and constructive review of the manuscript. To ensure that the B-mode ultrasound captures the same phase of the cardiac contraction cycle, we standardized the procedure by fixing the mouse's body position and probe angle during image acquisition. Subsequently, we adjusted the imaging plane to visualize the longest axis of the heart, ensuring clear exposure of the ascending aorta and aortic valve (typically visible on the right side of the B-mode). B-mode images were recorded once the mitral valve reached maximal contraction. It is possible that due to limitations in image layout or the static nature of the images, there was a misinterpretation regarding the visual correlation between the B- and M-mode images in Figure 2D. To address this, we have provided a more comprehensive description of the echocardiography protocol in the Materials and Methods section and updated the representative results accordingly (Response Fig. 1a). Similarly, the representative images in Figure 3B have also been replaced (Response Fig. 2b).

Response figure 1: (a) Representative echocardiography of WT mice injected with AAV9-Vector or AAV9-RBMS1 for 1 week and subsequently subjected to

sham or TAC surgery for 8 weeks. (b) Representative echocardiography of RBMS1^{fl/fl} and RBMS1-cKO mice with sham or TAC surgery.

2. Figure 2F - What are the "heart size" pictures of? They do not seem to be intact hearts.

Reply: We sincerely appreciate the reviewers' careful evaluation of the quality of the result. Given that the original heart size in Figure 2F was of insufficient quality and could potentially lead to misinterpretation, we have replaced it with a clearer and more representative set of results (Response Fig. 2).

Response figure 2: Heart size of WT mice injected with AAV9-Vector or AAV9-RBMS1 for 1 week and subsequently subjected to sham or TAC surgery for 8 weeks, scale bar=5 mm.

3. With regards to TAC surgery, are there any data that the induced pressure overload was similar among cohorts?

Reply: We sincerely appreciate the reviewers' constructive feedback regarding whether the pressure overload is comparable across different groups in the TAC surgical model. All TAC procedures in this study were performed by the same experienced operator. During surgery, a standardized 26-G needle was used to induce constriction of the aortic arch, thereby ensuring uniform aortic narrowing across all mice. Subsequently, echocardiographic assessments were conducted to measure the peak velocity of aortic arch blood flow (Response Fig. 3a-b), serving as an indicator of comparable pressure overload among the animals. However, due to potential variations in the anatomical position and angulation of the aortic arch following

TAC surgery, it was not always feasible to obtain reliable measurements of peak flow velocity in all mice during the ultrasound examination.

Response figure 3: (a) Representative peak velocity echocardiography of aortic arch blood flow in sh-Δe11 mice treatment with TAC surgery. (b) Statistical graph of peak velocity of aortic arch blood flow (n=5).

4. What exactly was measured in Supplementary 2L? In general, the figures legends need to be more explicit about what is shown.

Reply: We sincerely appreciate the reviewers' valuable feedback regarding the insufficient clarity of the legend in Supplementary Figure 2L. We fully agree that legends should be clear and self-explanatory. In Supplementary Figure 2L, we quantified the cell density in H&E-stained tissue sections to evaluate inflammatory cell infiltration in the hearts of mice following TAC surgery or treatment. To address this concern and improve the overall clarity of the figure presentation, we have revised the legend of Supplementary Figure 2L and provided descriptions of the statistical methods and software used in the Materials and Methods.

5. "Polarity histogram of sarcomere organization" is not a standard hypertrophy assay and needs to be explained/validated. The round shape and immature phenotype of iPSC-CMs has made them a highly controversial model for hypertrophy. The iCM experiments would be better performed in adult rat ventricular myocytes or at least neonatal rat

ventricular myocytes. Regardless, the number of cells analyzed and whether the datapoints represent independent CM lots needs to be specified.

Reply: We sincerely appreciate the reviewer's insightful comments regarding the novelty of our sarcomere analysis method, iPSC-CM model, primary cardiomyocytes, and sample capacity. We address each point below:

1. Explanation and Justification of the "Polarity Histogram of Sarcomere Organization" Method: A hallmark of pathological hypertrophy is the disorganization of the contractile apparatus, leading to inefficient contraction. We acknowledge that the polarity histogram is not a standard hypertrophy assay. While cell size and expression of fetal genes (like ANP, BNP, β -MHC) are standard markers, direct assessment of sarcomere structure and alignment provides complementary and mechanistically relevant information. It was developed specifically to quantify the degree and length of sarcomere alignment and organization in an objective manner from confocal images of cardiomyocytes^[1].

2. The rationale for utilizing the iPSC-CMs model in hypertrophy research: We fully acknowledge the reviewers' concerns regarding the immature phenotype of iPSC-CMs, which represents a well-documented limitation in the field. Current evidence suggests that iPSCs exhibit morphological heterogeneity upon differentiation into cardiomyocytes, presenting as round, oval, or irregular shapes. However, iPSC-CMs with higher purity tend to display round or oval morphology^[2]. In our differentiation protocol, iPSC-CMs with lower purity frequently exhibited irregular morphology, whereas purified iPSC-CMs predominantly displayed round or oval shapes (purity $\geq 90\%$). Furthermore, the purified iPSC-CMs used in this study must be utilized within one week; otherwise, non-cardiomyocyte contaminants may emerge, leading to morphological irregularities. Accumulating evidence supports the capacity of iPSC-CMs to elicit hypertrophic responses under various stimulatory

conditions^[3,4]. Therefore, this study employed AngII stimulation to investigate the regulatory role of RBMS1 in myocardial hypertrophy.

Our findings demonstrate that RBMS1 contributes to the progression of myocardial hypertrophy in mice via cardiomyocyte hypertrophy. As an RNA-binding protein, RBMS1 exhibits a high degree of evolutionary conservation, with a sequence homology of up to 96.77% between humans and mice. Given this conservation, we utilized iPSC-CMs to evaluate the regulatory effects of RBMS1 on human cardiomyocytes, thereby extending our understanding of its role in human myocardial hypertrophy. Additionally, we investigated the regulatory function of RBMS1 in primary rat cardiomyocytes. Our results indicate that RBMS1 overexpression exacerbates Ang II-induced hypertrophy in primary rat cardiomyocytes, while also promoting cytoskeletal remodeling and disorganization of sarcomere alignment (Response figure 4a-f).

3. An Explanation of Biological Replication: We sincerely apologize for any ambiguity regarding the experimental replication. In all experiments, the number of cells and data points reflect biologically independent samples. This information has been clearly clarified in the revised Materials and Methods.

Response figure 4: (a) Western blotting and quantification showing protein levels of β -MHC and ANP in primary rat cardiomyocytes transfected with Ad-RBMS1 in response to Ang II stimulation (n=4). (b) Quantification of mRNA levels of MYH7, NPPA, NPPB, and RBMS1 (n=6). (c) Representative immunofluorescence staining of α -actinin and quantification of cell surface area (n=6), scale bar=50 μ m. (d) Immunofluorescence staining of ACTN2 and phalloidine showed the disorganization of sarcomere and cytoskeleton (n=6), scale bar=20 μ m. (e and f) Representative polarity histogram of sarcomere organization and quantification of sarcomere length (n=70).

6. Decreased sarcomere length is not usually associated with pathological hypertrophy. Were the cells/tissued relaxed before staining in the various experiments?

Reply: We sincerely appreciate the reviewer's insightful comments. Pathological hypertrophy is typically characterized by an initial reversible or compensatory phase, which progress to an irreversible or decompensatory phase. During the reversible or compensatory stage, myocardial hypertrophy is often accompanied by the accumulation of cytoskeletal proteins and elongation of sarcomeres. However, as the condition transitions into the irreversible or decompensatory stage, structural deterioration occurs, including sarcomere disorganization and shortening, along with a reduction in cytoskeletal protein content^[5-8]. In this study, an 8-week TAC model was employed, in which the mice had developed decompensated myocardial hypertrophy, thereby exhibiting cytoskeletal remodeling.

In the immunofluorescence experiments, paraformaldehyde was used for direct fixation of cells and tissues without prior relaxation. This approach was chosen to avoid potential disruption of the actin cytoskeleton that may occur with the use of relaxing agents such as cytochalasin.

7. The authors should discuss briefly the mechanism by which RBMS1 regulates splicing, if known. Also, it would be appropriate to contrast briefly the findings presented herein with the results obtained in fibroblasts in their recent paper.

Reply: We sincerely thank the reviewer for these insightful suggestions, which will undoubtedly strengthen the discussion and contextualize our findings. We acknowledge that elucidating the precise molecular mechanism underlying RBMS1's role in alternative splicing regulation is a complex endeavor and was not the primary focus of the current study, which centered on identifying RBMS1-dependent splicing events and their functional consequences in cardiomyocytes. We briefly discussed the mechanism by which RBMS1 regulates alternative splicing and compared the results in fibroblasts.

The focus of this study is to elucidate the mechanism by which RBMS1 participates in myocardial hypertrophy through CTTN splicing variant in cardiomyocytes. Interestingly, this phenomenon differs from the alternative splicing events we previously observed in fibroblasts. In our earlier research, we demonstrated that RBMS1 in fibroblasts promotes cardiac fibrosis and heart failure by regulating LM07 alternative splicing, which subsequently activates the AP-1/TGF- β pathway^[9]. However, in cardiomyocytes, we identified splicing events that are distinctly different from those in fibroblasts. This indicates that the alternative splicing regulated by RBMS1 is cell-type-specific, potentially due to variations in transcriptional profiles or splicing regulatory mechanisms across different cell types. As an RBP, RBMS1 typically interacts with target RNA in a sequence-specific manner through its RNA recognition motif (RRM)^[10]. The regulation of alternative splicing by RBPs is a highly complex process, involving spliceosome assembly, splice site selection, and the recruitment or inhibition of core spliceosomal components or other regulatory proteins. Further investigation is required to delineate the precise molecular mechanisms through which RBMS1 governs the splicing of critical target genes.

8. Regarding PI3K - p100 α and p100 γ have different roles in physiological and pathological hypertrophy. Which isoform is regulated by RBMS1-CTTN?

Reply: We thank the reviewer for this insightful question. Current research has preliminarily demonstrated that P100 α plays a predominant role in physiological hypertrophy rather than in pathological hypertrophy^[11], whereas P100 γ is primarily involved in the induction of pathological hypertrophy^[12]. During physiological hypertrophy, early activation of p110 α enhances myocardial contractility and helps maintain cardiac output in the short term^[13]. Evidence suggests that P100 γ , rather than P100 α , is primarily responsible for early compensatory pathological hypertrophy^[14]. However, the regulatory role

of P100 α in advanced pathological hypertrophy remains ambiguous. Nevertheless, sustained activation of p110 α leads to excessive reactive oxygen species (ROS) production, mitochondrial dysfunction, disruption of calcium homeostasis, and myocardial fibrosis, thereby facilitating the progression of decompensated pathological hypertrophy^[15].

Our findings, validated at both *in vitro* and *in vivo* levels, demonstrated that RBMS1 promoted the expression of P100 α , while knockdown of $\Delta e11$ significantly suppressed P100 α production (Response Fig. 5a-b). Furthermore, Alpelisib, a specific inhibitor of P100 α , competitively binds to the ATP-binding pocket of P110 α , thereby inhibiting the generation of PIP3. Our study confirmed that Alpelisib effectively alleviates myocardial hypertrophy induced by RBMS1 and $\Delta e11$ (Supplementary Fig. 13 and 14). Collectively, these results indicated that in decompensated pathological hypertrophy, the RBMS1- $\Delta e11$ axis promotes the conversion of PIP2 to PIP3 via P100 α , thereby activating the PI3K-AKT signaling pathway. This information has been clearly clarified in the revised Manuscript and the Materials and Methods.

Response figure 5: (a) Western blotting and quantification showing protein levels of P100 α and P100 γ in RBMS1 overexpression NMCs treatment with Alpelisib in response to Ang II stimulation (n=4). (b) Western blotting and quantification showing protein levels of P100 α and P100 γ in RBMS1

overexpression NMCs transfected with si- $\Delta e11$ in response to Ang II stimulation (n=4).

Minor comments:

1. Introduction lines 54-67 - The description of pathological cardiac hypertrophy and heart failure is not exactly accurate and should be re-written. For example, hypertrophy can occur in the absence of primary hypertension and volume overload.

Reply: We sincerely appreciate the reviewers' insightful comments, we have re-described cardiac hypertrophy and made revisions in the introduction.

Cardiac hypertrophy is defined as an increase in myocardial mass, representing the heart's adaptation to increased workload or pathological stimuli. Pathological cardiac hypertrophy arises in response to sustained pathological stresses, which commonly include hemodynamic overload (e.g., hypertension, valvular disease) or volume overload, but can also be triggered by diverse factors such as genetic mutations, neurohormonal dysregulation (e.g., excessive catecholamines, angiotensin II), metabolic disorders, or certain toxins. This maladaptive form of hypertrophy is strongly associated with cardiovascular disease (CVD) and is a major risk factor for the development of heart failure (HF), contributing significantly to global morbidity and mortality. Key features often observed in pathological hypertrophy include cardiomyocyte enlargement, accumulation of excess extracellular matrix, and, as it progresses towards HF, potential cardiomyocyte loss. Despite advancements in evidence-based medicine and postoperative care, the risk of relapse and readmission for patients with pathological cardiac hypertrophy remains unacceptably high in many regions due to limitations of current treatment strategies. Molecular mechanism studies at multiple levels are crucial and may facilitate the development of novel interventions and therapeutic approaches for this condition.

2. The paper needs to be edited for English usage throughout, including the methods, e.g.: Line 130 - "flanjoined" Line 134 - "selfing" Line 237 - "dismissed"

Reply: Thank you very much for your careful review of our manuscript and for your constructive comments. We greatly appreciate you pointing out the issues with English usage, particularly the incorrect terms highlighted. We have carefully proofread the entire manuscript, including the Methods section, to improve clarity, grammar, and terminology consistency. We believe that these edits have significantly improved the quality of the manuscript.

3. Lines 173-187 - GSE135055 and SCP1303 need to be introduced. Likewise, the hearts used for protein levels. What was the physiological state of the TAC mice when the analysis was done - concentric hypertrophy or dilated?

Reply: Thank you for carefully reviewing our manuscript and providing your valuable suggestions. The GSE135055 dataset was obtained from the NCBI database (<http://www.ncbi.nlm.nih.gov/geo/>). It includes nine samples of healthy normal left ventricular tissues derived from brain-dead donors with normal cardiac function, as well as 21 samples from patients with heart failure, among which 18 were diagnosed with dilated cardiomyopathy (DCM), and the remaining three were diagnosed with myocarditis. The SCP1303 dataset was obtained from the Broad Institute Single Cell Portal (https://singlecell.broadinstitute.org/single_cell), which contains samples from 15 patients diagnosed with hypertrophic cardiomyopathy (HCM) and 16 samples from organ donors without a history of heart failure. We have already revised the corresponding sections in the Methods and Materials.

In this study, an 8-week transverse aortic constriction (TAC) surgery was conducted. During this period, the TAC mouse model exhibited typical characteristics of dilated hypertrophy. This conclusion was drawn based on the

significantly increased left ventricular internal diameter (LVID) observed in the echocardiographic results, which is a key indicator of eccentric hypertrophy. However, no significant changes were observed in the thickness of the left ventricular posterior wall (LVPW), which may be due to the fact that the 8-week TAC mice were still in the early stages of dilated hypertrophy. Moreover, the eight-week TAC mice exhibited a decline in cardiac function, consistent with the features of dilated hypertrophy. Therefore, the physiological condition analyzed in our study, particularly with regard to molecular changes such as protein expression levels, clearly corresponds to eccentric myocardial hypertrophy induced by pressure overload.

4. Line 190 -What line of BL/6 mice were used, N or J? AAV dose should be stated for each experiment.

Reply: Thank you for your thorough evaluation of the research. The mouse strain utilized in this study was C57BL/6J, which is widely employed in models of myocardial hypertrophy and heart failure. Based on the titer of the adeno-associated virus 9, the mice were administered AAV9-RBMS1 via tail vein injection at a concentration of 5×10^{11} vg/mL, and AAV9-sh-CTTN- Δ e11 via tail vein injection at a concentration of 1×10^{11} vg/mL. This information has been detailed in the Materials and Methods and Figure Legends.

5. Supplementary Figure 1 - the figure legend does not match the figure.

Reply: We sincerely appreciate the reviewers' meticulous observations and have accordingly revised the Figure legend in Supplementary Figure 1.

6. Supplement - ultrasound description - Vevo 2000? The methods description is insufficient - the acquired ultrasound views need to be properly described, as well as how calculations were performed.

Reply: We sincerely appreciate the reviewers' insightful comments on methods for echocardiography. We carefully described the detailed methods of

echocardiography and made modifications in the supplementary materials of the methods and materials.

Mice were anesthetized by inhalation of 2.5% isoflurane. Subsequently, the chest hair was removed by shaving, and the skin was further cleaned with depilatory cream to minimize ultrasound artifacts. The animals were then placed in a supine position on a physiological monitoring platform. An ultrasound probe (MX550D, 26-52 MHz) was selected for imaging. A small amount of conductive gel was applied to the copper plate of the monitoring platform, and the mouse's paw was affixed to it to acquire electrocardiographic and respiratory signals. A rectal temperature probe was inserted to continuously monitor core body temperature. The ultrasound probe was positioned vertically, perpendicular to the left ventricle. The probe angle was then adjusted to clearly visualize the mitral valve, left ventricular cavity, ascending aorta, and aortic valve. B-mode images were recorded once the mitral valve reached maximal contraction. The system was then switched to M-mode, and the M-mode cursor was placed at the posterior edge of the papillary muscle to measure left ventricular end-diastolic volume (EDV), end-systolic volume (ESV), left ventricular posterior wall in systole (LVPWs), posterior wall in diastole (LVPWd), internal dimension in systole (LVIDs), and left ventricular internal dimension in diastole (LVIDd), thereby obtaining the parasternal long-axis view (PLAX) of the left ventricle. Left ventricular ejection fraction (EF%) = $(EDV - ESV)/EDV \times 100$. Fractional shortening (FS%) = $[(LVIDd - LVIDs)/LVIDd] \times 100$. Next, the probe was rotated 90° to obtain the parasternal short-axis view (PSAX) of the left ventricle. In this view, the M-mode cursor was placed centrally within the left ventricle, typically at the level of the papillary muscles. Measurements obtained in this view demonstrated good consistency with those obtained in the long-axis view. The apical four-chamber view was used to assess mitral inflow velocities (E wave, A wave, and E/A ratio) using pulsed-wave Doppler (PW). When necessary,

color Doppler imaging was employed to confirm the location of mitral inflow. On the standard PSAX view, the probe was advanced toward the aortic root until the aortic valve became visible. The sample volume was then placed beneath the mitral valve, allowing for the acquisition of mitral inflow data from the apical four-chamber view. Throughout the entire ultrasound procedure, vital signs of the mice were closely monitored, including body temperature ($37 \pm 0.5^{\circ}\text{C}$) and heart rate (350-450 bpm). Anesthesia depth was adjusted as needed to maintain physiological stability. Ultrasound parameters, including gain, depth, and Doppler angle ($<60^{\circ}$), were optimized to ensure high-quality image acquisition.

7. Supplement line 131 - Is "genes" correct?

Reply: We sincerely thank the reviewers for their careful review and insightful comments on the manuscript. We apologize for the oversight during the writing process that may have led to confusion among readers. For single-cell RNA-seq data (SCP1303), low-quality cells were removed based on the following criteria: (i) total RNA counts (nCount_RNA) > 500 and $< 90,000$ (Response Fig. 6a); (ii) number of detected genes (nFeature_RNA) > 300 and $< 10,000$ (Response Fig. 6b); and (iii) percentage of mitochondrial transcripts (percent.mt) $< 5\%$ (Response Fig. 6c). Cells meeting all thresholds were retained for downstream analysis. The relevant content in the Materials and Methods has been revised accordingly.

Response figure 6: (a-c) Violin plots show the distribution of the number of detected genes (nFeature_RNA, top), total RNA counts (nCount_RNA, middle), and the percentage of mitochondrial transcripts (percent.mt, bottom) across individual samples (P1290–P1735). These metrics were used to filter cells, retaining only those with nCount_RNA > 500 and < 90,000, nFeature_RNA > 300 and < 10,000, and percent.mt < 5% for downstream analyses.

8. Abbreviations need to be defined in the figures - Figure 4B and Supplementary Figure 4E,F are not interpretable.

Reply: We are grateful for the reviewers' insightful comments on the manuscript. In response, we have provided descriptions of the abbreviations in the Figure legend of Figures 4B, as well as Supplementary Figure 4E and 4F. Skipping Exon (SE), Alternative 3' Splice Site (A3SS), Mutually Exclusive Exons (MXE), Retained intron (RI), Alternative 5' Splice Site (A5SS). The

relevant content in the Figure Legend has been revised accordingly.

9. Line 450 -WGCNA needs to be defined. Likewise, the meaning of the colors needs to be described.

Reply: We sincerely thank the reviewer for this valuable comment. We performed Weighted Gene Co-Expression Network Analysis (WGCNA), a systems biology method used to construct gene co-expression networks and identify gene modules associated with specific traits. Using GSE135055 bulk data, we filtered the top 75% of genes based on the median absolute deviation (MAD), selecting those with a MAD greater than 0.01 for the R package WGCNA (v1.72-5) to construct a co-expression network, with a soft threshold of 16. Genes were grouped into modules by hierarchical clustering, and each module was assigned a distinct color (e.g., turquoise, blue, yellow, green, magenta, pink, red, black, brown, grey) for identification. The color intensity within each cell represents the strength and direction of the correlation between the module eigengene (first principal component of the module expression matrix) and the sample trait. These colors do not carry intrinsic biological meaning but serve as identifiers for different modules. The relevant content in the Materials and Methods has been revised accordingly.

10. ANP, BNP, and β MHC should preferably be referred to by their gene names when discussing mRNA levels or least be capitalized consistently in the figures and legends.

Reply: We sincerely appreciate the reviewers' valuable and insightful comments regarding naming consistency. During our discussion of the mRNA levels of ANP, BNP, and β MHC, we have consistently used their corresponding gene names, Nppa, Nppb, and Myh7. These changes have been implemented throughout the entire manuscript to ensure uniformity and accuracy in terminology.

11. Tables should be provided of the genes regulated by alternative

splicing.

Reply: Thank you for your thorough review and suggestions regarding our manuscript. We conducted an analysis of selective splicing on the RNA sequencing dataset from the RBMS1 overexpression study using rMATS (v4.1.2). The castat threshold was set to 0.01, and we filtered the results to ensure that the false discovery rate (FDR) was less than 0.05. We have compiled and provided a detailed table of all regulated splicing targets. This table includes the gene names, splicing event categories, trends, and statistical significance, which are detailed in Dataset EV1.

Reference

1. Han X, Qu L, Yu M, Ye L, Shi L, Ye G, et al. Thiamine-modified metabolic reprogramming of human pluripotent stem cell-derived cardiomyocyte under space microgravity. *Signal Transduct Target Ther.* 2024 Apr 8;9(1):86. doi: 10.1038/s41392-024-01791-7.
2. Fatima A, Xu G, Nguemo F, Kuzmenkin A, Burkert K, Hescheler J, et al. Murine transgenic iPS cell line for monitoring and selection of cardiomyocytes. *Stem Cell Res.* 2016 Sep;17(2):266-272. doi: 10.1016/j.scr.2016.07.007.
3. Cheng YC, Hsieh ML, Lin CJ, Chang CMC, Huang CY, Puntney R, et al. Combined Treatment of Human Induced Pluripotent Stem Cell-Derived Cardiomyocytes and Endothelial Cells Regenerate the Infarcted Heart in Mice and Non-Human Primates. *Circulation.* 2023 Oct 31;148(18):1395-1409. doi: 10.1161/CIRCULATIONAHA.122.061736.
4. Zhang F, Zhou H, Xue J, Zhang Y, Zhou L, Leng J, et al. Deficiency of Transcription Factor Sp1 Contributes to Hypertrophic Cardiomyopathy. *Circ Res.* 2024 Feb 2;134(3):290-306. doi: 10.1161/CIRCRESAHA.123.323272.

5. Hein S, Kostin S, Heling A, Maeno Y, Schaper J. The role of the cytoskeleton in heart failure. *Cardiovasc Res*. 2000 Jan 14;45(2):273-8. doi: 10.1016/s0008-6363(99)00268-0.
6. Gupta A, Gupta S, Young D, Das B, McMahon J, Sen S. Impairment of ultrastructure and cytoskeleton during progression of cardiac hypertrophy to heart failure. *Lab Invest*. 2010 Apr;90(4):520-30. doi: 10.1038/labinvest.2010.43.
7. McDonald KS, Kalogeris TJ, Veteto AB, Davis DJ, Hanft LM. Myosin binding protein-C modulates loaded sarcomere shortening in rodent permeabilized cardiac myocytes. *J Gen Physiol*. 2025 May 5;157(3):e202413678. doi: 10.1085/jgp.202413678.
8. Dai Y, Amenov A, Ignatyeva N, Koschinski A, Xu H, Soong PL, et al. Troponin destabilization impairs sarcomere-cytoskeleton interactions in iPSC-derived cardiomyocytes from dilated cardiomyopathy patients. *Sci Rep*. 2020 Jan 14;10(1):209. doi: 10.1038/s41598-019-56597-3.
9. Li L, Guo J, Feng J, Li T, Xu B, Li W, et al. Deficiency of the RNA-binding protein RBMS1 improves myocardial fibrosis and heart failure. *Eur Heart J*. 2025 Jun 5;ehaf370. doi: 10.1093/eurheartj/ehaf370.
10. Ray D, Kazan H, Cook KB, Weirauch MT, Najafabadi HS, Li X, et al. A compendium of RNA-binding motifs for decoding gene regulation. *Nature*. 2013 Jul 11;499(7457):172-7. doi: 10.1038/nature12311.
11. McMullen JR, Shioi T, Zhang L, Tarnavski O, Sherwood MC, Kang PM, et al. Phosphoinositide 3-kinase(p110alpha) plays a critical role for the induction of physiological, but not pathological, cardiac hypertrophy. *Proc Natl Acad Sci U S A*. 2003 Oct 14;100(21):12355-60. doi: 10.1073/pnas.1934654100.
12. Oudit GY, Crackower MA, Eriksson U, Sarao R, Kozieradzki I, Sasaki T, et al. Phosphoinositide 3-kinase gamma-deficient mice are protected from

isoproterenol-induced heart failure. *Circulation*. 2003 Oct 28;108(17):2147-52. doi: 10.1161/01.CIR.0000091403.62293.2B.

13. McMullen JR, Amirahmadi F, Woodcock EA, Schinke-Braun M, Bouwman RD, Hewitt KA, et al. Protective effects of exercise and phosphoinositide 3-kinase(p110alpha) signaling in dilated and hypertrophic cardiomyopathy. *Proc Natl Acad Sci U S A*. 2007 Jan 9;104(2):612-7. doi: 10.1073/pnas.0606663104.

14. Naga Prasad SV, Esposito G, Mao L, Koch WJ, Rockman HA. Gbetagamma-dependent phosphoinositide 3-kinase activation in hearts with in vivo pressure overload hypertrophy. *J Biol Chem*. 2000 Feb 18;275(7):4693-8. doi: 10.1074/jbc.275.7.4693.

15. Bass-Stringer S, Bernardo BC, Yildiz GS, Matsumoto A, Kiriazis H, Harmawan CA, et al. Reduced PI3K(p110 α) induces atrial myopathy, and PI3K-related lipids are dysregulated in athletes with atrial fibrillation. *J Sport Health Sci*. 2025 Jan 16;14:101023. doi: 10.1016/j.jshs.2025.101023.

Reviewer #3 (Remarks to the Author):

This is an excellent report highlighting the role of RBMS1 in cardiac hypertrophy using an impressive array of transgenic and experimental mouse models of heart disease. The work provides also very deep mechanistic insights on how RBMS1 facilitating CTTN splice-switching and how it is linked to sarcomere dynamics/function and to heart disease. I have only some minor points for improvement:

Reply: We thank the reviewer for your constructive comments and have addressed them carefully below.

1. Many transgenic mouse models were used in the study, e.g. RBMS1 cKO which was bought from a company. However, all detailed information was missing how exactly this floxed line was made. Even if it was acquired commercially, the authors have to provide more detailed information in the supplement, e.g. on exact targeted exon, CRISPR construct, targeting strategy, methods used to generate this and other mouse lines which were not described in detail.

Reply: We sincerely appreciate the reviewers' diligent examination of the research methodology. Recognizing that ambiguous descriptions could potentially lead to confusion among readers, we have provided a comprehensive explanation regarding the construction method of transgenic mice.

MYH6-Cre mice and RBMS1^{flox/flox} transgenic mice were purchased from Cyagen Biosciences (Suzhou, China). Based on the genomic structure of RBMS1 and the conservation of its functional protein domains, two single-guide RNAs (sgRNAs) with high on-target scores and low off-target effects were designed using the online tool CRISPOR (<http://crispor.tefor.net/>). These sgRNAs were selected to target the 5' and 3' ends of exon 3 of RBMS1,

respectively, with their protospacer adjacent motif sequences located outside the intended loxP insertion sites to facilitate dual DNA cleavage and promote efficient homology-directed repair. The sgRNAs were prepared via in vitro transcription, which involved template preparation, transcription, purification, and quality control. A donor plasmid was constructed containing homologous arms, two loxP sites, and exon 3 of RBMS1. The assembly of these elements into a plasmid backbone was achieved using Golden Gate cloning. The donor plasmid was linearized by restriction enzyme digestion to enhance recombination efficiency. Purified sgRNAs were mixed with Cas9 protein at a molar ratio of 2:1 and incubated at room temperature for 10-15 minutes to form ribonucleoprotein complexes. The linearized donor plasmid was then added to the RNP mixture to prepare the final injection solution. Female C57BL/6J mice (3-4 weeks old) were superovulated by sequential injection of pregnant mare serum gonadotropin followed 48 hours later by human chorionic gonadotropin, and then mated with fertile adult male mice. The following day, the females were euthanized, and zygotes were collected from the oviducts. The zygotes were maintained in KSOM medium droplets in a 37°C incubator with 5% CO₂ before manipulation. The prepared injection mixture was delivered into the pronuclei of zygotes via pronuclear microinjection. After injection, the embryos were transferred to M16 medium and cultured in the incubator for 0.5–1 hours. Subsequently, the embryos were transplanted into the oviducts of pseudopregnant female mice. The resulting F0 pups were genotyped using RBMS1-specific primers, followed by breeding and establishment of the mouse line. The RBMS1^{flx/flx} transgenic mice and the MYH6-Cre tool mice were intercrossed to generate the αMHC-RBMS1^{flx/flx} transgenic mice.

2. The description of methods for echocardiography is weird and not standard for the field. Which instrument and which type of anesthesia was used for that in detail?

Reply: We sincerely appreciate the reviewers' insightful comments on methods for echocardiography. We carefully described the detailed methods of echocardiography and made modifications in the supplementary materials of the methods and materials.

Mice were anesthetized by inhalation of 2.5% isoflurane. Subsequently, the chest hair was removed by shaving, and the skin was further cleaned with depilatory cream to minimize ultrasound artifacts. The animals were then placed in a supine position on a physiological monitoring platform. An ultrasound probe (MX550D, 26-52 MHz) was selected for imaging. A small amount of conductive gel was applied to the copper plate of the monitoring platform, and the mouse's paw was affixed to it to acquire electrocardiographic and respiratory signals. A rectal temperature probe was inserted to continuously monitor core body temperature. The ultrasound probe was positioned vertically, perpendicular to the left ventricle. The probe angle was then adjusted to clearly visualize the mitral valve, left ventricular cavity, ascending aorta, and aortic valve. B-mode images were recorded once the mitral valve reached maximal contraction. The system was then switched to M-mode, and the M-mode cursor was placed at the posterior edge of the papillary muscle to measure left ventricular end-diastolic volume (EDV), end-systolic volume (ESV), left ventricular posterior wall in systole (LVPWs), posterior wall in diastole (LVPWd), internal dimension in systole (LVIDs), and left ventricular internal dimension in diastole (LVIDd), thereby obtaining the parasternal long-axis view (PLAX) of the left ventricle. Left ventricular ejection fraction (EF%) = $(EDV - ESV)/EDV \times 100$. Fractional shortening (FS%) = $[(LVIDd - LVIDs)/LVIDd] \times 100$. Next, the probe was rotated 90° to obtain the parasternal short-axis view (PSAX) of the left ventricle. In this view, the M-mode cursor was placed centrally within the left ventricle, typically at the level of the papillary muscles. Measurements obtained in this view demonstrated good consistency with those obtained in the long-axis view. The

apical four-chamber view was used to assess mitral inflow velocities (E wave, A wave, and E/A ratio) using pulsed-wave Doppler (PW). When necessary, color Doppler imaging was employed to confirm the location of mitral inflow. On the standard PSAX view, the probe was advanced toward the aortic root until the aortic valve became visible. The sample volume was then placed beneath the mitral valve, allowing for the acquisition of mitral inflow data from the apical four-chamber view. Throughout the entire ultrasound procedure, vital signs of the mice were closely monitored, including body temperature ($37 \pm 0.5^{\circ}\text{C}$) and heart rate (350-450 bpm). Anesthesia depth was adjusted as needed to maintain physiological stability. Ultrasound parameters, including gain, depth, and Doppler angle ($<60^{\circ}$), were optimized to ensure high-quality image acquisition.

3. I appreciate patient data provided for DCM patient group. However, reporting of healthy donor heart tissues seems rather intransparent. Where did these subjects come from - hearts rejected for transplantation, surgeries for other indications or similar? More information, also on the ethical approval is needed here.

Reply: We sincerely appreciate the reviewer's insightful comments concerning the procurement of healthy donor cardiac tissues. Healthy donor heart tissues were obtained from hearts with no history of cardiac disease at Zhongnan Hospital of Wuhan University, which were originally intended for heart transplantation. These organs were deemed unsuitable for transplantation due to donor-recipient incompatibility. The study was approved by the Ethics Committee of Zhongnan Hospital of Wuhan University (Approval No. 2022075K) in full accordance with the ethical principles outlined in the Declaration of Helsinki. The relevant content in the Materials and Methods has been revised accordingly.

23rd Sep 2025

Dear Prof. Liang,

Thank you for the submission of your revised manuscript to EMBO Molecular Medicine. I am pleased to inform you that we will be able to accept your manuscript pending the following final amendments:

1) Please implement referee #3 suggestion.

2) Authors:

- E-mail correspondence to Tengfei Pan could not be delivered. Please update their e-mail addresses and make sure to enter correct e-mail addresses for all authors in our submission system.

- Are Jinping Liu and Baofeng Yang corresponding authors? In that case, this information should be added to the manuscript title page, and an ORCID number should be linked to Jinping Liu's account in our system.

3) Figures: Please submit several Appendix figures (e.g. 5) as EV Figures. EV figures should be uploaded as individual, high resolution figure files and their legends placed after the main figure legends in the manuscript file under the heading "Expanded View Figure Legends". Please check "Author Guidelines" for more information.

<https://www.embopress.org/page/journal/17574684/authorguide#expandedview>

4) Author Checklist: Please complete the form by entering information about Author, Journal and Manuscript Number at the top of the page.

5) In the main manuscript file, please do the following:

- Please address all comments suggested by our data editors listed below:

o Figure legends:

1. Please note that the exact p values are not provided in the legends of figures 1A, C, D, F, I, L, M, N; 2E, G, H, I, L, M; 3C, E, F, G, I, J, K; 4H, I; 5C, 6C, E, F, G, I, L; 7C, E, F, G, I; 8C, E, D, G, I; 9B, C, E, H, J, L, M.

2. Please indicate the statistical test used for data analysis in the legends of figures 1A, B, C, D-F; G, I, J, K, L-N; 2B, C, E, G, H, I, K, L, M; 3C, E, F, G, I, J, K; 4C, H, I; 5A, C, E, F, G, H, J; 5C, 6C, E, F, G, I, L; 7C, E, F, G, I; 8C, E, D, G, H, I; 9B, C, E, H, J, L, M.

3. Please note that the box plots need to be defined in terms of minima, maxima, centre, bounds of box and whiskers, and percentile in the legends of figures 1A, C; 8C.

4. Please note that the scale bar needs to be defined for figure 4G.

5. Please note that the yellow and red arrows are not defined in the legend of figures 4D, F; 6J, 7J. This needs to be rectified.

- Limit keywords to max. 5.

- Add callouts Fig 3A.

- In Methods, provide the antibody dilutions that were used for each antibody

- In Methods, provide the statement that in addition to the informed consent and WMA Declaration of Helsinki the experiments also conformed to the principles set out in the Department of Health and Human Services Belmont Report.

- Author contributions: Please remove it from the manuscript and specify author contributions in our submission system. CRediT has replaced the traditional author contributions section because it offers a systematic machine-readable author contributions format that allows for more effective research assessment. You are encouraged to use the free text boxes beneath each contributing author's name to add specific details on the author's contribution. More information is available in our guide to authors:

<https://www.embopress.org/page/journal/17574684/authorguide#authorshipguidelines>

- Indicate in legends number and nature of replicates and exact p= values, not a range, along with the statistical test used. To keep the figures "clear" some authors found providing an Appendix table Sx with all exact p-values preferable. You are welcome to do this if you want to.

- Add heading "Figure legends" for the main figure legends.

6) Appendix:

- Move all Materials and Methods to the main Methods section in the manuscript file and remove the yellow highlights.

- Please provide an appendix file with higher figure resolution.

7) Tables: Please rename Table EV1 to Dataset Eva and update its callouts in the main text.

8) Funding: Please make sure that information about all sources of funding including all project numbers are complete in both our submission system and in the manuscript

9) Synopsis:

- Synopsis text: Please remove it from the main manuscript file and upload as a separate .doc file.

- Synopsis image: Please remove it from the main manuscript file and upload as a high-resolution .jpeg file 550 px-wide x 300-600 pixels high.

10) As part of the EMBO Publications transparent editorial process initiative (see our Editorial at

<http://embomolmed.embopress.org/content/2/9/329>), EMBO Molecular Medicine will publish online a Review Process File (RPF) to accompany accepted manuscripts. This file will be published in conjunction with your paper and will include the anonymous referee reports, your point-by-point response and all pertinent correspondence relating to the manuscript. Let us know whether

you agree with the publication of the RPF and as here, if you want to remove or not any figures from it prior to publication. Please note that the Authors checklist will be published at the end of the RPF.

11) Please provide a point-by-point letter INCLUDING my comments as well as the reviewer's reports and your detailed responses (as Word file).

I look forward to reading a new revised version of your manuscript as soon as possible.

Yours sincerely,

Zeljko Durdevic

Zeljko Durdevic
Senior Editor
EMBO Molecular Medicine

*** Instructions to submit your revised manuscript ***

1) a .docx formatted version of the manuscript text (including Figure legends and tables)

2) Separate figure files*

3) supplemental information as Expanded View and/or Appendix. Please carefully check the authors guidelines for formatting Expanded view and Appendix figures and tables at <https://www.embopress.org/page/journal/17574684/authorguide#expandedview>

4) a letter INCLUDING the reviewer's reports and your detailed responses to their comments (as Word file).

5) The paper explained: EMBO Molecular Medicine articles are accompanied by a summary of the articles to emphasize the major findings in the paper and their medical implications for the non-specialist reader. Please provide a draft summary of your article highlighting

This may be edited to ensure that readers understand the significance and context of the research.

Please refer to any of our published articles for an example.

6) Author contributions: the contribution of every author must be detailed in a separate section.

7) EMBO Molecular Medicine now requires a complete author checklist (<https://www.embopress.org/page/journal/17574684/authorguide>) to be submitted with all revised manuscripts. Please use the checklist as guideline for the sort of information we need WITHIN the manuscript. The checklist should only be filled with page numbers where the information can be found. This is particularly important for animal reporting, antibody dilutions (missing) and exact values and n that should be indicated instead of a range.

8) Every published paper now includes a 'Synopsis' to further enhance discoverability. Synopses are displayed on the journal webpage and are freely accessible to all readers. They include a short stand first (maximum of 300 characters, including space) as well as 2-5 one sentence bullet points that summarise the paper. Please write the bullet points to summarise the key NEW findings. They should be designed to be complementary to the abstract - i.e. not repeat the same text. We encourage inclusion of key acronyms and quantitative information (maximum of 30 words / bullet point). Please use the passive voice. Please attach these in a separate file or send them by email, we will incorporate them accordingly.

You are also welcome to suggest a striking image or visual abstract to illustrate your article. If you do please provide a jpeg file 550 px-wide x 300-600px high.

9) A Conflict of Interest statement should be provided in the main text

10) Please note that we now mandate that all corresponding authors list an ORCID digital identifier. This takes <90 seconds to complete. We encourage all authors to supply an ORCID identifier, which will be linked to their name for unambiguous name identification.

Currently, our records indicate that the ORCID for your account is 0000-0003-4446-7035.

Link Not Available

11) Include a Reagents and Tools Table as part of the Methods section, which can be downloaded from our author guidelines (<https://www.embopress.org/page/journal/17574684/authorguide#structuredmethods>)

Photos 400-800 DPI

*Additional important information regarding figures and illustrations can be found at <https://bit.ly/EMBOPressFigurePreparationGuideline>. See also figure legend preparation guidelines: <https://www.embopress.org/page/journal/17574684/authorguide#figureformat>

***** Reviewer's comments *****

Referee #1 (Remarks for Author):

The authors have addressed my previous questions. I do not have additional questions.

Referee #3 (Remarks for Author):

The authors have addressed all my comments satisfactorily. I was also asked to evaluate whether the comments raised by the Reviewer #2 were also adequately addressed. In general, the authors made a great job to address all his/her comments as well. However, the comment regarding the figure legends (which were too short and occasionally lacking some details) was meant for all figures, not just the one figure legend which was expanded. Maybe the authors could carefully revise the remaining figure legends and add some additional information. At minimum, they should mention which statistical test(s) was used to obtain P values shown in all figures.

******* Reviewer's comments *********Referee #1 (Remarks for Author):**

The authors have addressed my previous questions. I do not have additional questions.

Reply: Thank you for your final confirmation and for your time and effort in reviewing our manuscript. We are very pleased to hear that you have no further questions. Your previous comments were immensely helpful in improving the quality of our work.

Referee #3 (Remarks for Author):

The authors have addressed all my comments satisfactorily. I was also asked to evaluate whether the comments raised by the Reviewer #2 were also adequately addressed. In general, the authors made a great job to address all his/her comments as well. However, the comment regarding the figure legends (which were too short and occasionally lacking some details) was meant for all figures, not just the one figure legend which was expanded. Maybe the authors could carefully revise the remaining figure legends and add some additional information. At minimum, they should mention which statistical test(s) was used to obtain P values shown in all figures.

Reply: Thank you very much for your positive feedback and for acknowledging that we have satisfactorily addressed all your previous comments. We are also grateful that you took the time to evaluate our responses to Reviewer #2's concerns. We sincerely appreciate you pointing out the need for more comprehensive improvements to the figure legends. We have carefully revised all figure legends to ensure they are sufficiently detailed and declared the statistical test used to generate the *P*-values.

17th Oct 2025

Dear Prof. Liang,

We are pleased to inform you that your manuscript is accepted for publication and is now being sent to our publisher to be included in the next available issue of EMBO Molecular Medicine.

Zeljko Durdevic
Senior Editor
EMBO Molecular Medicine
